# PI3K Inhibitors in Cancer: Clinical Implications and Adverse Effects

**DOI:** 10.3390/ijms22073464

**Published:** 2021-03-27

**Authors:** Rosalin Mishra, Hima Patel, Samar Alanazi, Mary Kate Kilroy, Joan T. Garrett

**Affiliations:** Department of Pharmaceutical Sciences, College of Pharmacy, University of Cincinnati, Cincinnati, OH 45267-0514, USA; mishrarn@ucmail.uc.edu (R.M.); patel2h2@mail.uc.edu (H.P.); alanazsa@mail.uc.edu (S.A.); kilroymk@mail.uc.edu (M.K.K.)

**Keywords:** cancer, *PIK3CA*, resistance, PI3K inhibitors

## Abstract

The phospatidylinositol-3 kinase (PI3K) pathway is a crucial intracellular signaling pathway which is mutated or amplified in a wide variety of cancers including breast, gastric, ovarian, colorectal, prostate, glioblastoma and endometrial cancers. PI3K signaling plays an important role in cancer cell survival, angiogenesis and metastasis, making it a promising therapeutic target. There are several ongoing and completed clinical trials involving PI3K inhibitors (pan, isoform-specific and dual PI3K/mTOR) with the goal to find efficient PI3K inhibitors that could overcome resistance to current therapies. This review focuses on the current landscape of various PI3K inhibitors either as monotherapy or in combination therapies and the treatment outcomes involved in various phases of clinical trials in different cancer types. There is a discussion of the drug-related toxicities, challenges associated with these PI3K inhibitors and the adverse events leading to treatment failure. In addition, novel PI3K drugs that have potential to be translated in the clinic are highlighted.

## 1. Introduction

PI3K/AKT signaling is involved in important physiological and pathophysiological functions that drive tumor progression such as metabolism, cell growth, proliferation, angiogenesis and metastasis [1,2]. Pharmacological or genetic suppression of this signaling causes cancer cell death and regression of tumor growth. The PI3K pathway is activated via point mutation of the *PIK3CA* gene or inactivation of the phosphatase and tensin homolog (*PTEN*) gene [3]. Activation of this pathway occurs in approximately 30–50% human cancers and results in resistance to various anti-cancer therapies [4,5].

PI3K proteins are classified into three main classes (I, II and III) based on the substrate specificities and structural characteristics. Class I PI3Ks are categorized into two subtypes (A and B) based on the mode of regulation. Class IA PI3Ks form dimers containing a regulatory (p85α, p85β, p55α, p55γ, p50α) and a catalytic (p110α, p110β, p110δ) subunit. Class I PI3Ks act downstream of both G protein-coupled receptors (GPCRs) and receptor tyrosine kinases (RTKs). In addition, these regulatory subunits play an important role in stabilization of the p110 catalytic subunits and in suppression of the basal lipid kinase activity [6]. Class IB PI3Ks which are activated downstream of GPCRs form heterodimers consisting of p110γ as catalytic subunit with other regulatory subunits such as p101, p87 or p84 [7,8]. This class consists of three isoforms (PI3K-C2α, PI3K-C2ß and PI3K-C2γ). These isoforms have a RAS-binding domain (RBD), a helical domain, and a catalytic domain but lack a regulatory domain [7]. The class III PI3Ks also known as vacuolar protein sorting 34 (VPS34) heterodimerize with membrane-associated VPS15 regulatory subunit. This VPS34-VPS15 complex is ubiquitously expressed in mammals regulating several functions such as phagocytosis, autophagy, endocytosis and intracellular trafficking [7,9]. Some reports have considered mTOR as class IV PI3K-related kinase whose catalytic domain is associated with ataxia-telangiectasia (ATM), fluorescence recovery after photo bleaching (FRAP) and transformation/transcription domain-associated protein (TRRAP) and FK506-binding protein 12 (FKBP12)/rapamycin (FRB)-binding domain and a C-terminal FAT domain (FATC). There are two Huntington elongation factor 3, PR65/A subunit of protein phosphatase 2A, TOR (HEAT) repeats in the N-terminal region which modulate the protein–protein interactions in mTOR signaling [10,11].

Small-molecule inhibitors targeting PI3K consist of pan, isoform-specific and dual PI3K/mTOR inhibitors. Wortmanin and LY294002 were the first-generation PI3K inhibitors that belong to the non-isoform-specific category [12,13,14]. Wortmanin with irreversible inhibition, lack of selectivity and adverse effects resulted in termination of its clinical trials [15]. Poor solubility, bioavailability and rapid degradation of LY294002 prevented its further biological evaluation and clinical studies [16]. To date, five PI3K inhibitors (Copanlisib, Idelalisib, Umbralisib, Duvelisib and Alpelisib) have been approved by the United States Food and Drug Administration (FDA). This has encouraged clinicians and researchers to test other PI3K inhibitors in both preclinical and clinical settings with the goal to identify a potent PI3K inhibitor with significant clinical efficacy, low toxicities and optimum bioavailability. In this review, we discuss the challenges associated with successful development of PI3K inhibitors including the aberrant mutations/amplification of genes such as PI3K and/or PI3K-driven molecules, the role of these mutations and other molecules in PI3K-mediated drug sensitivity, and compensatory signaling mechanisms resulting in PI3K inhibitor resistance causing treatment failure in the clinic. In addition, we have summarized active, not-recruiting and completed trials along with detailed clinical outcomes for PI3K inhibitors in a wide variety of cancers along with the adverse events.

## 2. Signaling Molecules and Factors Contributing to PI3K Inhibitor Resistance

PI3K signaling is deregulated in a variety of cancers. There are several factors that ultimately result in resistance to PI3K inhibitors including (1) inactivation or loss of PTEN activity, (2) mutations and amplification of PI3K, (3) drug-related toxicities, (4) feedback upregulation leading to compensatory mechanisms, (5) non-coding RNA in regulating PI3K signaling, (6) enhanced insulin production upon PI3K inhibition, (7) selection of patient population in clinical trials and (8) miscellaneous resistance mechanisms, as summarized below.

### 2.1. Inactivation or Loss of PTEN Activity

*PTEN* is a well-characterized tumor suppressor which is a negative modulator of the PI3K pathway. *PTEN* regulates PI3K via dephosphorylating phosphatidylinositol-3, 4, 5-trisphosphate (PIP3). The loss of or gain in function is seen in several cancers including breast, brain and prostate [17,18]. Increased PI3K signaling and tumorigenesis is observed in *PTEN* knock-in mice with *PTEN* mutations [18]. Loss of PTEN lipid phosphatase activity causes the activation of AKT in *PTEN*-null cancers [18,19]. A study by Juric et al. indicates that loss of *PTEN* can lead to clinical resistance to a PI3K inhibitor, Alpelisib in breast cancer [20]. These studies indicate the possible role of *PTEN* in modulating response to PI3K inhibitors in different cancers.

### 2.2. Mutations and Amplification of PI3K

The p110α subunit of *PIK3CA* is frequently mutated and amplified (~30%) in a variety of cancers [21,22,23]. There are hotspot mutations such as *PIK3CA*E545K in exon 9 in the helical PI3K homology domain that suppresses inhibition of p110α by p85 regulatory subunit. Another mutation is *PIK3CA*H1047 present in the catalytic subunit in exon 20 which enhances the interaction of p110α with the lipid membrane [24,25]. E542K is another crucial PI3K mutation with significant oncogenic potential associated with elevated in vitro catalytic activity [26,27].The helical and kinase mutations in p110α domain induce oncogenic activity based on different interaction between the PI3K regulatory subunit, p85 with RAS-GTP. The oncogenic activity induced by helical domain mutations is independent of binding to p85 subunit. However, requires interaction with RAS-GTP. However, the kinase domain mutations are active without RAS-GTP binding but dependent on their interaction with p85. In addition, co-existence of both the domain mutations increases the p110 function and tumorigenic activity synergistically [27]. A recent study has shown that C2 domain deletions in *PIK3CA* activate PI3K signaling significantly and also enhance the sensitivity to PI3Kα inhibitors [28]. The deregulation of PI3K signaling leads to several oncogenic activities such as cancer cell proliferation, invasion, migration, glucose transport and angiogenesis that regulate tumor progression [23,24].

Genetically engineered mouse models (GEMMs) using *PIK3CA* mutations demonstrate important tumorigenic potential associated with PI3K signaling. For example, the *PIK3CA*H1047R knock-in mouse model promoted conditional activation of the PI3K pathway and induced breast tumorigenesis [29]. Another study indicated that *PIK3CA* H1047R mutation with *PTEN* deletion led to the development of ovarian adenocarcinoma and granulosa cell tumor in a mouse model [30]. Further, overexpression of wild-type catalytic subunits (p110ß, p110δ and p110γ) is known to induce an oncogenic phenotype [31]. The role of p110ß is well established in cancers including breast and prostate [32,33]. Although the mechanism of action of this subunit is not well known, some reports still indicate that it works via GPCR signaling [34]. E633K is a p110ß helical mutation first reported in HER2+ breast cancer [35,36]. *PIK3CA*D1067V is another recurrent somatic mutation in the p110β subunit which induced in vitro and in vivo cancer cell growth via the activated PI3K signaling pathway. Pharmacologic inhibition using a specific p110β inhibitor (TGX-221) inhibited growth in patient-derived renal cell carcinoma cells with endogenous *PIK3CA*D1067V and epidermal growth factor receptor (EGFR)-mutant lung adenocarcinoma cells as well as in NIH-3T3 cells engineered to express this mutation. In addition, expression of this mutation is known to promote Erlotinib resistance in EGFR-mutant lung adenocarcinoma cells [37]. PI3Kδ plays a crucial role in myeloid cell activities such as inflammation-driven cell infiltration, neutrophil oxidative burst, immune complex-mediated macrophage activation, mast cell maturation and degranulation [38,39]. It also plays an important role in inducing cancer cell proliferation and AKT activation in myeloid leukemia [40]. PI3Kδ point mutations have been identified in a panel of diffuse B-cell lymphomas [41]. It is known to regulate in vitro chemotactic migration in response to EGF and share similar biological function in breast cancer cells and macrophages [42].

PI3Kγ is expressed in immune cells of myeloid origin that regulates innate immunity in both cancer and inflammatory cells [39]. PI3Kγ regulates solid tumor neovascularization, contributes to chemotactic response and reactive oxygen species (ROS) production in neutrophils [43,44]. All these studies indicate the key role of PI3K and PI3K isoform-mediated signaling along with site-specific driver mutations that leads to oncogenic transformation of cancer cells which could ultimately lead to drug resistance.

### 2.3. Drug-Related Toxicities Affect Sustained Target Suppression

One major hurdle in the development PI3K inhibitors is the inability to achieve optimal drug-target blockade in tumors due to drug-related toxicities in patients. Toxicities from small-molecule PI3K inhibitors depend on their PI3K isozyme specificity. For example, the adverse effects associated with PI3Kα inhibitors are mostly rash and hyperglycemia, and the side effects associated with δ subunits are mostly gastrointestinal, transaminitis and myelosuppression. The pan-PI3K inhibitors share common dose-dependent toxicities such as fatigue, diarrhea, rash and hyperglycemia. Dual PI3K inhibitors have a broader toxicity profile compared to isoform or pan-PI3K inhibitors. Isoform-specific PI3Kδ inhibitors have a better effect in B-cell malignancy compared to pan or dual PI3K inhibitors because of the toxicity profile [45]. Several studies have indicated that intermittent dosing schedules of PI3K drugs have a better safety profile over a continuous dosing pattern. A study by Hudson et al. showed that when administrated intermittently PI3Kα/δ inhibitor AZD8835 showed better anti-tumor efficacy in breast cancer xenograft models as a monotherapy and in combination with an ER inhibitor (Fulvestrant) and CDK4/6 inhibitor (Palbociclib) that suppressed AKT activation and induced cell death [46]. Another study demonstrated that intermittent administration of mitogen-activated protein kinase kinase (MEK) inhibitor, GDC-0973 with a PI3K inhibitor (GDC-0941) triggered cell death and tumor growth inhibition [47]. However, dosing schedules, suboptimal dose selection and on-target toxicities of several PI3K inhibitors failed to cause the complete and sustained inhibition of PI3K signaling, challenging the efficacy of these PI3K inhibitors in the clinic.

### 2.4. Feedback Upregulation Leading to Compensatory Mechanism

Drug resistance associated with pharmacological inhibition of PI3K inhibitors is caused due to feedback upregulation of the PI3K/AKT/mTOR pathway involving RTKs, growth factors and downstream transcription factors. One such transcription factor is forkhead box (FOXO), which represses RTKs or other adaptors that activate the PI3K pathway, such as human epidermal growth factor receptor 3 (HER3), insulin-like growth factor 1 receptor (IGF1R), EGFR, fibroblast growth factor receptor (FGFR) and insulin receptor (InsR) [48,49]. Therefore, inhibiting PI3K/AKT signaling suppresses the FOXO phosphorylation, causing FOXO-dependent repression of RTKs leading to derepression of molecules downstream of AKT such as S6K and growth factor receptor-bound protein 10 (GRB10), ultimately resulting in activation of multiple RTKs and partial maintenance of PIP3 [50]. Partial maintenance of PIP3 is also mediated by PI3K isoform p110ß in luminal breast cancer cells where PI3K is hyperactivated due to HER2-HER3 dimers [51]. Another study showed the key role of FOXO-mediated adaptive resistance in matrix-attached cancer cells in response to inhibition of PI3K/mTOR signaling [52]. Another FOXO-dependent adaptive response is via upregulation of Rictor, resulting in enhanced AKT activation in renal cancer cells [53].

Another transcription factor, signal transducer and activator of transcription 5 (STAT5) plays a crucial role in mediating resistance to PI3K inhibitors in vitro. A study by Britschgi et al. showed that STAT5 is activated by Janus kinase 2 (JAK2) that evoked a positive feedback loop to dampen the efficacy of PI3K/mTOR inhibition via secretion of interleukin-8 (IL-8) in several cell lines and primary breast tumors. The data also showed that pharmacological and genetic inhibition of JAK2 combined with PI3K/mTOR inhibition abrogated this feedback loop [54]. Treatment with the AKT inhibitor AZD5363 activates several RTKs, estrogen receptor 1 (ESR1) mRNA, and ERα-mediated transcription of insulin-like growth factor 1 (IGF1) and IGF2 ligands [55]. In another study, expression of ESR1 mRNA and ER-related proteins are increased upon PI3K inhibition using Alpelisib. These drug-related transcriptional alterations were abrogated via anti-ER drugs [56]. Treatment with Alpelisib in ER+ breast cancer cells or patients with primary tumors resulted in activation of ERα via lysine methyltransferase 2D (KMT2D)-dependent FOXA1-PBX1 complex [57]. All these data highlight the role of ERα-mediated feedback upregulation in response to PI3K inhibition that might result in drug insensitivity in PIK3CA-driven cancers.

### 2.5. Non-Coding RNA in Regulating PI3K Signaling

Non-coding RNAs (ncRNAs) are commonly employed for RNAs that do not encode proteins which are involved in cancer drug resistance [58]. These ncRNAs include siRNA, piRNA, miRNA and lncRNA. In general, ncRNAs function to regulate gene expression at the transcriptional and post-transcriptional level. Several lncRNAs are known to interact with PI3K to activate signaling [59]. For example, lncRNA CRNDE is overexpressed in several cancers to activate cell proliferation and growth via PI3K-dependent pathways [60,61,62]. lncRNA OIP5-AS1 is another ncRNA that activates PI3K signaling and cause Cisplatin resistance in osteosarcoma [63]. OIP5-AS1 loss causes miR-410 accumulation that facilitates cell cycle progression, proliferation and apoptosis inhibition by targeting Krüppel-Like Factor-10 (KLF10) via activating the PI3K/AKT/mTOR pathway in multiple myeloma [64]. lncRNA CCAT1 regulates thyroid and squamous cancer cell migration via activation of the PI3K/AKT pathway [65,66]. lncRNA H19 inhibits melanoma cell migration and invasion via inactivating PI3K/AKT/NFĸB signaling [67]. lncRNA HOTAIR promotes Cisplatin resistance in gastric cancer by targeting miR-126/miR-34a through activation of the PI3K/AKT pathway [68,69]. It is also involved in multidrug resistance to Imatinib in a PI3K/AKT-dependent signaling in myeloid leukemia cells [70]. It also regulates cancer cell growth, metastasis and apoptosis via PI3K/AKT-dependent pathway in melanoma, adenocarcinoma and glioma [71,72,73]. lncRNA NEAT1 plays a key role in PI3K/AKT-mediated tumorigenesis in several cancer types [74,75]. HULC is another lncRNA that is involved in bladder cancer cell proliferation via regulation of ZIC2 and PI3K/AKT signaling [76]. It induces leukemia cell proliferation and suppresses angiogenesis in gliomas via an AKT-dependent pathway [77,78]. AB073614 and PTTG3P are known to promote cancer cell proliferation, migration and invasion via PI3K/AKT-dependent signaling in colorectal and hepatocellular carcinoma, respectively [79,80]. MALAT1 is another crucial lncRNA which is upregulated in several cancers and induces tumorigenic phenomenon via miRNA-mediated PI3K/AKT-dependent signaling [81,82,83,84,85]. lncRNA ATB induces bladder cancer cell proliferation, migration, and invasion via AKT/mTOR-mediated signaling [86]. lncRNA BC087858 induces acquired resistance to EGFR-TKIs by activating PI3K/AKT and epithelial–mesenchymal transition (EMT)-dependent signaling in lung cancer [87].

Linc00659 and Linc00152 induce cancer cell growth and inhibit apoptosis in colorectal and lung cancer, respectively, via PI3K/AKT-mediated pathway activation [88,89]. In similar lines, Linc00462 and Linc01296 regulate different hallmarks in tumor progression via AKT signaling [90,91]. There are several other lncRNAs including UCA1, ecCEBPA, Ftx, RMEL3, lncARSR, BDLNR, ANRIL, ROR, MYD88, RNA-422 and PlncRNA-1 that are upregulated and mediate PI3K-dependent signaling in different cancers [92]. Other lncRNAs which are downregulated yet known to mediate tumorigenesis via PI3K/AKT-dependent signaling include linc003121 in thyroid cancer [93], RP4 in colorectal cancer [94], MEG3 in endometrial cancer [95], GAS5 in various cancers [96,97]. In summary, there is a long list of ncRNAs including those discussed above which are promising candidates to be targeted to overcome PI3K inhibitor resistance in clinic.

### 2.6. Enhanced Insulin Production upon PI3K Inhibition

The p110α subunit of PI3K is known to mediate cellular responses to insulin signaling in different diseases including cancer. Targeting this subunit via specific PI3K inhibitors inhibits the insulin-dependent signaling, which in turn promotes glycogen breakdown in liver and inhibits glucose uptake in skeletal muscle and adipose tissues, resulting in hyperglycemia. However, this effect is compensated via the insulin feedback pathway that activates PI3K/AKT/mTOR signaling in tumor cells compromising the therapeutic response to PI3K inhibitors. It has been shown that use of dietary and pharmaceutical approaches can also target this insulin feedback, resulting in enhanced efficacy and reduced toxicities due to PI3K inhibitors [98]. Molinaro et al. have shown that insulin-mediated PI3K/AKT signaling is dependent on redundant PI3Kα/β activities and is mediated by RAS [99]. In another study in acute leukemia patients, the effects of PI3K/AKT/mTOR inhibitors and insulin varied among the patient subsets, and it was noted that the variations in insulin responsiveness are associated with differential susceptibility to metabolic targeting [100]. An increase in the plasma levels of insulin that is associated with hyperglycemia is also observed in patients treated with PI3K inhibitors and hence enhanced insulin level serves as a pharmacodynamics surrogate marker for such patients [101,102].

### 2.7. Selection of Patient Population in Clinical Trials

Selection of accurate patient population with active *PIK3CA* mutations determines the efficacy of PI3K inhibitors. For example, in a phase I study, the clinical benefit rate (CBR) to a PI3K inhibitor, Alpelisib was 44% in tumors with *PIK3CA* hotspot mutations versus 20% in those patients with WT *PIK3CA* [101]. In another study, patients with *PIK3CA* H1047R mutation had a better survival rate (36%) over subjects with WT tumors [103]. In addition, significant improvement in the progression-free survival (PFS) was seen for patients with *PIK3CA* mutant-specific ctDNA treated with Alpelisib, indicating that the efficacy of PI3Kα inhibitors is dependent on *PIK3CA*-mutant tumors [104]. However, patients with other concurrent alterations in *TP53*, *FGFR1* and *KRAS* did not benefit from Alpelisib, indicating that exclusion of these additional resistance biomarkers could enhance the efficacy of PI3K inhibitors in cancers [105]. There is evidence suggesting that *PIK3CA* hotspot mutations do not always capture all PI3K-dependent tumor genotypes that potentially respond to PI3K inhibitors. Hence, selection of the optimal patient population with activating *PIK3CA* mutations and identification of other genotypes conferring PI3K pathway dependence including *PIK3CA* mutations or amplifications must be included in PI3K inhibitor-based trials [45].

### 2.8. Miscellaneous Resistance Mechanisms

In addition to those mentioned above, there are other factors and mechanisms of resistance and/or compensatory pathways derived from both clinical and in vitro lab studies, which affect PI3K signaling. These include (a) *HRAS* and *KRAS* mutations which reduce susceptibility to PI3K inhibitors while knockdown of these has shown to improve sensitivity to PI3K inhibitors [106,107]. (b) NOTCH-MYC pathway/*MYC* amplification: NOTCH pathway and downstream induction of c-MYC confer resistance to PI3K inhibitors whereas knocking down *MYC* reverses resistance to PI3K/mTOR dual inhibitors [108]. Another report indicated that *MYC* amplification is associated with eIF4E-mediated resistance to dual PI3K inhibitors [109]. (c) Other factors that confer resistance to PI3K inhibitors include ribosomal S6 kinases 3/4 (RSK3/4), p21-activated kinase1 (PAK1), mitogen- and stress-activated kinase 1 (MSK1), lysine demethylase 6B (KDM6B), insulin-like growth factor-binding protein 5 (IGFBP5) and cyclin-dependent kinase 4/6 (CDK4/6) [110,111,112,113,114]. (d) FGFR1amplifications [115]. Overexpression and/or amplification of AXL, serum/glucocorticoid regulated kinase family member 3 (SGK3), PIM1, S-phase kinase-associated protein 2 (SKP2), phosphoinositide-dependent kinase-1-serine/threonine-protein kinase1 (PDK1-SGK1) [116,117,118,119,120]. In addition, there are other factors leading to resistance mechanisms such as a high level of purine-related metabolites, elevated glycolysis with dysregulated mitochondrial signaling leading to loss of *PTEN* and PPP2R2B expression in response to dual PI3K/mTOR inhibitors [121,122,123], macrophages in the tumor microenvironment contributing to PI3K inhibitor resistance via the NFĸB pathway [124], and lack of PI3K mutant-specific inhibitors currently available [45]. These are other pathways which need to be explored and targeted in the development of successful PI3K inhibitors.

## 3. Recent Advances in PI3K Inhibitor-Based Therapies

There are several PI3K inhibitors including pan, isoform-specific and dual PI3K/mTOR inhibitors which are tested in various phases of human clinical trials. In this section, we describe the active not-recruiting and completed studies involving PI3K inhibitors. We have also briefly mentioned about novel pan-PI3K inhibitors yet to be tested in clinical trials including RIDR-PI-103 that is currently investigated in our laboratory. Figure 1 highlights various drugs targeting PI3K signaling.

### 3.1. Pan-PI3K Inhibitors

Pan-PI3K inhibitors target all four isoforms (α, ß, δ, and γ) of class I PI3K and are ATP-competitive inhibitors. These inhibitors are associated with several adverse effects due to unselective blockage of PI3K signaling. Herein, we have listed pan-PI3K inhibitors which are in different stages of human clinical trials. Out of all pan-PI3K inhibitors, only Copanlisib has been approved by the FDA. 

#### 3.1.1. Buparlisib/NVP-BKM120/BKM120

Buparlisib/NVP-BKM120/BKM120 is a potent oral reversible bioavailable pan-PI3K inhibitor targeting p110 -α,-ß, -δ, -γ isoforms with IC50 values of 52, 166, 116 and 262 nM, respectively [125]. Few of the active not-recruiting and completed trials testing Buparlisib in phase I and phase II are mentioned below. Most of the clinical trials using Buparlisib are either sponsored or are in collaboration with Novartis Pharmaceuticals. For example, a phase I active trial is testing Buparlisib with Rituximab, a chimeric monoclonal antibody against CD20 in patients with relapsed or refractory (R/R) indolent B-cell lymphoma. The study was sponsored by the Ohio State University Comprehensive Cancer Center. The main aim of the trial was to determine the maximum tolerated dose (MTD) and dose-limiting toxicities (DLT) of this drug combination [126].

There are several completed studies with Buparlisib without any significant clinical outcomes as summarized below. The trials with specific clinical outcomes and adverse events are summarized in Table 1. An early phase I study aimed to determine the grade of inhibition of PI3K/mTOR signaling in a pre-surgery setting with Buparlisib along with potential biomarker assessment for a pathologic complete response in breast cancer patients. This was sponsored by Grupo Hospital de Madrid [127]. A phase II trial sponsored by University Hospital, Essen investigated the efficacy and safety of Buparlisib with tamoxifen (an ER inhibitor) in ER/PR+, HER2- breast cancer patients with prior exposure to anti-hormonal therapy [128]. A phase I multicenter, non-randomized open-label dose-escalation study aimed to investigate the MTD of Buparlisib in combination with LEE011 (a CDK4/6 inhibitor) and Letrozole (a non-steroidal aromatase inhibitor) in HR+, HER2- post-menopausal women with locally advanced or metastatic breast cancer (MBC) was completed without significant outcomes [129]. A phase I study tested the efficacy of Buparlisib in Chinese patients with advanced solid tumors [130].

Another phase I study of Buparlisib was explored in combination with chemotherapeutics Cisplatin and Etoposide in patients with small-cell lung cancer (SCLC). The study was sponsored by University of California. The primary objective of the trial was to determine the safety and feasibility and the secondary objectives were to establish the MTD, DLT and pharmacokinetic parameters of this triple drug combination regimen [131]. It was also investigated with LEE011 plus Fulvestrant along with another PI3K-α-specific class I inhibitor (BYL719) to explore the clinical utility of this drug combination in ER+/HER2- locally advanced or metastatic breast cancer patients [132]. A phase I dose-escalation treatment regimen included Buparlisib administrated in a concurrent treatment arms with chemotherapy (Capecitabine) or anti-HER2 targeted therapy in patients with MBC. The first treatment arm tested Buparlisib plus Capecitabine, the second arm tested BYL719 plus Capecitabine, the third arm tested BKM120 with Capecitabine plus Trastuzumab (a monoclonal antibody targeting HER2) and the last arm tested the concurrent treatment of Buparlisib with Capecitabine plus Lapatinib (a HER2-specific inhibitor). The study was initiated by the UNC Lineberger Comprehensive Cancer Center in collaboration with Novartis Pharmaceuticals and aimed to achieve the tolerability, safety, and MTD of these regimens [133]. A phase II study sponsored by US Oncology Research investigated the safety and efficacy of Buparlisib in combination with chemotherapy, Capecitabine in breast cancer patients with brain metastases [134]. Another chemotherapy (Docetaxel) was tested with Buparlisib in an early phase I pilot study to identify the adverse effects and the recommended phase 2 dose (RP2D) of Buparlisib in patients with locally advanced metastatic solid tumors who cannot be treated with surgery. The study was sponsored by Roswell Park Cancer Institute in collaboration with NCI and Novartis Pharmaceuticals [135]. A multicenter, open-label phase I study sponsored by the UNC Lineberger Comprehensive Cancer Center aimed to define the MTD of Buparlisib with a chemotherapy regimen folinic acid (leucovorin)-fluorouracil-oxaliplatin (mFOLFOX6) in advanced metastatic pancreatic cancer patients [136]. A phase I trial aimed to evaluate the preliminary anti-tumor effect, the safety profile/tolerability of BKM120 or BEZ235 (another PI3K/mTOR inhibitor) with Letrozole (non-steroidal aromatase inhibitor) used as endocrine therapy in post-menopausal MBC patients. The study was sponsored by the Vanderbilt-Ingram Cancer Center [137]. A phase I study sponsored by Washington University School of Medicine in collaboration with NCI aimed to determine the toxicity profile and the MTD of Buparlisib with Fulvestrant in estrogen receptor (ER)+ stage IV breast cancer patients [138].

A phase I study evaluating the best dose of Buparlisib in combination with chemotherapies (Carboplatin and Pemetrexed disodium) was completed in patients with stage IV lung cancer. The study was sponsored by City of Hope Medical Center with NCI and aimed to figure out the RP2D, DLTs of this combination regimen [139]. Other chemotherapies were also tested in combination with Buparlisib. For example, in a phase Ib study sponsored by Fondazione Michelangelo, Buparlisib was tested with Cisplatin (in group 1) or with Carboplatin (in group 2) to identify the RP2D of these drug combinations for patients with advanced solid tumors [140]. A phase I pilot study aimed to identify the clinical benefits, overall response rate (ORR) and toxicities associated with Buparlisib treatment in patients with non-hodgkin lymphoma (NHL). The trial was sponsored by Mayo Clinic with NCI [141]. Two phase I/II trials tested the efficacy of Buparlisib in combination with either thoracic radiotherapy or with Erlotinib, an EGFR inhibitor in non-small-cell lung cancer (NSCLC) patients [142,143]. An open-label, phase II trial evaluated Buparlisib to determine the PFS for patients with R/R primary central nervous lymphoma (PCNSL) and R/R secondary central nervous system lymphoma (SCNSL) [144].

#### 3.1.2. CH5132799/PA-79

CH5132799/PA-79 is a novel oral pan-class I PI3K inhibitor with a strong inhibitory activity against the PI3Kα isoform (IC50 = 0.014 μM). This drug has promising anti-tumor activity in PI3K mutation-driven cancers and xenograft mice models [145,146]. To date, there is only one phase I study that has tested CH5132799 as described in Table 1.

#### 3.1.3. Pilaralisib/XL147/SAR245408

Pilaralisib/XL147/SAR245408 is a reversible pan-class I PI3K inhibitor against the PI3Kα/δ/γ isoforms with respective IC50 values of 39 nM/36 nM/23 nM in cell-free assays although known to be less potent to PI3Kβ isoform [147,148]. Most trials using Pilaralisib were sponsored by Sanofi. In an early phase I study sponsored by Sanofi in collaboration with Merrimack Pharmaceuticals, Pilaralisib was tested with MM-121/SAR256212, an anti-HER3 antibody with a goal to determine the MTD and RP2D of this drug combination in patients with advanced solid tumors [149]. Another such phase I dose-escalation trial tested the clinical efficacy of Pilaralisib with MSC1936369B, an oral MEK inhibitor in locally advanced or metastatic solid tumors to evaluate the safety, tolerability and initial PK profile of this drug combination [150]. Similar phase I and II studies sponsored by Sanofi, respectively, investigated Pilaralisib as monotherapy in patients with advanced solid tumors or recurrent endometrial cancer [151,152].

#### 3.1.4. ZSTK474

ZSTK474 is an oral ATP-competitive pan-PI3K class I inhibitor with week effectiveness against mTOR [153]. To date, Zenyaku Kogyo Co., Ltd., Tokyo, Japan has sponsored all the known clinical trials using this drug. ZSTK474 was tested as a monotherapy in open-label phase I studies in Japanese patients and other cancer patients with advanced solid tumors to evaluate the safety, tolerability, efficacy and initial PK profile [154,155].

#### 3.1.5. Sonolisib/PX-866

Sonolisib/PX-866 is an improved semi-synthetic analog of Wortmannin, which is an oral, irreversible pan-PI3K inhibitor that had demonstrated significant anti-tumor effect in squamous cell carcinoma of head and neck (SCCHN) patients with PI3K mutation [156]. It exhibits strong inhibitory activity against PI3K110 -α, -δ and -γ isoforms but poor activity against the ß isoform [157]. A phase I/II study tested Sonolisib in combination with Docetaxel in patients with solid tumors [158]. The study was sponsored by Cascadian Therapeutics which also evaluated Sonalisib as monotherapy in patients with advanced solid tumor in yet another early phase I trial [159]. A phase I/II open-label study determined the safety and efficacy of Sonolisib with Cetuximab-based chemotherapy in a combination treatment regimen. In the phase I part of the trial, Sonolisib plus Cetuximab was administrated in patients with castration resistant colorectal (CRC) cancer or progressive, recurrent SCCHN. The phase II part of the study was a randomized evaluation of the anti-tumor activity and safety of Sonolisib plus Cetuximab versus Cetuximab alone in patients with either incurable metastatic CRC who had a prior history of progression or recurrence on chemotherapy-based treatment regimens including Irinotecan and Oxaliplatin, or were resistant to Irinotecan treatment (group 1) or in metastatic SCCHN (group 2) [160].

#### 3.1.6. Pictilisib/GDC-0941/RG7321

Pictilisib/GDC-0941/RG7321 is an oral bioavailable potent PI3Kα/δ inhibitor with an IC50 of 3 nM and has weak selectivity against the p110β (10-fold) and p110γ (25-fold) isoforms. It binds and competes with the ATP-binding pocket to inhibit PI3K signaling pathway [161,162]. It also had a better tolerability and safety profile in Japanese patients with advanced NSCLC [163]. Most of the trials using Pictilisib are sponsored by Genentech. In one such open-label, multicenter, phase Ib dose-escalation study, Pictilisib was administered with one of three chemotherapies regimens: Arm A: Paclitaxel plus Carboplatin in Bevacizumab-ineligible NSCLC patients, Arm B: Paclitaxel, Carboplatin plus Bevacizumab in Bevacizumab-eligible NSCLC patients and Arm C: Pemetrexed, Cisplatin with Bevacizumab in Bevacizumab-eligible NSCLC patients to assess the tolerability, safety and PK profile of pictilisib [164]. Another phase II study tested Pictilisib with the above chemotherapy regimens in NSCLC patients [165]. A phase Ib open-label, three arm and dose-escalation study investigated the safety, tolerability, PK and activity of oral Pictilisib administered with either intravenous (IV) infusion of humanized monoclonal antibody Trastuzumab covalently linked to the cytotoxic agent DM1 (T-DM1) or IV infusion of Trastuzumab in HER+ breast cancer progressed on Transtuzumab therapy [166]. Another phase I study tested the safety and efficacy of Pictilisib in advanced or metastatic tumors [167].

#### 3.1.7. Copanlisib/BAY 80-6946/Aliqopa

Copanlisib/BAY 80-6946/Aliqopa is an intravenous highly potent and reversible pan-class I PI3K inhibitor with significant activity against α and δ isoforms. Copanlisib was brought into market by Bayer and approved by the FDA for treatment of patients with relapsed follicular lymphoma (FL) [168,169]. In a phase I trial, Copanlisib was tested with Paclitaxel in patients with advanced cancer to determine its MTD and RP2D [170].

There are several active trials which are not currently enrolling patients for Copanlisib testing as summarized below. For example, a phase Ib/II active trial is investigating Copanlisib in Japanese patients with relapsed indolent B-cell NHL [171]. In another phase I study (ROCOCO), Copanlisib is currently being explored with Rogaratinib (a FGFR inhibitor) to evaluate the safety, tolerability and the MTD of this drug combination in patients with locally advanced or metastatic solid tumors with a primary objective to find the RP2D. The secondary objective of the trial is to characterize the PK profile and anti-tumor efficacy [172]. A phase I/II study is testing the safety and efficacy of Copanlisib with chemotherapy, Gemcitabine in patients with R/R peripheral T-cell or NK/T-cell lymphoma. The study is sponsored by Chonnam National University Hospital in collaboration with Consortium for improving survival of lymphoma and Bayer to determine the DLTs, MTD and RP2D [173]. In a large-scale randomized phase III study (CHRONOS-2 and 3), Copanlisib is treated alone or in combination with Rituximab in Rituximab-refractory indolent NHL patients [174,175]. In further extension of this phase III study (CHRONOS-4) Copanlisib is explored with standard immune-chemotherapy versus placebo or immunotherapy control to assess the PFS in the above patients [176]. A phase Ib/II study (COPAN-ORL06) sponsored by UNICANCER is assessing the MTD, efficacy and RP2D of Copanlisib in combination with Cetuximab in patients with recurrent and/or metastatic head and neck squamous cell carcinoma (HNSCC) with a *PIK3CA* mutation/amplification and/or a *PTEN* loss [177]. A phase I/II trial is investigating the best dose and side effects of Copanlisib when given with Letrozole and Palbociclib in HR+ and HER2- stage I–IV breast cancer patients [178]. A phase II study is testing the safety and efficacy of Copanlisib plus Gemcitabine and Cisplatin in patients with advanced cholangiocarcinoma. The study is sponsored by the H. Lee Moffitt Cancer Center and Research Institute in collaboration with Bayer [179].

#### 3.1.8. B591

B591, a dihydrobenzofuran-imidazolium salt is a novel pan-PI3K inhibitor with potent inhibitory activity against class I PI3K. This has shown significant inhibition of cellular PI3K/AKT signaling with robust anti-tumor activity in a set of cancer cell lines. However, it has not entered into clinical trials [180].

#### 3.1.9. TG-100-115

TG-100-115 is a pan-PI3K inhibitor, developed by Sanofi, which has significant activity against γ/δ isoforms with IC50 values of 83 nM/235 nM in cell-free assay. However, TG-100–115 has less effect against α/β isoforms [181]. This drug is in phase I and phase II trials in other diseases but not in cancer. However, Song et al. have shown that TG100-115 can be used as a potent TRPM7 kinase inhibitor and a potent suppressor of breast cancer cell migration [182].

#### 3.1.10. RIDR-PI-103

The pyridinylfuranopyrimidine inhibitor, PI-103 is a pan-PI3K inhibitor which targets p110α/β/δ/γ isoforms with IC50 values of 2 nM/3 nM/3 nM/15 nM in cell-free assays and has a strong activity against PI3Kα isoform. However, it failed in clinical trials due to toxicity and poor bioavailability issues [183]. Therefore, a bioisostere of PI-103 (RIDR-PI-103) was synthesized with a structural modification containing a boronate in place of a phenolic hydroxyl group which enhanced the bioavailability of PI-103 [184]. RIDR-PI-103 differs from the boron-containing compound as it releases PI-103 under oxidative stress in tumor microenvironment. This self-cyclizing prodrug has shown significant in vitro anti-cancer efficacy in acute myeloid leukemia [185]. In addition, our laboratory has shown that RIDR-PI-103 in combination with the chemotherapeutic, doxorubicin impaired PI3K signaling, increased DNA damage response and suppressed in vitro breast cancer cell growth [186]. Active not-recruiting and completed trials with specific clinical outcomes for pan-PI3K inhibitors are summarized in Table 1. 

**Table 1 ijms-22-03464-t001:** Summary of trials, outcomes and adverse events associated with pan-PI3K inhibitors in various phases of clinical studies.

Treatment	Status	Sponsor	Phase and NCT	Clinical Outcomes	Adverse Effects
**BKM120/Buparlisib**					
Buparlisib (Bup) in combination with Irinotecan (Iri) in previously treated advanced colorectal cancer patients [187]	Completed	University of Kansas Medical Center	I, NCT01304602	11 patients enrolled and 6 evaluated for toxicities. The combination did not show any significant toxicities in these patients (*n* = 6). The PK of Iri was unaltered by addition of Bup. The MTD was not reached. Cohort 0 (*n* = 3; Iri 120 mg/m^2^ + Bup 50 mg/d): cycle 1 vs. cycle 2 pharmacokinetics mean C_max2_/C_max1_ and mean AUC_2_/AUC_1_ were 0.89 + 0.16 and 1.07 + 0.16, respectively. Cohort 1 (*n* = 3; Iri 150 mg/m^2^ + Bup 120 50 mg/d).	Nausea, vomiting, fatigue, diarrhea, increase in AST/ALT and hyperglycemia.
Buparlisib in combination with Everolimus (Eve) in patients with advanced solid tumors [188]	Completed	Emory University in collaboration with Novartis Pharmaceuticals	I, NCT01470209	Total *n* = 43 patients enrolled. The combination was well tolerated and safe in these patients. The MTD and RP2D for Eve and Bup was 5 and 60 mg, respectively, when on continuous daily schedule. There was no evidence of drug–drug interaction with concurrent administration of Eve and Bup. Paired skin biopsies for baseline and cycle 1 patients demonstrated target engagement with modulation of mTOR/PI3K signaling pathway biomarkers. There was a marked reduction in pS6 and p4EBP1 levels in cycle 1 biopsies compared to baseline. The mean dose-normalized C_max_, T_1/2_ and dose-normalized AUC_(0-infinity)_ were 12.5 ng/mL/mg (95% CI: 5.6–28.3 ng/mL/mg), 118 h (95% CI: 14–1224 h) and 2041.7 ng × h/mL/mg (95% CI: 319.7–27249.9 ng × h/mL/mg), respectively. As per RECIST 1.1 criteria: The median PFS and OS were 2.7 months (95% CI: 1.8–4.2 months) and 9 months (95% CI: 6.4–13.2 months), respectively. 11% and 89% of 27 patients demonstrated PD and SD, respectively.	Diarrhea, nausea, hyperglycemia, hypokalemia, muscular pain, anorexia, fatigue and elevated ALT/AST. 7 patients had additional DLTs such as mucositis, acute kidney injury and urinary tract infection. Gr 4 and 5 adverse effects were rarely observed.
Buparlisib in Japanese patients with advanced solid tumors [189]	Completed	Novartis Pharmaceuticals	I, NCT01283503	Bup had a manageable safety profile with favorable PK. It showed anti-tumor activity in this cohort of patients (*n* = 15) The RP2D was determined as 100 mg/day. As per RECIST criteria, v1.0: 40% and 46.7% of 15 patients demonstrated SD and PD, respectively. The disease control rate was 40%. Bup was rapidly absorbed. A C_max_ of 1–1.5 h was observed with T_1/2_ of ~40 h. The C_max_ and AUC_(0–24)_ increased proportionately with an increase in doses from 25 to 100 mg/day. A Gr 4 DLT-abnormal liver function was observed.	Rash, increased blood insulin levels, increased ALT/AST and increased eosinophil count.
Buparlisib in patients with advanced solid tumors [190]	Completed	Novartis Pharmaceuticals	I, NCT01068483	Treatment was safe and well tolerated with a favorable PK profile (*n* = 35). It demonstrated anti-tumor activity and clear evidence of target inhibition. 1 TNBC patient with *KRAS* mutation demonstrated a confirmed PR. The drug exhibited dose-dependent PD effect on fasting C-peptide, [^18^F] FDG-PET, pS6 and fasting blood glucose. 100 mg/d Bup was established as the MTD. At a dose of 100 mg/d, day 28 (*n* = 17), Bup was rapidly absorbed after oral administration. The T_max_ was found to be 1.25 h. The mean C_max_ and AUC_(0–24)_ were 1850 ng/mL and 22,500 h × ng/mL, respectively. T_1/2_ was ~40 h. 52% of patients had SD.	Mood alteration (depression, anxiety, euphoria), epigastralgia, rash, diarrhea, anorexia, nausea, fatigue, hyperglycemia, pruritus and mucositis.
Buparlisib in combination with MEK162/Binimetinib in patients with advanced solid tumors [191]	Completed	Array Biopharma, now a wholly owned subsidiary of Pfizer	I, NCT01363232	*n* = 89 patients enrolled in the study. The combination showed promising activity in patients with ovarian cancer with *RAS/BRAF* mutation. The MTD for Bup and MEK162 was established at 90 mg/day and 45 mg twice daily dose, respectively. The RP2D for Bup was determined as 80 mg/day and MEK162 45 mg twice daily dose. Other dosing strategies such as pulsatile dosing should be adopted for further trials as continuous dosing led to intolerable toxicities. As per RECIST v1.1 criteria for patients treated with RP2D dose (*n* = 69): Median PFS for patients with *RAS/BRAF*-mutant ovarian cancer was 3.7 months (95% CI: 1.8 months–N.E.). 27.8% of patients with *RAS/BRAF*-mutant ovarian cancer and 7.7% of the *EGFR*-mutant NSCLC group achieved a PR. The disease control rate for patients with *RAS/BRAF*-mutant ovarian cancer was 61.1% (95% CI: 35.7–82.7%). The C_max_ and AUC_τ_ for combination on day 15 were 764.3 ng/mL and 11690.71 h × ng/mL, respectively. The PK of MEK162 was not altered by addition of Bup. Comparison of baseline and post-baseline tumor samples exhibited inhibition of PI3K and MAPK signaling as combination therapy resulted in downregulation of pERK and pS6 signaling.	Central serous retinopathy, diarrhea, stomatitis, pneumonia, vomiting, nausea, maculopapular rash, increase in ALT and elevation in blood creatine phosphokinase.
Buparlisib in combination with Carboplatin and Paclitaxel in patients with advanced solid tumors [192]	Completed	Memorial Sloan Kettering Cancer Center in collaboration with Novartis Pharmaceuticals and Sai Life Sciences	I, NCT01297452	The combination was not well tolerated. There were no significant outcomes (*n* = 14).	-
Buparlisib in combination with Trametinib (Tram)/GSK1120212 in selected advanced solid tumors [193]	Completed	Novartis Pharmaceuticals	Ib, NCT01155453	Combination of Bup and Tram showed promising anti-tumor activity in KRAS-mutant ovarian cancer patients (*n* = 113). The MTD for Bup and Tram was determined as 70 mg plus 1.5 mg daily dose, respectively. According to the BLRM, the RP2D for patients with *BRAF/RAS* solid tumors for Bup and Tram was found to be 60 mg plus 1.5 mg daily dose. Ovarian cancer patients (*n* = 21): The median PFS was 7 months (95% CI: 4.2–12.9 months) and median OS was not met. The ORR was 28.6%. For patients (*n* = 8) treated with 60 mg Bup and 1.5 mg Tram, the ORR was 50%. The overall CR, PR and disease control rates were 4.8%, 23.8% and 76.2%, respectively. 10% of patients experienced target lesion reduction. NSCLC patients (*n* = 17): The median PFS and OS were 4 months (95% CI: 1.8–5.3 months) and 5 months (95% CI: 3.9–N.E.). 6% of the patient with a *KRAS* mutation attained a confirmed PR. The best overall response was 53%. Pancreatic cancer patients (*n* = 24): The median PFS and OS were 2 months (95% CI: 1.8–3.4 months) and 5 months (95% CI: 3.8–5.8 months). Bup exhibited rapid absorption following oral administration at RP2D with a median T_max_ of 2.95 h (95% CI: 1.5–12.3 h) at day 15. At RP2D, the geometric mean values of C_max_ and AUC_0-24_ on day 15 were 522 ng/mL (coefficient of variation 47.04%) and 6607 ng × h/mL (coefficient of variation- 32.3%), respectively. At RP2D the geometric mean values of accumulation ratio on day 15 and 28 were 2.12 (coefficient of variation- 47.04%) and 2.05 (coefficient of variation- 44.88%), respectively. Steady state for both the drugs was achieved within 15 days. A reduction in the expression of p-S6 and p-ERK was observed compared to baseline. Ovarian cancer patients with *KRAS* G12V mutation appeared to respond better to this combination in comparison to patients with other alterations.	Stomatitis, dysphagia, diarrhea, nausea, vomiting, increased creatine kinase, AST/ALT, maculopapular rash and dermatitis aceniform.
Buparlisib in combination with Hedgehog signaling pathway inhibitor, Sonidegib/LDE225 (LDE) in patients with advanced solid tumors [194]	Completed	Novartis Pharmaceuticals	Ib, NCT01576666	The combination of Bup and LDE was well tolerated and consistent with the PK phase I studies (*n* = 46). 63% of patients exhibited disease progression. The MTD was not reached and no drug–drug interaction was observed. The inter-individual variability of Bup and LDE PK was 30% and 67%, respectively.	Increased creatine phosphokinase, increased AST/ALT, aphasia, increased blood alkaline phosphatase, nausea and fatigue.
Buparlisib in combination with Paclitaxel (Pac) for patients with advanced solid tumors and with Trastuzumab (Tras) for HER2+ breast cancer [195]	Completed	Novartis Pharmaceuticals	I, NCT01285466	Bup can be safely administered in combination with Pac and Tras for advanced solid tumors or HER2+ breast cancer patients. Bup (40–120 mg/d) + Pac (70−80 mg/m^2^ IV weekly) (*n* = 53): Median exposure was 15 weeks. The MTD for Bup and Pac was established as 100 mg/d and 80 mg/m^2^ IV weekly, respectively, for 28 days. Bup exposure was lower than in a single-agent study. Bup did not modify the PK of Pac. As per RECIST criteria, the ORR was 17%. The major reason for discontinuation was due to a PD of 68%. Bup (100 mg/d) + Pac + Tras (2 mg/kg) (*n* = 11): Median exposure was 17 weeks. The MTD was Bup (100 mg/d), Pac (80 mg/m^2^ IV weekly) and Tras (2 mg/kg weekly) for 28 days. As per RECIST criteria, the ORR was 27%. The major reason for discontinuation was due to a PD of 64%.	Bup + Pac: hyperglycemia and neutropenia. Bup+ Pac +Tras: neutropenia and diarrhea.
Buparlisib in combination with mAb targeting VEGF-A; Bevacizumab (Bev) in metastatic renal cell carcinoma patients [196]	Completed	Dana-Farber Cancer Institute in collaboration with Beth Israel Deaconess Medical Center	I, NCT01283048	Bup and Bev were tolerable and showed preliminary activity in VEGF-refractory metastatic renal cell carcinoma (*n* = 32). The MTD for Bup was 80 mg/d. The PR was 11% (95% CI: 1–33%) for those treated at MTD. The OS, PFS and median TTF were 18% (95% CI: 4–N.E.); 9% (95% CI: 2–9%) and 4% (95% CI: 2–9%). 2 patients had an activating *PIK3CA* mutation, 1 of which achieved a PR (TTF-13 months) and the other exhibited a 16% tumor shrinkage (TTF-9 months).	Rash, pruritis, transaminitis, cognitive disturbance, anxiety, depression and suicidal ideation and elevated lipase and amylase.
Buparlisib in patients with advanced leukemia [197]	Completed	M.D. Anderson Cancer Center in collaboration with Novartis Pharmaceuticals	I, NCT01396499	Bup administered at 80 mg/d (MTD) was efficacious and well tolerated in patients with R/R acute leukemia. 14 patients were enrolled in the study (1 patients had acute lymphoblastic leukemia, 12 patients had acute myeloid leukemia and 1 patient had mixed phenotype leukemia). 1 patient exhibited SD and had the longest duration on study for 82 days when treated at MTD. The median OS was 75 days (range: 10–568 days). 3 patients had 3q26 chromosomal abnormality and these patients exhibited longest OS with median survival of 360 days (range: 278–568 days). The other 11 patients who lacked the 3q26 chromosomal abnormality had a median survival of 57 days (range: 10–125 days). A decrease in p-pS6K (in 5/7 patients) and p-FOXO3/total FOXO3 (in 4/6 patients) levels was observed post-Bup treatment. The mean quantitative inhibition of p-pS6K and p-FOXO3/total FOXO3 was 65% (range: 32–100%) and 93% (range: 89–100%), respectively. As per RPPA analysis, PRAS 40, MDM2, RPS32 and BRD4 were proteins that were significantly downregulated pre- and post-Bup treatment.	Confusion, fatigue, mucositis nausea, elevated serum bilirubin and dysphagia.
Buparlisib in combination with Temozolamide (Tem) and Radiation Therapy in newly diagnosed glioblastoma patients [198]	Completed	Novartis Pharmaceuticals	I, NCT01473901	Due to challenging safety profile and inability to achieve the MTD, the sponsor decided not to pursue the use of Bup in newly diagnosed glioblastoma patients.	-
Buparlisib in combination with Bevacizumab (Bev) in R/R glioblastoma multiforme patients [199]	Completed	SCRI Development Innovations, LLC in collaboration with Novartis Pharmaceuticals	I/II, NCT01349660	Combination of Bup and Bev was well tolerated (*n* = 68) and demonstrated clinical efficacy even in patients who received prior Bev treatment (*n* = 13). The median duration of treatment was 16 weeks (range: 19–77 weeks). The median OS and PFS were 10.8 months (95% CI: 9.1–22 months) and 5.3 months (95% CI: 3.8–7.5 months), respectively. 51% of these patients demonstrated disease progression. The most common reasons for treatment discontinuation were toxicity and disease progression. For patients who received prior Bev treatment, the CBR was 46%.	Hyperglycemia, fatigue, CNS symptoms, increased ALT, confusion, psychosis and mood alteration.
Buparlisib in combination with mAb targeting EGFR, Panitumumab (Pani) in patients with metastatic/advanced *RAS*-WT colorectal cancer [200]	Completed	Canadian Cancer Trials Group in collaboration with Pfizer	Ib, NCT01591421	The combination of Bup (given 5 days a week) and Pani (6 mg/kg by IV route biweekly) was well tolerated (*n* = 19). Both drugs were administrated in 3 DLs. The median PFS was 2 months. 9 patients had PD and 7 patients exhibited SD (the median duration was 5.4 months [range: 3.7–8.4 months]). The phase II study was stopped, as the predefined futility rule for response was not achieved. DL 1: (60 mg Bup/day and 6 mg/kg Pani biweekly; *n* = 6): Due to toxicity issues, DL1 was not tolerable and doses were de-escalated to DL-1. 1 patient had a PR. DL-1 (40 mg Bup and 6 mg/kg Pani every 2 weeks; *n* = 3): The combination was well tolerated but the dose of Bup was subtherapeutic. DL-1b (60 mg Bup given 5 days a week and 6 mg/kg Pani biweekly; *n* = 10): A total of 22 cycles were administered to 10 patients: 1 patient received 5 cycles and the median duration was 2 cycles. Planned dose intensity for Bup and Pani were 40% and 60%, respectively. No reduction in DLs was required.	DL 1: mucositis, fatigue, palmar-plantar erthrodyesthesia, rash, acneifrom, hypomagnesemia and increased AST/ALT. DL-1: rash, hypomagnesemia and increased AST. DL-1b: rash, mucositis, pruritus, dry skin, increase AST/ALT, hyperglycemia, nausea, anorexia and rash acneiform.
Buparlisib in combination with Abiraterone Acetate in CRPC patients [201]	Completed	Novartis Pharmaceuticals	I, NCT01634061	The MTD was not reached for these patients (*n* = 25). The combination was not efficacious and showed no clinically meaningful benefit. No further studies are planned.	-
Buparlisib in patients with advanced solid tumors [202]	Completed	Novartis Pharmaceuticals	I, NCT01068483	Bup exhibited the ability to inhibit pAKT along with potential to downregulate pS6 levels further downstream at 100 mg daily in these patients (*n* = 83).	-
Buparlisib in combination with tyrosine kinase inhibitor, Imatinib in patients with gastrointestinal stromal tumor for whom treatment failed prior to Imatinib and Sunitinib therapy [203]	Completed	Novartis Pharmaceuticals	Ib, NCT01468688	The combination failed to provide additional benefits compared to current therapies that are available for these patients (*n* = 60). Further development of this combination was terminated due to lack of objective response.	-
Buparlisib in combination with a *BRAF* inhibitor, Vemurafenib (Vem) in advanced *BRAF*V600E/K-mutant melanoma patients [204]	Completed	University of California, San Francisco in collaboration with Novartis Pharmaceuticals	I/II, NCT01512251	The combination of Bup and Vem was not well tolerated and no further studies were proposed. Patients (*n* = 8; 3 patients Vem naïve and 5 patients with Vem refractory melanoma) were treated with 60 mg Bup daily and 720 mg Vem twice a day for 28 days. Dose escalation was not performed due to toxicity and lower doses.	Vomiting, myalgia, arthralgia, vomiting. Febrile neutropenia, rash, elevated AST/ALT, adrenal insufficiency and hypotension.
Buparlisib in combination Carboplatin or Lomustine in patients with recurrent glioblastoma multiforme [205]	Completed	Novartis Pharmaceuticals	Ib/II, NCT01934361	The combination did not demonstrate sufficient anti-tumor efficacy compared to single-agent Lom or Carbo. The study did not proceed to phase II trial.	-
Buparlisib in R/R CLL patients [206]	Completed	Canadian Cancer Trials Group in collaboration with Novartis Pharmaceuticals	II, NCT02340780	Bup demonstrated significant toxicities and further testing of Bup in these patients was ceased (*n* = 14). The data also indicated that basal raptor expression in CLL patients correlated with clinical response to Bup.	-
Buparlisib in R/R NHL patients [207]	Completed	Novartis Pharmaceuticals	II, NCT01693614	Bup demonstrated clinical activity in patients with R/R NHL (*n* = 72) along with sustained tumor burden reduction in some patients and disease stabilization. The median follow up time was 4.6 months. DLBCL (*n* = 26): Median number of prior anti-neoplastic regimes was 3 (range: 1–12). The median duration of exposure to Bup was 7.5 weeks (range: 1.7–76 months). The median time to respond and duration of response was 2.1 months (range: 1.8–3.5 months) and 2.2 months (95% CI: 1.2–N.E.), respectively. The median PFS and estimated 6 month PFS rate was 1.8 months (95% CI: 1.5–4%) and 12.6% (95% CI: 2.3–32.0%), respectively. The median OS was 5.2 months (95% CI: 3.1–N.E.). The CR and PR were 15.4% and 26.9%, respectively. The ORR and disease control rates at 6 months were 11.5% (95% CI: 2.5–30.2%) and 30.8% (95% CI: 14.3–51.8%), respectively. MCL (*n* = 22): Median number of prior anti-neoplastic regimens was 2 (range: 1–6). The median duration of exposure to Bup was 20.6 weeks (range: 1.7–54.1 months). The median TTR and DoR was 1.8 months (range: 0.9–9.4 months) and not achieved, respectively. The median PFS and estimated 6 month PFS rate was 11.3 months (95% CI: 3.8–N.E.) and 68.6% (95% CI: 39.8–85.7%), respectively. The median OS was 8.2 months. The CR and PR were 31.8% and 27.3%, respectively. The ORR and disease control rates at 6 months were 22.7% (95% CI: 7.8–45.4%) and 81.8% (95% CI: 59.7–94.8%), respectively. FL (*n* = 24): Median number of prior anti-neoplastic regimes was 2 (range: 1–9). The median duration of exposure to Bup was 16.3 weeks (range: 3.3–81.3 months). The median time to respond and duration of response was 3.5 months (range: 1.6–5.9 months) and 11 months (95% CI: 3.9–N.E.), respectively. The median PFS and estimated 6 month PFS rate was 9.8 months (95% CI: 3.8–N.E.) and 60.7% (95% CI: 31.7–80.6%), respectively. The median OS was 12.1 months. The CR and PR were 41.7% and 12.5%, respectively. The ORR and disease control rates at 6 months were 25% (95% CI: 9.8–46.7%) and 87.5% (95% CI: 67.6–97.3%), respectively.	Hyperglycemia, nausea, fatigue, neutropenia, diarrhea, hematologic abnormalities, increased AST/ALT, constipation, depression and anxiety.
Buparlisib in patients with metastatic transitional cell carcinoma of the urothelium [208]	Completed	Memorial Sloan Kettering Cancer Center in collaboration with Novartis Pharmaceuticals	II, NCT01551030	Bup did not significantly improve the two-month PFS rate as compared to standard chemotherapy. The results further encouraged an expansion cohort for patients with *PIK3CA* mutations.	-
Buparlisib in patients with recurrent glioblastoma with PI3K pathway activation [209]	Completed	Brigham and Women’s Hospital, Massachusetts General Hospital, M.D. Anderson Cancer Center, Memorial Sloan Kettering, Cancer Center University of California, San Francisco, University of California, Los Angeles, University of Utah in collaboration with Novartis Pharmaceuticals	II, NCT01339052	Bup was well tolerated in patients with PI3K-activated recurrent glioblastoma but exhibited minimal efficacy as single agent. In spite of significant brain penetration, it lacked clinical efficacy due to the inability to block the PI3K pathway when evaluated in tumor tissues.	-
Buparlisib in patients with PI3K-activated tumors (SIGNATURE) [210]	Completed	Novartis Pharmaceuticals	II, NCT01833169	A total of 146 patients were enrolled (colorectal cancer *n* = 18; sarcoma *n* = 14; ovarian cancer *n* = 12; cervical cancer *n* = 11; HNSCC *n* = 11 and anal cancer; *n* = 10). Bup was well tolerated. However, it had limited efficacy in these patient populations as a single agent.	-
Buparlisib as second-line therapy for patients with advanced endometrial cancer [211]	Completed	Novartis Pharmaceuticals	II, NCT01289041	Patients with activated PI3K (*n* = 49) and non-activated PI3K (*n* = 21) status were enrolled. Activated PI3K patient population: According to RECIST v1.0 criteria, the ORR was 2%. The median PFS and OS were 1.9 months (95% CI: 1.8–3.2 months) and 8.9 months (95% CI: 6.3–16.2 months), respectively. Non-activated PI3K patient population: According to RECIST v1.0 criteria, the ORR was 4%. The median PFS and OS were 1.9 months (95% CI: 1.6–3.3 months) and 14.2 months (95% CI: 8.6–24 months), respectively.	Nausea, vomiting, hyperglycemia, anemia, fatigue, dehydration, increased AST/ALT, decreased appetite, hydronephrosis, anxiety and rash.
Buparlisib in PI3K pathway activated NSCLC patients (BASALT-1) [212]	Completed	Novartis Pharmaceuticals	II, NCT01297491	Bup failed to show efficacy in PI3K pathway activated NSCLC patients (squamous *n* = 30; non-squamous *n* = 33).	-
Buparlisib in TNBC patients [213]	Completed	SOLTI Breast Cancer Research Group in collaboration with Novartis Pharmaceuticals, Dana-Farber Cancer Institute and Stand Up To Cancer (NCT01629615); Dana-Farber Cancer Institute (NCT01790932)	II, NCT01790932 and NCT01629615	No confirmed objective responses were observed and Bup was not associated with a strong clinical efficacy in TNBC patients as a single agent (*n* = 50).	-
Buparlisib for initial or recurrent metastatic endometrial cancer post-first-line therapy for patients not able to undergo radiotherapy and/or local surgery (ENDOPIK) [214]	Completed	ARCAGY/GINECO GROUP	II, NCT01397877	Bup had an unfavorable safety profile and limited anti-tumor activity in these patients (*n* = 40). The trial was terminated before the end of recruitment for evaluating toxicity.	-
Neoadjuvant Buparlisib in combination with Trastuzumab and weekly Paclitaxel in HER2+ primary breast cancer patients (NeoPHOEBE) [215]	Completed	Novartis Pharmaceuticals in collaboration with Breast International Group, German Breast Group and SOLTI Breast cancer Research Group	II, NCT01816594	Addition of Bup to Tras + Pac regimen was not feasible and it failed to significantly improve the pathological CR in these patients (*n* = 50).	-
Buparlisib in patients with R/R PCNSL and SCNSL [216]	Completed	Memorial Sloan Kettering Cancer Center in collaboration with Novartis Pharmaceuticals	II, NCT02301364	Even though patients tolerated Bup with acceptable toxicities, there was a lack of clinical response due to a Bup CNS concentration below IC50, which indicated that Bup lacked single-agent activity in these patients (*n* = 4).	-
Buparlisib in recurrent/progressive HNSCC (PIK-ORL) [217]	Completed	Centre Leon Berard in collaboration with National Cancer Institute-France and Foundation ARC	II, NCT01737450	36 patients without *PIK3CA* mutation were enrolled. The median duration of treatment was 8 weeks (range: 4 days 55.9 weeks). As per RECIST 1.1 criteria, the 2 months disease control rate was 38.9%. No overall response was observed.	Asthenia, hyperglycemia, lymphopenia, depression, anxiety, nausea, diarrhea, anemia, increase in leucocyte count and decrease in hematocrit.
Neoadjuvant Buparlisib in men with high-risk localized prostate cancer [218]	Completed	University of California, San Francisco in collaboration with Novartis Pharmaceuticals	II, NCT01695473	Bup demonstrated pharmacodynamic efficacy in these patients (*n* = 11). The study was terminated due to lack of accrual. There was a significant inhibition of the PI3K pathway along with no significant changes in apoptosis or proliferation due to short duration of treatment with Bup prior to analysis.	-
Buparlisib in combination with Paclitaxel (Pac) in HER2- locally advanced or metastatic breast cancer patients with or without PI3K activation (BELLE-4) [219]	Completed	Novartis Pharmaceuticals	II/III, NCT01572727	Combination of Bup and Pac failed to improve PFS in these patients (Bup; *n* = 207 and placebo; *n* = 209). The trial was suspended after phase II.	-
Buparlisib in combination with Fulvestrant (Ful) in postmenopausal HR+, HER2-, endocrine-resistant locally advanced or MBC patients refractory to aromatase inhibitors (BELLE-2) [220]	Completed	Novartis Pharmaceuticals	III, NCT01610284	Combination of Bup and Ful was effective and provided clinically meaningful benefits to these patients (Bup; *n* = 576 and placebo + Ful; *n* = 571). No further studies were conducted due to toxicity associated with the combination. The median PFS in Bup group was 6.9 months (95% CI: 6.8–7.8 months) compared to 5 months (95% CI: 4–5.2 months) in the placebo group (hazard ratio: 0.78 (95% CI: 0.67–0.89).	Increased AST/ALT, rash, hyperglycemia, diarrhea, fatigue, nausea and vomiting.
Buparlisib in combination with Cisplatin and IMRT in high risk LASCCHN patients [221]	Active, not recruiting	Dana-Farber Cancer Institute in collaboration with Novartis Pharmaceuticals	I, NCT02113878	The combination was well tolerated and showed promising activity in these patients (*n* = 23). The median follow up was 12 months (range: 3–24 months); 2 patients (9%) had recurrence and 1 of these patients died.	Rash, mucositis, neutropenia, anorexia, anemia, confusion and dysphagia.
Buparlisib in combination with a BTK inhibitor, Ibrutinib (Ibru) in patients with R/R DLBCL, MCL and FL [222]	Active, not recruiting	Memorial Sloan Kettering Cancer Center in collaboration with Janssen Scientific Affairs, LLC and Novartis Pharmaceuticals	I, NCT02756247	The combination was well tolerated and had a predictable safety profile (*n* = 13; DLBCL-5, FL-2 and MCL-6). The R2PD for Bup and Ibru was 100 and 560 mg, respectively. For long-term therapy and tolerability, dose reduction might be required. Bup (80 mg) and Ibru (420 mg) (*n* = 6): 3 patients had a CR and 1 patient had SD. Bup (80 mg) and Ibru (560 mg) (*n* = 4): 1 patient had a CR and 2 patients had SD. Bup (100 mg) and Ibru (560 mg) (*n* = 3): 2 patients had PR.	Anemia, rash, fatigue, anorexia, diarrhea, leukocytosis, leukopenia, gastric reflux, hyperbilirubinemia, mood change and hypertension.
Buparlisib in combination with LGX818/Encorafenib (BRAF inhibitor) and MEK162/Binimetinib (MEK inhibitor) in patients with advanced *BRAF*V600-mutant melanoma (LOGIC-2) [223]	Active, not recruiting	Pfizer	II, NCT02159066	The triple therapy was feasible depending on genetic alterations but low clinical activity was observed (*n* = 6). Further exploration is needed to identify the patterns of resistance susceptible to use this combination in these patients.	-
Buparlisib in combination with mAb targeting EGFR, Cetuximab in patients with recurrent or metastatic head and neck cancer [224]	Active, not recruiting	University of Chicago in collaboration with NCI	I/II, NCT01816984	The combination was well tolerated and demonstrated limited evidence of activity in pretreated patients (*n* = 12) [HPV+; *n* = 5 and HPV-; *n* = 4]. The median OS was 280 days (HPV+; OS = 370 days; HPV-; OS = 191 days). 1 and 4 patients achieved a PD and SD, respectively. Further plans are proposed to test this combination in a larger cohort.	Fatigue, anorexia and maculopapular rash.
**CH5132799/PA-79**					
CH5132799 (CH5) in patients with advanced solid tumors [225]	Completed	Chugai Pharma Europe Ltd.	I, NCT01222546	CH5 was well tolerated with evidence of clinical activity and the MTD was 48 mg twice daily in patients with advanced solid tumors (*n* = 38). Following a single dose of CH5132799, T_max_ and T_1/2_ were 2.60 and 10.2 h, respectively. At MTD, the mean C_max_ and AUC_(last)_ were 172 ng/mL and 1270 ng × h/mL. The steady state was reached at day 8 and the mean C_max_ and AUC_0–24h_ were 175 ng/mL and 1550 ng/h, respectively. Significant inhibition of p-AKT was observed above dose 32 mg. Decrease in FDG avidity between baseline and at day 8 was observed in 74% of patients. As per RECIST criteria, no CR or PR was achieved. Disease stabilization was observed in 8 patients up to 16 weeks, 2 of which had *PIK3CA* mutation.	Increased AST/ALT, fatigue, diarrhea, rash, stomatitis, hyperglycemia, posterior reversible encephalopathy syndrome and nausea.
**XL147/SAR245408/Pilaralisib**					
XL147/Pilaralisib (Pila) in combination with Erlotinib in patients with solid tumors [226]	Completed	Sanofi	I, NCT00692640	The combination had limited anti-tumor activity and efficacy in these patients with solid tumors (*n* = 35). The combination is no longer under investigation in these patients. Safety findings of this combination study were similar to those of single-agent Pila.	-
XL147 in patients with lymphoma or solid tumors [227,228]	Compl eted	Sanofi	I, NCT00 486135	XL147 was well tolerated with a favorable safety profile and was associated with PI3K pathway inhibition. The RP2D of Pilaralisib was established as 600 mg QD for patients with refractory advanced solid malignancies (*n* = 28). The median time to achieve maximum concentration was 8–22 h and the terminal elimination T_1/2_ was 70–88 h. There was a 5–13-fold drug accumulation after daily dosing. 9 paired tumor biopsies from patient treated with 600 mg XL147 demonstrated robust but partial PI3K pathway inhibition along with reduction in mTOR1 biomarkers such as pEBP1T70 (39–73%), pS6S240/S244 (68–70%) and biomarkers for mTOR2 such as pAKTS473 (55–61%) and pPRAS40T246 (50–68%). A reduction in pMEK and pERK expression was also observed [227].	Fatigue, decreased appetite, hypersensitivity, maculopapular rash, diarrhea and hyperglycemia [227]. Diarrhea, hyperglycemia, headache, pneumonia, lymphopenia and thrombocytopenia [228].
				XL147 was well tolerated in heavily pretreated R/R lymphoma (*n* = 10) and CLL (*n* = 5) patients. Each patient received a median of 4 treatment cycles of XL147 [228].	
XL147 in combination with Paclitaxel (Pac) and Carboplatin (Carbo) in patients with solid tumors [229]	Completed	Sanofi	I, NCT00756847	In spite of XL147 having a favorable safety profile, it failed to demonstrate significate anti-tumor activity in combination with Pac and Carbo in these patients (*n* = 58).	-
XL147 in combination with Trastuzumab (Tras) with or without Paclitaxel (Pac) in patients with HER2+ MBC progressed in previous Trastuzumab regimen [230]	Completed	Sanofi	I/II, NCT01042925	The combination had an acceptable safety profile in these patients. The MTD for XL147 was 400 mg once daily. XL147 demonstrated similar PK when compared with previous studies for XL147 as monotherapy. No correlation between *PIK3CA* mutation and response was established. Arm 1 (XL147 + Tras; *n* = 20): The median PFS was 11 weeks. 5.3% of patients had PFS for more than 24 weeks. No patient demonstrated a clinical response. Arm 2 (XL147+ Tras+ Pac; *n* = 21): The median PFS was 21.1 weeks. 40% of these patients had PFS more than 24 weeks. 20% and 55% of patients had a PR and SD, respectively. 23% of patients had PFS lasting for more than 24 weeks.	Rash, neutropenia, diarrhea, fatigue, headache, anemia, hyperglycemia, vomiting, nausea, peripheral neuropathy and decreased appetite.
XL147 in combination with Letrozole (Let) in patients with HR+, HER2- breast cancer refractory to a non-steroidal aromatase inhibitor [231]	Completed	Sanofi	I/II, NCT01082068	The combination had an acceptable safety profile but limited efficacy in these patients (*n* = 37). The MTD was 400 mg QD XL147 and 2.5 mg Let. XL147 did not interact pharmacokinetically with Let. XL147 had an impact on glucose homeostasis. There was no association between PI3K mutation and efficacy of XL147. The median duration of treatment was 14.7 weeks (range: 0.3–70 weeks). The ORR was 4% (90% CI: 0.2–18.8%) and 4% of patient had a PR. 41.7% and 45.8% of patients had a SD and PD, respectively. The PFS rate at 6 months and median PFS was 17% (90% CI: 6–34%) and 8 weeks (90% CI: 7.7–16.1 weeks), respectively. 33% of patients demonstrated PFS at 24 weeks.	Increased AST/ALT, rash, diarrhea, nausea, vomiting and fatigue.
XL147 in patients with solid tumors or lymphoma [232]	Completed	Sanofi	I, NCT01943838	18 patients were enrolled (*n* = 14 patients with solid tumors and *n* = 4 patients with lymphoma). SD was seen in 5 patients. The duration of treatment ranged from 3 to 8 weeks. At steady state, the median T_max_ was 6.0 h, with a mean C_max_ of 135 μg/mL and AUC_τ_ of 2690 μg × h/mL. Exposure at steady state was similar between 400 (DL1) and 600 mg (DL2), with an accumulation ratio of 4-fold. On DL2, 3 DLTs were seen in 2 patients: 2 patients had events of Gr 3 skin rash and 1 asymptomatic Gr 4 lipase increase. The RP2D was identified at 600 mg daily dose.	Constipation, hypertension, nausea, vomiting, decrease in appetite, dry skin and fatigue.
**Sonolisib/PX-866**					
PX-866 in recurrent/metastatic CRPC patients [233]	Completed	NCIC Clinical Trials Group in collaboration with Oncothyreon Canada Inc.	II, NCT01331083	PX-866 had modest toxicity, with no significant efficacy as a single agent in these patients (*n* = 68). The trial failed to meet prior benchmarks for further development.	-
PX-866 in patients with glioblastoma multiforme at time of first relapse or progression [234]	Completed	NCIC Clinical Trials Group in collaboration with Cascadian Therapeutics Inc.	II, NCT01259869	Even though PX-866 was well tolerated, the ORR was low. The study failed to meet its primary end point and establish a statistically significant association between clinical outcomes and relevant biomarkers for these patients (*n* = 33). 21% of patients had durable SD.	-
**Pictilisib/GDC-0941/RG7321**					
Pictilisib (Pic) in combination with Paclitaxel (Pac) ± Bevacizumab (Bev)/Trastuzumab (Tras) + Letrozole (Let) in patients with recurrent or MBC [235]	Completed	Genentech, Inc.	Ib, NCT00960960	The combination had a manageable safety profile along with anti-tumor activity in these patients (*n* = 69). The study was divided into 3 parts. Part 1: Pic (60–330 mg oral dose on days 1–21 of each 28 day cycle) + Pac (90 mg/m^2^ IV on days 1, 8, and 15 of each 28 day cycle) ± Bev (10 mg/kg IV on days 1 and 15 of each 28 day cycle) (*n* = 20); Part 2A: Pic (oral daily dose for 5 of 7 consecutive days) + Pac (90 mg/m^2^ IV) in *n* = 18 patients; Part 2B: Pic + Pac + Bev (*n* = 15); Part 2C: Pic + Pac + Tras (2–4 mg/kg) (*n* = 9); Part 3: Pic (260 mg) + Let (2.5 mg) (*n* = 7). The MTD and RP2D of Pic was defined as 100 mg when given with Pac + Bev on a “21 + 7” dosing schedule; 250 mg ± Pac + Bev on a “5 + 2” dosing schedule; 260 mg + Pac + Trans administrated on a “5 + 2” dosing schedule)/Let (a continuous dosing schedule). Part 1: The mean DoR and PFS was 8.9 months (95% CI: 6.47–11.1 months) and 5.8 months (95% CI: 3.52–10.87, respectively). Part 2A: The PFS was 5.0 months (95% CI: 3.71—N.E.). Part 2B: The mean DoR and PFS was 8.8 months (95% CI: 4.40–15.34 months) and 7.5 months (95% CI: 4.60–10.41), respectively. Part 2C and 3: The median PFS was 14.8 months (95% CI: 3.52–16.62) and 5.4 months (95% CI: 1.87- N.E.), respectively. 3.4% and 29.3% of all these patients demonstrated a CR and PR, respectively. There was no drug-drug interaction between Pic and Pac as demonstrated by PK analysis.	Nausea, diarrhea, fatigue, rash, alopecia, neutropenia, vomiting, stomatitis, cough, peripheral neuropathy and decreased appetite.
Pictilisib in combination with Erlotinib (Erl) in advanced solid tumors [236]	Completed	Genentech, Inc.	I, NCT00975182	The combination could be safely administered to these patients (*n* = 57), with RP2D being 340 mg Pic (5 days on and 2 days off schedule) and 100 mg Erl. However, this combination demonstrated limited anti-tumor activity.	-
Pictilisib in patients with advanced/metastatic solid tumors for which no therapy exists or is ineffective or intolerable [237]	Completed	Genentech, Inc.	I, NCT00876122	Pic was safe, well tolerated, had a dose-proportional PK, PD activities (at highest level of drug exposure: AUC >20 h × umol/L) and showed anti-tumor activity in these patients; *n* = 60). The RP2D of Pic was established as 330 mg once daily dose. Pic was rapidly absorbed on oral administration and the T_max_ was 2 h (range: 0.5–8 h). Terminal plasma elimination T_1/2_ was between 13.1–24.1 h. The drug accumulation index (AUC_Day15_/AUC_Day1_) was 1.2–2.2. There was modest drug accumulation following multiple dosing of Pic. There was a concentration- and dose-dependent decrease in pAKT levels on days 1 and 15 evaluated in platelet-rich plasma along with reduction in p-S6 and p-AKT levels in tumor biopsies. When 330 mg Pic was administered, up to 90% inhibition of p-AKT was observed 1–3 h post-dosing and the effect was sustained for 24 h. As per RECIST criteria, 1 patient with *BRAF*V600E-mutant metastatic melanoma achieved a PR. As per GCIG-CA125 criteria, 1 patient heavily pretreated, platinum-refractory advanced epithelial ovarian cancer with *PIK3CA* amplification and *PTEN* loss demonstrated a radiologic SD for 4 months. Target modulation was confirmed in 7/32 patients as indicated by significant increase in plasma and glucose levels along with more than 25% decrease in ^18^F-FDG uptake by PET.	Maculopapular rash, nausea, fatigue, vomiting, dysgeusia, decreased appetite and hyperglycemia.
Pictilisib in combination with Paclitaxel in locally recurrent or MBC patients (PEGGY) [238]	Completed	Genentech, Inc.	II, NCT01740336	The study failed to meet its primary end point and there was no significant clinical benefit with this combination for MBC patients (*n* = 183).	-
Pictilisib in combination with Fulvestrant in ER+ advanced/MBC patients resistant to aromatase inhibitor therapy (FERGI) [239]	Completed	Genentech, Inc.	II, NCT01437566	The combination did not significantly improve the PFS. The use of Pic in these patients was limited due to toxicity issues (*n* = 168). No further studies were proposed to test Pic in this setting.	-
Pictilisib in combination with Anastrozole (Ana) in ER+ breast cancer patients (OPPORTUNE) [240]	Completed	-	II, ISRCTN26131497	The combination significantly augmented tumor cell proliferation, as marked by increased Ki67 expression in luminal B primary breast cancer subtype. Significant interaction between molecular subtype and response to treatment was observed (*p* = 0.03), indicating that Ki67 suppression was greater in the combination group vs. group treated with Ana alone in these patients, irrespective of baseline Ki67 expression or PgR status. No significant Ki67 response for Pic was observed for patients with luminal A subtype tumors (*p* = 0.98). No significant correlation between *PIK3CA* mutation and activity of Pic was established. Pic (260 mg once a day) + Ana (1 mg/day) (*n* = 26): Mean % suppression and end of treatment Ki67 expression was 83.8% and 2.9%, respectively. The anti-proliferative response rate for PAM50 luminal B tumors was 86.5%. Patients with PR- luminal B status demonstrated greatest anti-proliferative effect from combination treatment. The combination was also more effective in patients with PgR- luminal A subtype breast cancer. Ana (*n* = 49): Mean % suppression and end of treatment Ki67 expression was 66% and 6.1%, respectively. The anti-proliferative response rate for PAM50 luminal B tumors was 63.6%.	Rash, diarrhea, fatigue, hyperglycemia, nausea, stomatitis, dyspepsia and headache.
**Copanlisib/BAY 80-6946/Aliqopa**					
Copanlisib (Cop) in advanced solid tumors [241]	Completed	Bayer	I, NCT00962611	A total of 57 patients were enrolled. The MTD of Cop was identified as 0.8 mg/kg. Cop exposure was dose proportional, with no accumulation and the peak exposure positively correlated with transient hyperglycemia post-infusion. 16 of 20 patients treated at the MTD had reduced [^18^F] FDG-PET uptake; 7 patients (33%) experienced a reduction >25%. 1 patient achieved a CR who had endometrial carcinoma with both *PIK3CA* and *PTEN* mutations and *PTEN* loss and 2 patients had a PR; both had metastatic breast cancer. Among the 9 NHL patients, 6 patients with FL responded (1 CR and 5 PRs) and 1 patient with DLBCL had a PR by investigator assessment; 2 patients with FL who achieved CR (per post hoc independent radiologic review) were on treatment >3 years.	Nausea and Hyperglycemia.
Copanlisib in Japanese patients [242]	Completed	Bayer	I, NCT01404390	10 patients were recruited; 3 patients received Cop at 0.4 mg/kg dose and 7 received Cop at 0.8 mg/kg. No patients treated at 0.4 mg/kg had a DLT and the MTD of Cop was found to be 0.8 mg/kg. Cop PK exposures displayed near dose proportionality without any accumulation. There were no subjects who achieved a CR or PR, and disease control rate was found to be 40.0%.	Hyperglycemia, constipation and hypertension.
Copanlisib plus the MEK inhibitor, Refametinib (Ref) in advanced cancer patients [243]	Completed	Bayer	I, NCT01392521	In this dose-escalation (*n* = 49) and expansion (*n* = 15) study, the MTD was defined at Cop 0.4 mg/kg weekly and Ref 30 mg twice daily dose. There was no drug–drug interaction observed. MEK-ERK inhibition and decreased tumor FDG uptake were reported during treatment. Best response was SD in *n* = 21 patients.	Fatigue, diarrhea, nausea and acneiform rash. DLTs included oral mucositis increased ALT/AST, rash acneiform, hypertension and diarrhea.
Copanlisib in patients with solid tumors and NHL patients [244]	Completed	Bayer	I, NCT02155582	63 patients received Cop at 0.4 or 0.8 mg/kg on days 1, 8, and 15 of a 28 day cycle. PRP pAKT levels demonstrated sustained reductions from baseline post-Cop treatment [median inhibition: 0.4 mg/kg, 73.8% (range-94.9 to 144.0); 0.8 mg/kg, 79.6% (range-96.0 to 408.0)]. Tumor pAKT was lowered versus baseline with Cop 0.8 mg/kg in paired biopsy samples. Dose-related transient plasma glucose elevations were seen. Cop plasma exposure significantly correlated with alterations in plasma pAKT levels and glucose metabolism markers. There were 2 CRs and 6 PRs in patients with lymphoma and solid tumors combined together; 7 of 8 responders received Cop 0.8 mg/kg. The RP2D was defined at 0.8 mg/kg (or flat-dose equivalent of 60 mg) in patients with solid tumors and lymphoma.	Hyperglycemia, fatigue and hypertension.
Copanlisib with either Gemcitabine or Cisplatin (Cis) plus Gemcitabine (Gem) in advanced cancer patients [245]	Completed	Bayer	I, NCT01460537	50 patients were enrolled and treated with Cop in Arms A and B in dose escalation (*n* = 16) and dose expansion (*n* = 20), respectively. In Arm A, 8 patients received Cop at 0.6 mg/kg + Gem and 8 patients at 0.8 mg/kg + Gem (1000 mg/m^2^) in dose escalation. There was 1 DLTs. Cop 0.8 mg/kg + Gem 1000 mg/m^2^ weekly in a 3 weeks on/1 week off schedule was identified as the MTD and RP2D of this drug combination. In Arm B, 20 patients received Cop at 0.8 mg/kg with Gem + Cis and no DLTs were observed. No treatment-related deaths or relevant PK interactions were observed. Of the 2 patients enrolled in treatment schedule B with BTC, 1 achieved a CR and 3 had a PR according to RECIST criteria out of which 2 were enrolled in expansion cohort.	Hyperglycemia, nausea, diarrhea, anemia and fatigue.
Copanlisib in Chinese patients [246]	Completed	Bayer	I, NCT03498430	13 patients (11 with FL and 2 with MZL) received treatment out of which 12 patients were evaluated for efficacy. Cop PK exposure profiles were in the same range as of those from previous studies of Cop in non-Chinese patients in clinical dose of 60 mg. The ORR was 50% (95% CI: 21.1, 78.9) with 6 patients achieving a best response of a PR in 1 patient and SD in 6 patient. The disease control rate was 100% (95% CI: 73.5, 100.0). The median duration of SD was 163 days (range 106–218, including censored values).	Hyperglycemia, transient hypertension both Gr 3. However, no Gr 4 or Gr 5 adverse events observed.
Copanlisib in R/R diffuse large B-cell lymphoma (DLBCL) patients [247]	Completed	Bayer	II, NCT02391116	The full-analysis (FAS) and per-protocol sets (PPS; ≥3 doses, post-baseline scans and NGS/COO data) included 67 and 40 patients with DLBCL, respectively. In the PPS, there were 22 GCB DLBCL (2 mutants), 16 ABC DLBCL (6 mutants), and 2 unclassifiable. The ORR in the PPS was 25% (10 of 40 patients), with 5 CRs, 5 PRs, 12 SD among all patients. The ORR was 13.6% with 1 CR in GCB patients and 37.5% with 4 CRs (25%) in ABC patients. Response to Cop was 25% in patients with (2/8) and without (8/32) CD79B mutations. 5 of 10 ABC DLBCL-WTCD79B patients and 1 GCB DLBCL-mCD79B responded (ongoing > 17 cycles) to treatment. NGS analysis in 54 patients detected 348 mutations; *BCL2* (54%), *TP53* (41%), *BCL6* (30%), *MYC* (22%), *CD79B* (19%)/*A* (6%), *MYD88* (19%), *TNFAIP3* (17%), *CARD11* (13%) and *NFKBIA* (9%). Response to Cop was not significantly different based on *BCL2*, *BCL6*, *MYC* and *MYD88* mutations. There were 14 Gr 5 AEs (none drug-related). Cop treatment had encouraging responses, especially in the ABC subtype with a manageable toxicity.	Diarrhea, nausea, fatigue, fever, transient hypertension and hyperglycemia.
I. Itraconazole or Rifampin (Rif) on the PK of a single intravenous dose of Copanlisib; II. Copanlisib on cardiovascular safety in advanced solid tumors and NHL patients [248]	Completed	Bayer	I, NCT02253420	The geometric LS mean for C_max_ and AUC for co-treatment of Rif (60 mg) + Cop (600 mg) was reduced by 14% and 56%, respectively, versus Cop alone. Cop clearance increased by 2.3-fold from 22.6 to 51.4 L/h with co-administration of Rif. Cop geometric mean T_1/2_ value reduced from 39 h (Cop alone) to 17 h (Cop + Rif). Geometric mean C_max_ for Cop minor metabolite M-1 was 4-fold higher with Rif vs. Cop alone. Geometric LS mean AUC for M-1 was 0.614 (39% reduced) (90% CI: 0.504 to 0.747) compared to Cop alone. Treatments were well tolerated. Co-administration of Rif reduced Cop AUC and increased its principal metabolite M-1, consistent with the proposed metabolic pathway. The results indicated that concomitant use of Cop with strong inducers of CYP3A4 must be avoided.	-
Copanlisib in patients with relapsed, indolent, or aggressive NHL [249]	Active, not-recruiting	Bayer	II, NCT01660451	33 patients with indolent lymphoma and 51 with aggressive lymphoma received Cop. FL (48.5%) and peripheral T-cell lymphoma (33.3%) were the most common histologic subtypes. 80 patients were evaluated for efficacy. The OR rate was 43.7% (14/32) in the indolent cohort and 27.1% (13/48) in the aggressive cohort. Median PFS was 294 days (range 0–874) and 70 days (range 0–897), respectively; median duration of response was 390 days (range 0–825) and 166 days (range 0–786), respectively. Molecular analyses showed enhanced anti-tumor activity in tumors with enhanced PI3K pathway gene activation.	Hyperglycemia, hypertension, diarrhea and neutropenia.
Copanlisib in patients with persistent or recurrent endometrial cancer [250]	Active, not-recruiting	NRG Oncolog in collaboration with National Cancer Institute (NCI)	N/A, NCT02728258	The percentage of risk for all causes of mortality was 7/11 (63.64%).	Anemia, hyperglycemia, rash and cellulitis were serious adverse events.

## 4. Isoform-Specific Inhibitors

Isoform-specific PI3K inhibitors cause selective inhibition of specific PI3K isoforms (α/β/γ or δ). Out of all the isoform-specific drugs, four drugs which included Alpelisib, Idelalisib, Umbralisib and Duvelisib are approved by the FDA. These inhibitors reduce off-target toxicities although these require narrow patient selection. 

### 4.1. BYL719/Alpelisib/Piqray

Alpelisib/BYL719/Piqray is a potent oral selective PI3Kα inhibitor targeting *PIK3CA*-mutated cancers [251,252]. The role of Alpelisib has also been identified outside cancer in *PIK3CA*-related overgrowth syndromes (PROS) which are caused by mosaic gain-of-function mutations in the *PIK3CA* gene. It is shown that daily oral administration of Alpelisib in the mouse model of PROS post-tamoxifen administration, effectively suppressed *PIK3CA* pathway activation, which prevented the PROS phenotype and enhanced survival [253]. Alpelisib was approved by the FDA for *PIK3CA*-mutated, HR+, HER2- advanced breast cancer who had received endocrine therapy previously [254,255]. Most of the studies using Alpelisib are sponsored by Novartis Pharmaceuticals. One such trial tested Alpelisib in a phase I study with a HSP90 inhibitor (AUY922) to determine the MTD and/or recommended dose of expansion (RDE) of this drug combination in patients with advanced gastric cancer [256]. A dose-escalation phase Ib trial involved Alpelisib with Imatinib (a RTK inhibitor) in gastrointestinal stromal tumor (GIST) patients who had relapsed with Imatinib and Sunitinib (a RTK inhibitor) [257]. A phase Ib/II investigated Alpelisib with HER3 monoclonal antibody (LJM716) versus Taxane or Irinotecan with previously treated esophageal squamous cell carcinoma patients. The aim of the study was to evaluate the MTD and/or RP2D of this drug combination [258]. A phase Ib/II study investigated Alpelisib with an oral pan-PIM kinase inhibitor (LGH447/PIM447) in relapsed myeloma patients. The study aimed to explore the MTD and RP2D of this combination [259]. Another phase II trial tested Alpelisib in Chinese patients with NSCLC [260].

There are several active not-recruiting studies testing Alpelisib in various stages of clinical trials as mentioned below. A phase II study (SAFIR PI3K) is underway in patients with *PIK3CA-*mutated advanced breast cancer which compares Alpelisib and ER inhibitor (Fulvestrant) versus chemotherapy in these patients. The trial sponsored by UNICANCER aims to determine whether this drug combination could prolong the PFS versus maintenance chemotherapy in the above subjects [261]. A phase I study is currently testing Alpelisib with a potent, oral bioavailable selective estrogen receptor degrader (SERD), LSZ102 in advanced or metastatic ER+ breast cancer patients progressed after endocrine therapy. The study aims to identify the safety, tolerability, PK and anti-tumor efficacy of this combined regimen [262]. A phase I study is testing Alpelisib in locally advanced squamous cell carcinoma head and neck cancer (LA-SCCHN) patients in combination with standard radiation and chemotherapy, Cisplatin. The study is sponsored by University Health Network, Toronto in collaboration with Novartis [263].

### 4.2. Serabelisib/INK-117/MLN-117/TAK-117

Serabelisib/INK-1117/MLN-1117/TAK-117 is a potent oral PI3K inhibitor with an IC50 of 21 nmol/L against PI3Kα isoform. The first in human phase I study evaluated the safety and efficacy of Serabelisb in patients with advanced solid tumors [264]. A phase II study is investigating Serabelisib with oral mTORC1/2 inhibitor Sapanisertib (TAK-228) in TNBC patients to target the DNA repair pathway. The study is sponsored by Baylor Research Institute in collaboration with Takeda [265]. A phase II study evaluated the efficacy and safety of MLN0128 and the combination of MLN0128 plus Serabelisib versus Everolimus in the treatment of metastatic clear-cell renal cell carcinoma (mccRCC) patients who progressed on vascular endothelial growth factor (VEGF)-targeted therapy [266]. A phase I study tested Serabelisib with mTORC1/2 inhibitor Sapanisertib (TAK-228) in patients with advanced non-hematological patients to determine the DLTs, MTD and/or RP2D of this drug combination [267].

### 4.3. GSK2636771

GSK2636771 is an orally bioavailable PI3Kβ inhibitor with >900-fold selectivity over the PI3Kα/γ isoforms and >10-fold selectivity over the PI3Kδ isoform. A phase I study tested GSK2636771 with Enzalutamide in male patients with CRPC. The study was sponsored by GlaxoSmithKline and aimed to evaluate the safety and clinical efficacy to help guide the selection of the RP2D [268]. To date, there are few active non-recruiting trials using GSK2636771. For example, a phase Ib/IIa, open-label, dose-finding study is evaluating the safety, efficacy and clinical activity of GSK2636771 with Paclitaxel in patients with higher grade of gastric adenocarcinoma [269]. NCI is currently testing GSK2636771 in cancer patients with a *PTEN* alteration [270,271].

### 4.4. Idelalisib/GS-1101/CAL-101/Zydelig

Idelalisib/GS-1101/CAL-101/Zydelig, an oral PI3Kδ inhibitor with an IC50 value of 2.5 nM in cell-free assays was the first FDA-approved PI3K inhibitor for the treatment of patients with R/R CLL or LL/FL who progressed on prior therapies [272]. A phase Ib study sponsored by Novartis Pharmaceuticals tested Idelalisib in a dose-escalation trial with BCL201 (elective BH3-mimetic inhibitor of BCL-2) in patients with FL and MCL [273]. The primary objective of the study was to determine the safety and tolerability of this drug combination. Other completed trials with published clinical outcomes and adverse events for Idelalisib treatment are summarized in Table 2.

There are several active not-recruiting trials using Idelalisib. For example, a pilot phase I study is investigating Idelalisib with ACY-1215/Ricolinostat (a histone deacetylase inhibitor) in patients with R/R CLL. The study was sponsored by Dana-Farber Cancer Institute in collaboration with Acetylon Pharmaceuticals [274]. Idelalisib is currently being tested in combination therapies with either Rituximab plus TRU-016, mAb targeting CD37 (phase I) or with Tirabrutinib, an oral bruton’s tyrosine kinase (BTK) inhibitor (phase II) in such patients [275,276]. Another phase II trial (ILIAD) is testing Idelalisib in patients with relapse DLBCL [277]. A phase II (COSMO) is evaluating the safety and efficacy of MOR00208 with Idelalisib or Venetoclax in R/R CLL/SLL patients treated earlier with a BTK inhibitor [278].

### 4.5. Zandelisib/ME-401/PWT-143

Zandelisib/ME-401/PWT-143 is an oral, selective PI3Kδ inhibitor which is in early phase of clinical trials mostly in lymphoma patients. Zandelisib is being investigated in Japanese patients with B-cell NHL in a phase I study sponsored by Kyowa Kirin Co. Ltd., Tokyo, Japan [279].

### 4.6. AMG 319

AMG 319 is a PI3Kδ-specific inhibitor developed by Amgen. AMG 319 was tested in patients with R/R lymphoid malignancies to evaluate its tolerability, safety and PK profile [280].

### 4.7. YY-20394/Linperlisib

YY-20394/Linperlisib is an oral highly selective PI3Kδ inhibitor which is studied in patients with R/R B-cell malignancies. This drug was developed by Shanghai Ying Li Pharmaceutical Co. Ltd., Shanghai, China [281].

### 4.8. INCB050465/Parsaclisib/IBI-376

INCB050465/Parsaclisib/IBI-376 is a next-generation highly selective potent PI3Kδ inhibitor initially developed by Incyte Corporation, which sponsored a phase I study to evaluate the safety and tolerability of a pan-PIM inhibitor (INCB053914) in combination with Parsaclisib in R/R DLBCL patients [282].

There are two active non-recruiting phase I trials investigating Parsaclisib in B-cell lymphoma patients [283,284]. Another phase I study is testing Parsaclisib with Pembrolizumab in higher-grade solid tumors including lung cancer [285]. In addition, several phase II studies are exploring Parsaclisib as monotherapy in FL patients (CITADEL 203) and in MCL patients (CITADEL-204) [286,287]. Parsaclisib is being tested ± BTK inhibitor (CITADEL 205) and in combination with Bendamustine plus Obinutuzumab in patients with B-cell malignancies (CITADEL 102) [288,289].

### 4.9. Umbralisib/TGR-1202/RP5264

Umbralisib/TGR-1202/RP5264 is a next-generation highly specific, oral inhibitor targeting the δ isoform of PI3K. There are many trials testing Umbralisib sponsored by TG Therapeutics. One such phase I trial evaluated the safety and efficacy of Umbralisib as a monotherapy or with nab-Paclitaxel plus Gemcitabine or FOLFOX in patients with R/R solid tumors [290]. A phase I study evaluated the efficacy and safety of Umbralisib with Obinutuzumab (anti-CD20 monoclonal antibody) and Chlorambucil (chemotherapy) in CLL patients [291]. Another phase I study tested the efficacy of Umbralisib with Brentuximab vedotin, an anti-body drug conjugate (ADC) against CD30 in patients with HL [292].

Phase II/III studies are actively testing Umbralisib as a monotherapy in patients with non-follicular iNHL and with Ublituximab, anti-CD20 monoclonal antibody in combinational therapy in previously untreated CLL patients [293,294]. A phase I study sponsored by the Vanderbilt-Ingram Cancer Center is currently testing Umbralisib with Ruxolitinib in patients with myeloproliferative neoplasm [295]. Umbralisib has been recently approved by the FDA for treatment of patients with R/R marginal zone lymphoma (MZL) who had at least one prior anti-CD20-based therapy and for adults with R/R follicular lymphoma (FL) relapsed on three prior lines of systemic therapy [296].

### 4.10. Tenalisib/RP6530

Tenalisib/RP6530 is a novel, potent PI3Kδ/γ inhibitor with activity more than 300-fold and 100-fold over α and β isoforms, respectively. It was developed by Rhizen Pharmaceuticals and mostly tested against B-cell malignancies [297,298,299]. An open-label, phase I/II trial is assessing the safety and efficacy of Tenalisib with a histone deacetylase (HDAC) inhibitor, romidepsin in patients with R/R T-cell lymphoma [300].

### 4.11. Taselisib/GDC-0032/RG-7604

GDC-0032/Taselisib/RG-7604 is a potent class I PI3K inhibitor having strong activity against the PI3K α, δ and γ isoforms. It exhibits less potency to the ß isoform. There are several ongoing and completed trials using Taselisib in human subjects. Genentech has sponsored all the trials for Taselisib. A phase I study is investigating the safety and tolerability of Taselisib alone in patients with advanced solid tumors or NHL and with Fulvestrant in advanced HER2- and HR+ breast cancer patients [301]. Taselisib is being tested in a phase II study in patients with different cancers (excluding breast) with *PIK3CA* mutation but without *KRAS* mutation or *PTEN* loss. The study is sponsored by the NCI [302].

### 4.12. AZD8186

AZD8186 is a specific ß/δ inhibitor, developed by AstraZeneca, which is currently in phase I/II clinical studies [303].

### 4.13. AZD8835

AZD8835 is a δ/α-specific PI3K inhibitor, developed by AstraZeneca, with one trial listed so far in patients with solid tumors [304].

### 4.14. IPI-145/Duvelisib/INK1197

Duvelisib/IPI-145/INK1197 is an oral highly selective PI3K δ/γ inhibitor with IC50 values of 243 pM and 50 nM, respectively, in cell-free assay. Duvelisib (September 2018, NDA 311155) was approved by the FDA for treatment of patients with R/R CLL/SLL [305,306]. A multicenter open-label phase I study aimed to investigate the safety and PK of Duvelisib in Japanese subjects with R/R lymphoma was completed in 2017. The study was sponsored by AbbVie in collaboration with Infinity Pharmaceuticals [307]. A phase II study evaluating the clinical activity, long-term safety, and overall survival rate of Duvelisib in patients with hematologic malignancies was recently completed without significant clinical outcomes [308].

### 4.15. Leniolisib/CDZ173

Leniolisib/CDZ173 is a potent and selective PI3Kδ inhibitor. Most of the studies using Leniolisib are sponsored by Novartis Pharmaceuticals. Although no studies are known in cancer but it is implicated in immunodeficiency disorders such as activated PI3Kδ syndrome/p110δ-activating mutation causing senescent T cells, lymphadenopathy and immunodeficiency (APDS/PASLI) where it has been tested in phase II/III trials. Rao et al. have shown that treatment with Leniolisib resulted in dose-dependent inhibition of PI3Kδ pathway hyperactivation in cells expressing APDS-causative p110δ variants and in T-cell blasts derived from patients. In addition it was shown that oral administration of Leniolisib in six APDS patients resulted in dose-dependent inhibition of the PI3K/AKT pathway evaluated via ex vivo assay and had enhanced immune dysregulation. Thus, Leniolisib is known to be well tolerated and has improved laboratory and clinical parameters in patients with APDS, supporting the specific inhibition of PI3Kδ as a promising new targeted therapy in patients with APDS and other diseases characterized by hyperactivation of the PI3Kδ pathway [309]. Leniolisib is also tested in patients with primary sjögren’s syndrome (PSS) in phase II trials where its oral administration resulted in inhibition of phosphorylated AKT in ex vivo stimulated B cells, decreased the serum CXCL13 levels and reduced the frequency of circulating follicular T helper-like cells [310].

### 4.16. Eganelisib/IPI-549

Eganelisib/IPI-549 is a potent PI3K-γ inhibitor with >100-fold selectivity with an IC50 value of 16 nM. Eganelisib is the first-in-class, oral, selective PI3K-γ inhibitor identified in preclinical studies which reprogrammed macrophages from an immune-suppressive to an immune-activating phenotype and overcome resistance to checkpoint inhibitors [311]. A phase I study is evaluating the safety, efficacy, PK and PD of Eganelisib in combination with Etrumadenant/AB928 (a selective dual adenosine receptor, A2aR/A2bR antagonist) plus pegylated liposomal doxorubicin (PLD) in patients with TNBC or gynecological tumors [312]. Active not-recruiting and completed trials with specific clinical outcomes are summarized in Table 2.

**Table 2 ijms-22-03464-t002:** Summary of trials, outcomes and adverse events associated with isoform-specific PI3K inhibitors in various phases of clinical studies.

Treatment	Status	Sponsor	Phase and NCT	Clinical Outcomes	Adverse Effects
**BYL719/Alpelisib/Piqray**					
Alpelisib (Alp) + Paclitaxel (Pac) in advanced solid tumors [313]	Completed	Novartis Pharmaceuticals	I, NCT02051751	19 patients were treated with Alp once daily (300 mg, *n* = 6; 250 mg, *n* = 4; 150 mg, *n* = 9) plus Pac (80 mg/m^2^) once weekly. 5/12 patients evaluated for MTD had DLTs. The MTD of Alp was identified as 150 mg once daily when given with Pac (80 mg/m^2^ once weekly dose). Best overall response was PR in 3, SD in 11, PR in 4 patients, respectively, with 1 deceased patient in total *n* = 18 patients evaluated for response. AUC_(0-inf)_ and C_max_ of Pac was comparable on cycle 1 of both day 1 and 8 and was independent of Alp dose. C_max_ and AUC_0–24_ of Alp increased in a dose-dependent manner with few exceptions. The steady-state exposure of Alp (150–300 mg) had no impact on Pac metabolism. T_max_ of Alp on cycle 1 day 8 ranged from 3.17 to 4.17 h and independent of Alp dose.	The most common TEAEs were diarrhea, hyperglycemia, anemia, asthenia and nausea. Gr 3/4 TEAEs were hyperglycemia, anemia, diarrhea, lymphopenia, neutropenia and leukopenia.
Alpelisib + Gemcitabine (Gem) and Paclitaxel in advanced pancreatic cancer [314]	Completed	H. Lee Moffitt Cancer Center and Research Institute in collaboration with Novartis Pharmaceuticals	I, NCT02155088	15 patients were enrolled with cohort 1 and 2 (*n* = 3), cohort 3 (*n* = 9), but 4 were replaced as 3 patients withdrew consent before evaluation and 1 patient missed over 10 days of treatment. Drugs were given as follows cohort 1: (Alp: 250 mg/day + Gem: 800 mg/m^2^ + Pac: 125 mg/m^2^); cohort 2 (Alp: 250 mg/day + Gem: 1000 mg/m^2^ + Pac: 125 mg/m^2^); cohort 3 (Alp: 300 mg/day + Gem: 1000 mg/m^2^ + Pac: 125 mg/m^2^); cohort 4 (Alp: 350 mg/day + Gem: 1000 mg/m^2^ + Pac: 125 mg/m^2^). 1 patient developed PRES. 1 patient in cohort 3 had a DLT (Gr 3 nausea and vomiting). Out of 8 patients evaluated for response, 2 patients had SD, 5 had PRs and 1 had PD. The median PFS was 5.36 months (95% CI: 1.6—10 months) and overall survival was 8.74 months (95% CI: 3.8–21.2 months), respectively. The triple combination could be safely given to patients with a dose of Alp upto 250 mg per day.	Hyperglycemia, anemia, reduced neutrophil count, nausea and vomiting.
Alpelisib + Letrozole (Let) or Exemestane (Exe) in patients with HR+ locally advanced unresectable or MBC [315]	Completed	Memorial Sloan Kettering Cancer Center with Novartis Pharmaceuticals	I, NCT01870505	14 patients were enrolled and treated in two Arms. Arm A (*n* = 7; Alp + Let) and Arm B (*n* = 7; Alp + Exe). Alp was administrated at 250 mg (DL-1) or 300 mg (DL0). All patients in both arms were evaluated and demonstrated similar toxicities. DLTs seen included maculopapular rash, hyperglycemia and abdominal pain. *PIK3CA* status was found to be WT/mutant/unknown in 5/8/1 patients, respectively. Out of 9 patients evaluated for response, 5 patients had SD and 1 patient had a PR with liver metastasis. Continuous Alp dosing with Let or Exe showed promising results. However, onset of maculopapular rash encouraged the use of dermatological approaches with a non-continuous dosing schedule for further studies.	Common TEAEs were Gr 2 mucositis, hyperglycemia, dyspepsia, fatigue. Gr 3 maculopapular rash, skin biopsies showed non-specific inflammation, hyperglycemia and abdominal pain. Gr 4 TEAEs not reported.
Alpelisib + Fulvestrant (Ful) in ER+ advanced breast cancer patients with *PIK3CA* alterations [316]	Completed	Novartis Pharmaceuticals	I, NCT01219699	87 patients received escalating once daily doses of Alp (300 mg; *n* = 9, 350 mg; *n* = 8 and 400 mg; *n* = 70) with fixed-dose of Ful (500 mg/daily). 1/10 patients had DLT during escalation study with Alp dose of 400 mg. The MTD and RP2D of Alp given with Ful (400 mg daily dose) was 300 mg/day. 9 patients permanently discontinued therapy owing to AEs. The median PFS at the MTD of Alp was 5.4 months (95% CI: 4.6–9.0 months). The median PFS with Alp (300–400 mg) once daily dose plus Ful in patients with *PIK3CA* alterations was 9.1 months (95% CI: 6.6–14.6 months) vs. patients with WT tumors which was 4.7 months (95% CI: 1.9–5.6 months). ORR in patients with *PIK3CA* alterations (*n* = 29) was 29% (95% CI, 17–43%), with no objective tumor responses in the WT group (*n* = 32). AUC_(0-24)_ to Alp (300 and 350 mg/day) plus Ful was similar to single-agent Alp at the same doses in a fed state. C_max_ profiles were similar across doses.	Gr 3 events such as vomiting, fatigue, decreased appetite and Gr 2 diarrhea observed. Gr 3/4 AEs were hyperglycemia and maculopapular rash.
Alpelisib in adult Japanese patients with advanced solid tumors [317]	Completed	Novartis Pharmaceuticals	I, NCT01387321	33 patients were enrolled in the trial. 25/33 patients received Alp at 90–400 mg/day in a dose escalation and 8/33 received Alp at 350 mg/day (RP2D) in the expansion cohort, respectively. 3 DLTs observed in 2/5 patients in the escalation cohort at 400 mg Alp. The MTD was not determined. The ORR (CR or PR) was found to be 3.0% (1/33 patients) (95% CI: 0.1–15.8). The DCR (CR or PR or SD) was 57.6% (19/33 patients). The DCRs at RP2D in the expansion cohort and in all patients were 75.0% (6/8 patients) and 78.6% (11/14 patients), respectively. The median PFS at RP2D was 3.4 months (95% CI: 2.8–5.6).The estimated β for C_max_, AUC_(0__–24)_, and AUC_(0-last)_ were 0.82 (90% CI: 0.499–1.142), 1.06 (90% CI: 0.791–1.327) and 1.05 (90% CI: 0.772–1.337), indicating that the dose proportionality of Alp. pAKT, pS6 and p4EBP1 levels in skin biopsy decreased post-Alp treatment. *PIK3CA* mutations or amplifications were seen in 7; *TP53* alterations in 11; *BRCA2* in 3; *KRAS*, *BRAF* and *PTEN* mutations in 7, 3 and 1 patients, respectively.	Hyperglycemia, maculopapular rash, diarrhea and decreased appetite.
Alpelisib + BGJ398 (BGJ) in solid patients with *PIK3CA* mutation [318]	Completed	Novartis Pharmaceuticals	I, NCT01928459	A total 62 patients were recruited. The MTD was defined as 125 mg/day BGJ in combination with 300 mg/day Alp based on 32 patients’ data obtained during dose escalation. DLTs were seen in 4 patients. 24 patients were enrolled in the expansion cohort. At the MTD, 61% of patients had ≥1 BGJ/Alp dose reduction. 8 patients had PRs, out of which 4 patients were confirmed but with different cancers. One patient with FGFR3-TACC3 fusion urothelial carcinoma had a PR and a complete shrinkage lasting 4 months.	Diarrhea, fatigue, nausea, hyperphosphatemia and hyperglycemia.
Alpelisib + Trastuzumab-MCC-DM1 (T-DM1) in HER2+ MBC patients progressed on prior Trastuzumab or Taxane therapy [319]	Completed	Northwestern University with NCI and Novartis Pharmaceuticals	I, NCT02038010	The MTD for Alp was found to be 250 mg daily (*n* = 17). 5 patients in cohort 1 had 2 DLTs. Median follow-up was 11.6 months (range, 0.3–19.5). Median PFS was 6 months (95% CI: 2.9–10.6). The median PFS in 11 patients without prior T-DM1 was 4.3 months (95% CI: 2.0–8.8) and in 6 patients with prior T-DM1 was 10.6 months (95% CI: 1.6–12.6). Combination of Alp 250 mg/daily + T-DM1 was considered as a safe dose in HER2+ MBC patients with prominent anti-tumor activity, even in previously treated patients with T-DM1.	Hyperglycemia, fatigue, nausea, rash, weight loss, hypertension and pancreatitis.
Alpelisib + Everolimus (Eve) and Alpelisib + Everolimus + Exemestane (Exe) in patients with advanced solid tumors or HR+/HER2- breast cancer [320]	Completed	Novartis Pharmaceuticals	I, NCT02077933	In double treatment group, 13 patients received Alp: 300 mg/day (*n* = 6), 250 mg (*n* = 6) or 200 mg (*n* = 1) plus Eve 2.5 mg daily dose. With Alp (300 mg/daily) treatment group, 2/3 patients had DLTs (Gr 3 diarrhea or Gr 2 hyperglycemia), 2/6 patients at 250 mg Alp had DLTs of Gr 2 hyperglycemia or Gr 4 hypocalcemia (*n* = 1 each) and 1/1 patient at 200 mg had DLTs of Gr 3 diarrhea and stomatitis. The MTD/RDE was defined as Alp at 250 mg/daily + Eve 2.5 mg/daily. 4 patients at the MTD/RDE had SD. PK parameters for Alp and Eve were not significantly altered by co-treatment. In the triplet combination arm, 7 patients received Alp at 200 mg daily with Eve 2.5 mg/daily and Exe at 25 mg. For this arm, the MTD/RDE was achieved at Alp 200 mg + Eve 2.5 mg + Exe 25 mg. Of 7 patients enrolled for triple treatment, 1 had a Gr 3 acute kidney injury.	Common AEs were hyperglycemia, diarrhea and enhanced lipid levels. For triple combination, common AEs were fatigue and enhanced γ-glutamyltransferase levels.
Alpelisib + Binimetinib/MEK162 in advanced solid tumors [321]	Completed	Array BioPharma	I, NCT01449058	58 patients were enrolled and received Alp (80–270 mg once daily) with MEK162 (30 or 45 mg twice-daily dose) in 28 day cycle. DLTs were observed in 9 patients (19%).The MTD of this combination was defined at Alp 200 mg once daily + MEK162 45 mg twice daily doses. Out of 4 ovarian cancer patients with *KRAS*-mutation, 3 had PRs. PR was also confirmed in 1 patient with *NRAS*-mutant melanoma and 1 patient with *KRAS*-mutant endometrial cancer (unconfirmed). SD lasting over 6 weeks was reported as best response for 18 patients (31%).	Diarrhea, nausea, vomiting, reduced appetite, rash, pyrexia, fatigue and hyperglycemias were common AEs. Other side effects were elevated creatine phosphokinase levels, gastrointestinal and skin disorders.
LGX818/Encorafenib (Enco) + Cetuximab(Cetux) or with Cetuximab and Alpelisib in patients with *BRAF*-mutant metastatic colorectal cancer [322]	Completed	Pfizer	II, NCT01719380	Total of 102 patients were enrolled (double, *n* = 50; triplet, *n* = 52). Patients in the doublet group were treated with Enco 200 mg daily once dose + Cetux per label. Patients in the triplet arm received an additional Alp at 300 mg oral daily dose with Enco + Cetux. The median PFS of triplet vs. doublet groups was 5.4 months (95% CI: 4.1–7.2) and 4.2 months (95% CI: 3.4–5.4), respectively, with a hazard ratio of 0.69 (95% CI: 0.43–1.11). The ORR was 27% (95% CI: 16–41%) in the doublet and 22% (95% CI: 12–36%) in the triplet, respectively, after 73 events. The interim OS analysis (triplet vs. doublet) showed a hazard ratio of 1.21 (95% CI: 0.61–2.39). The median OS was 15.2 month on the triplet group and was not reached on the double combination after 35 events.	Hyperglycemia, anemia and increased lipase levels.
Alpelisib + Letrozole (Let) or Letrozole + Buparlisib (Bup) in HR+ and HER2- postmenopausal women (NEO-ORB) [323]	Completed	Novartis Pharmaceuticals	II, NCT01923168	A total 257 patients were enrolled (Let + Alp; *n* = 131 and placebo + Let; *n* = 126). Patients were randomized into 2 cohorts (*PIK3CA*-WT or MUT) and received Let (2.5 mg/day) + Alp (300 mg/day)/Bup (100 mg/day) or placebo. The primary objective of the study was not met; ORR in the Alp compared to placebo arm was 43% vs. 45% and 63% vs. 61% in the MUT and WT cohorts, respectively. p-AKT levels decreased significantly in Alp+ Let arm vs. placebo + Let in MUT cohort (cycle 1 of day 15). There were no substantial differences in Ki-67 levels in the Alp + Let vs. placebo + Let treatment arms. ORR was lower in both arms of MUT vs. WT cohort.	Hyperglycemia, diarrhea, rash, nausea, fatigue, stomatitis, decreased appetite, alopecia and headache.
Alpelisib/Buparlisib + Tamoxifen (Tam) and Goserelin Acetate in premenopausal women with HR+/HER2- locally advanced or MBC (B-YOND) [324]	Completed	Novartis Pharmaceuticals	I, NCT02058381	12 patients were enrolled. Gr 1, 6 patients treated with Alp (350 mg QD) + Tam (20 mg QD) + Goserelin Acetate (3.6 mg Q28D). There was no DLTs observed. In Gr 2, 6 patients were treated with Bup (100 mg QD) + Tam (20 mg QD) and Goserelin Acetate (3.6 mg Q28D). In this group, 1 Gr 3 DLT (increase ALT/AST) was observed. Alp with Tam + Goserelin Acetate had a manageable toxicity profile vs. Bup Tam + Goserelin Acetate. No patients in group 1, and 5/6 patients in group 2 had discontinued treatment due to AEs.	Group 1. Hypokalemia, rash, anemia, leukopenia and infections, with no Gr 4 toxicities. Group 2. Liver toxicity, psychiatric disorders, rash, hypertension and hyperglycemia.
Alpelisib, LJM716 (LJM) and Trastuzumab in HER2+ MBC patients [325]	Active, not recruiting	Memorial Sloan Kettering Cancer Center in collaboration with Novartis Pharmaceuticals	I, NCT02167854	8 patients were enrolled. Patients received Trans weekly IV at 2 mg/kg, LJM at 20 mg/kg, and escalating dose of PO QD Alp starting at 250 mg. 7 patients received Alp at starting dose of 250 mg. 1 patient treated at 300 mg had a DLT (supraventricular tachycardia). Another patient had a DLT of Gr 3 transaminitis at the Alp dose of 250 mg. Overall, toxicities limited drug delivery with only 72%/83% of total planned Alp/LJM doses given. 5/6 patients with SD. Gastrointestinal and metabolic toxicities limited drug delivery and dose escalation of Alp although anti-tumor activity was seen in triple combination of Alp + LJM + Trans.	Diarrhea, hyperglycemia, hypokalemia, mucositis and transaminitis.
Alpelisib + LEE011(LEE) with Letrozole (Let) in adult patients with advanced ER+ breast cancer [326]	Active, not recruiting	Novartis Pharmaceuticals	I/II, NCT01872260	10 patients were treated with 600 mg LEE + Let (group A1) and 7 patients with 300 mg Alp + Let (group A2). 1 DLT (Gr 4 neutropenia) seen in group A1. PK for LEE and Alp on days 1 and 21 and Let on day 1 are comparable to single agent. In group 1 (*n* = 6): 1 patient had PR, 2 patient had SD, 1 had NCRNPD and 2 patients had PD. In group 2 (*n* = 5); 2 patients had SD, and 3 patients had NCRNPD. Let + LEE or Alp had an acceptable safety profile and preliminary signs of clinical activity.	Common AEs were neutropenia and nausea in group A1. Hyperglycemia, decreased appetite, diarrhea and nausea in group A2.
Alpelisib + Cetuximab (Cetux) + IMRT in stage III–IVB HNSCC patients [327]	Active, not recruiting	Memorial Sloan Kettering Cancer Center with Novartis Pharmaceuticals	I, NCT02282371	11 patients were evaluated (stage III-1, IVA-9, IVB-1; oropharynx primary-9, unknown primary-2; HPV+ −10). Patients with stage III–IVB HNSCC received Cetux at 400 mg mg/m^2^ IV as loading dose prior to IMRT, followed by 250 mg weekly infusions of Cetux during IMRT. Alp was given in 3 DLs: 200 mg/day (DL1), 250 mg/day (DL2) and 300 mg/day (DL3) during IMRT in a 3 + 3 dose-escalation study. 3 patients treated on DL1 or DL2 did not have DLTs. 2 patients on DL 3 had a DLT of Gr 3 mucositis. All patients with completed treatment demonstrated a CR and disease-free condition. 1 of 6 patients with an activating *PIK3CA* mutation had a rapid response on serial intra-treatment MRIs. Alp at 300 mg exceeds the MTD and a RP2D would be proposed on completion of expansion of DL2.	Most common AEs in all grades included mucositis, weight loss and hyperglycemia. Gr 3 and 4 AEs seen in at least 2 patients include: mucositis, dysphagia and decreased lymphocyte count.
Alpelisib + Letrozole (Let) in patients with HR+ post-menopausal MBC patients [105]	Active, not recruiting	Vanderbilt-Ingram Cancer Center in collaboration with NCI	I, NCT01791478	26 patients enrolled and received Let at 2.5 mg/day and Alp at 300 mg/day. The MTD of Alp with Let was 300 mg/day (*n* = 20) and 1 DLT (Gr 3 rash) was observed at 350 mg of Alp (*n* = 2). 23/26 patients evaluated for response; 3 discontinued due to toxicities and 5 had PR. The CBR (lack of disease progression ≥6 months) observed in 9 patients was 35%; (95% CI: 17%, 56%) which included 44% in patients with *PIK3CA* mutation and 20% in *PIK3CA* WT tumors; (95% CI: 17%, 56%). The CBR (lack of disease progression ≥12 months) in rest 8 patients was 31%; 95% CI [14%; 52%]. 6 of these 8 patients had *PIK3CA* mutation. Common alterations seen in >10% of patients were *TP53*, *GATA3*, *BRCA1/2*, *PIK3CA* mutations with MCL1, FGFR1, CCND1, FGF3/4/19, CCND1 amplifications and deletions/truncations in *PTEN. PIK3CA*^H1047R^ was the most common mutation and patients with this mutation had a durable response while other with *PIK3CA* P447_L455del had PR. Patients with *FGFR1* and *FGFR2* amplification, *KRAS* and *TP53* mutations progressed on treatment. 2 patients with *PIK3CA* and activating *ESR1* mutation in their primary tumor showed SD or PR.	Hyperglycemia, nausea, fatigue, diarrhea and rash.
Alpelisib + Paclitaxel (Pac) in locally advanced HER2- breast cancer [328]	Active, not recruiting	University of Kansas Medical Center in collaboration with Novartis Pharmaceuticals	I/II, NCT02379247	10 patients enrolled in a phase I 3 + 3 dose-escalation study and treated with 3 doses of oral Alp (250 mg/day, 300 mg/day and 350 mg/day) + 100 mg/m^2^ nab-Pac on days 1,8,15 of 28 day cycles. No DLTs were observed in phase I with above doses of Alp. In phase II, *n* = 33 patients were treated with Alp 350 mg/day at RP2D + 100 mg/m^2^ nab-Pac on days 1,8,15 of 28 day cycle. ORR for patients who received RP2D of Alp was 55% (18/33). Median PFS was 9 months (95% CI: 6–12). ORR in total 42 patients evaluated for response was 57% (24/42) (CR in 2, PR in 22) and 21% of patients had SD. 40% (17/42) showed *PIK3CA* mutation in ct DNA or tissue. Patients with *PIK3CA* mutation had an ORR of 65% with PFS of 13 months vs. 7 months in those without *PIK3CA* mutations (hazard ratio = 0.39, *p* = 0.03).	Hyperglycemia, neutropenia, anemia and diarrhea were common AEs in phase II study.
Alpelisib + Fulvestrant (Ful) in *PIK3CA*-mutated HR+ advanced breast cancer patients who had received endocrine therapy previously (SOLAR-1) [254]	Active, not recruiting	Novartis Pharmaceuticals	III, NCT02437318	572 patients enrolled into two cohort with *n* = 341 patients with *PIK3CA* mutations. Patients were treated with Alp (300 mg/day) and Ful (500 mg IV on day 1 and 15 of cycle 1 and on day 1 of subsequent 28 day cycle). In patients with *PIK3CA* mutation; PFS at a median follow-up of 20 months was 11.0 months (95% CI: 7.5–14.5) in Alp + Ful arm (*n* = 169) vs. 5.7 months (95% CI: 3.7–7.4) in placebo + Ful arm (*n* = 172) (hazard ratio: 0.65; 95% CI: 0.50–0.85). The median PFS was 7.4 months (95% CI: 5.4–9.3) in the Alp+ Ful arm (*n* = 115) and 5.6 months (95% CI: 3.9–9.1) in the placebo + Ful arm (*n* = 116) (hazard ratio 0.85; 95% CI; 0.58 to 1.25). In the *PIK3CA* mutation cohort treated with Alp+ Ful (*n* = 169); 1 patient had CR, 44 patients had PR, 58 had SD and 16 patients had PD. In the *PIK3CA* mutation cohort treated with placebo + Ful (*n* = 172); 2 patients had CR, 20 patients had PR, 63 patients had SD and 53 patients had PD. ORR in patients without *PIK3CA*-mutation was 26.6% in Alp + Ful arm (*n* = 45) vs. 12.8% in placebo + Ful arm (*n* = 22), respectively.	Hyperglycemia, rash and diarrhea.
Alpelisib/Buparlisib with PARP inhibitor, Olaparib (Ola) in ovarian cancer or TNBC patients [329]	Completed	Dana-Farber Cancer Institute with Novartis Pharmaceuticals and AstraZeneca	Ib, NCT01623349	A total patients (*n* = 34) were enrolled in dose-escalation (*n* = 28) and dose expansion (*n* = 6). 30 patients had ovarian and 6 had breast cancer. The patients were treated as in 4 DLs as Alp 250 mg/daily + Ola 100 mg twice a day (DL0); Alp 250 mg/daily + Ola 200 mg twice a day (DL1); Alp 300 mg daily + Ola 200 mg twice a day (DL 2) and Alp 200 mg daily + Ola 200 mg twice a day (DL 3). The MTD and RP2D were identified as Alp 200 mg daily and Ola 200 mg twice daily both administrated orally. Of the 28 ovarian cancer patients in expansion study, 10 (36%) had PR, 3 (11%) had PD, 14 (50%) had SD and 1 (4%) had best OR using RECIST v1.1. 64% of ovarian cancer patients were gBRCAwt and 93% were platinum R/R. ORR of 31.3% was seen in platinum resistant patients vs. 33.3% in *BRCA*wt patients. The response rate of Ola/Alp in g*BRCA*mut patients was 30% in all patients and 33.3% in g*BRCA*mut patients with platinum resistant disease. Alp or Ola C_max_ values remained unaffected by each other dosing.	Hyperglycaemia, nausea and increased ALT were common TEAEs.
**Serabelisib/INK-117/MLN-117/TAK-117**					
Serabelisib (Sera) in patients with advanced cancer [264]	Completed	Millennium Pharmaceuticals, Inc	I, NCT01449370	71 patients were enrolled with *PIK3CA* mutation in *n* = 61 patients. Sera was given at once daily dose ranging from 100 to 300 mg (*n* = 24); intermittently on Mon, Wed and Fri (MWF) at a dose of 200–1200 mg (*n* = 27); on Mon, Tue, Wed (MTuW) at a dose of 200–900 mg (*n* = 20). The MTD of Sera for the once daily schedule was defined at 150 mg. For both intermittent schedules, the MTD was 900 mg. The mean T_1/2_ of Sera was ~11 h. AUC_ss_ of Sera was ~105,000 ng/h/mL for once-daily dosing and ~470,000 ng/h/mL for the intermittent dosing schedule. C_avg_ for once daily dosing was ~16 h/week vs. ~75 h/week for the two intermittent dosing schedules. Sera at dose 200 and 900 mg had reduced pS6 expression in skin biopsies (*n* = 60). 17 patients (28%; all *PIK3CA* mutation) had SD lasting 3 months. CBR was 34% rates (40%, 32%, 31%), and median durations of clinical benefit (4.8, 4.8, 5.3 months) were comparable between the once daily, MWF and MTuW dosing schedules, respectively.	Hyperglycemia, elevated ALT/AST, nausea, vomiting and diarrhea.
Sapanisertib/TAK228 (Sap) + Serabelisib + Paclitaxel and Sapanisertib + Paclitaxel (Pac) in patients with endometrial cancer [330]	Completed	Millennium Pharmaceuticals, Inc. in collaboration with EUTROC and ENITEC	II, NCT02725268	Patients were enrolled and treated as follows: Group 1: Pac 80 mg/m^2^ (*n* = 90). Group 2: Pac 80 mg/m^2^ + Sap 4 mg (*n* = 90). Group 3: Sap 30 mg (*n* = 41). Group 4: Sap 4 mg + Sera 200 mg (*n* = 20). Group 4: PFS was 2 months (95% CI: 1.5–3.3 months) and OS was 11.1 months (95% CI: 2.7–17.5 months). Group 4 vs. Group 1: The hazard ratio for PFS was 2.56 (95% CI: 1.46–4.48). The hazard ratio for OS was 1.43 (95% CI: 0.79–2.57).The ORR was 35%, odds ratio was 0.47 (95% CI: 0.16–1.37). The CBR was 5% with odd ratio 0.07(95% CI: 0.01–0.67).	Fatigue, enhanced AST/ALT, diarrhea, anemia, constipation, decrease appetite and asthenia.
**GSK2636771**					
GSK2636771 (GSK) in advanced solid tumors patients with *PTEN* deficiency [331]	Completed	GlaxoSmithKline	I, NCT01458067	53 patients were enrolled into 7 dose escalations and treated with GSK (25–500 mg) in 3 and 4 pharmacodynamic cohorts (50–350 mg). The MTD/RP2D was defined at 400 mg QD. Hypophosphatemia and hypocalcemia were the DLTs observed at the MTD. C_max_ was achieved at 4–6 h post-single and repeat dosing with T_1/2_ of 17–38.6 h was obtained across cohorts. Enhancement in C_max_ and AUC_(0-24 h)_ values were dose proportional up to 350 mg, but not at 400 and 500 mg. p-AKT levels decreased in >50% in 18 patients and >80% in 12 patients post-1–10 h of dosing (*n* = 20). 1 patient with prostate cancer had a PR and 13 patients had SD.	Diarrhea, vomiting, nausea, fatigue, anemia, abdominal pain and decreased appetite.
**Idelalisib/GS-1101/CAL-101**					
Idelalisib (Ide) in patients with R/R hematologic malignancies [332,333,334]	Completed	Gilead Sciences	I, NCT00710528	A total of 64 patients with iNHL were enrolled. Ide was administrated at dose 50–350 mg once or twice daily in the primary study and 150–350 mg in the extension study, respectively. Post-48 weeks of treatment, 30% of patients (*n* = 19) enrolled into a further extension study. Ide displayed non-time-dependent linear pharmacokinetics which was a less-than dose-proportional increase in exposure and achieved steady state by day 8. Ide treatment caused disease regression in 46/54 (85%) of patients with an ORR of 30/64 (47%) (95% CI: 34–60%) with 1 patient had a CR (1.6%), PR reported in 25 patients (39%) and MR in 4 patients (6%) and 25 patients (39%) had SD, 4 patients (6%) had PD, and 5 patients (8%) were not evaluable. Median duration of response was 18.4 months (range, 0.03–34), median PFS was 7.6 months (range, 0.03–37). 46.4% of patients remained progression free at 48 weeks. 35 patients treated on lower dose or intermittent dosing regimens had a median PFS of 3.7 (0.5–33) months [332]. 54 patients with CLL were enrolled. Ide was given daily or twice daily at 50–350 mg. 54% of patients (*n* = 29) discontinued. DLTs were not observed within 4 weeks of therapy and MTD could not be defined. Decreased p-AKT T308 levels and CLL-derived cytokines was seen in these CLL patients after 4 weeks Ide therapy. The ORR was 72% (95% CI, 58.4–83.5%) (39/54), with 39% (21/54) of patients meeting the criteria for PR per iwCLL 2008 criteria and 33% (18/54) meeting PR criteria in the presence of treatment-induced lymphocytosis. The TTR in *n* = 39 patients was 1.0 month and DoR was 16.2 months. The median PFS patients (*n* = 54) was 15.8 months and median OS was not reached with 75% of patients surviving at 36 months. The median PFS for patients (*n* = 28) treated at ≥150 mg (RP2D) BID of Ide was 32 months vs. 7 months (*n* = 26) patients treated at lower doses. The ORR for patients (*n* = 13) with del17p and/or *TP53* mutation with 5 prior regimens or 85% of whom were refractory to their most recent prior treatment was 54% (7/13), and the median PFS was 3 months. Out of these patients, 7 were treated at doses of ≥150 mg BID and the median PFS was 5 months. The median PFS for patients without del17p and/or *TP53* mutation and treated at doses of ≥150 mg BID or higher was 41 months. In addition, Ide treatment resulted in nodal responses in 81% of patients [333]. In this study, 40 patients with R/R MCL were treated with 50–350 mg QD or BID Ide for 48 weeks. C_max_, C, and AUC_τ_ showed modest increases in exposure above the dose level of 150 mg BID. ORR reported in 16/40 patients was 40% (95% CI: 24.9–56.7), CR in 2/40 patients (5%), PR in 14/40 patients (35%), PD in 4/40 (10%) and SD in 19/40 patients (47.5%). The median TTR for the 16 patients was 1.1 months (range, 0.9–9.4 months). Median DoR (*n* = 16) was 2.7 months (95% CI: 1–8.1 months; 0.3–28.2 months), median PFS (*n* = 40) was 3.7 months, (95% CI: 2.7–8.2 months; 0.03–30.1 months). 9 (18%) patients discontinued therapy due to AEs [334].	In NHL patients, Gr 3 or higher AEs were diarrhea, fatigue, nausea, rash, pyrexia and chills. Other abnormalities included neutropenia, anemia, thrombocytopenia and elevation in serum transaminase level [332]. In CLL patients Gr 3 or higher AEs included pneumonia, neutropenic fever, colitis, cellulitis and diarrhea [333]. In MCL patients common AEs observed were diarrhea, nausea, pyrexia, fatigue, rash, decreased appetite, upper respiratory infection, pneumonia and ALT/AST increase [334].
Idelalisib in Japanese patients with R/R iNHL and CLL [335]	Completed	Gilead Sciences	I, NCT02242045	A total of 6 patients with FL (*n* = 3)/CLL (*n* = 3) were enrolled and received Ide at a dose of 200 mg BID. No DLTs were observed in these patients. C_max_ was achieved at 2.50 h (1.50–4.00 h). The mean Ide plasma concentrations reduced over time but remained detectable in most patients at 12 h. 5 patients (FL: *n* = 2, CLL: *n* = 3) had PR with a median duration was 14.5 months (range, 3.7–31.3 months).	Common AEs were diarrhea, gastritis, insomnia and pyrexia. Common ≥ Gr 3 AEs were diarrhea, increased transaminase levels and decreased appetite.
Lenalidomide (Lena) with or without Idelalisib in R/R MCL or FL patients [336]	Completed	Alliance for Clinical Trials in Oncology in collaboration with NCI, Gilead Sciences and Celgene Corporation	I, NCT01838434 and NCT01644799	A total of 11 patient which included MCL (*n* = 3) and FL (*n* = 8) were enrolled. Ide was given at 150 mg BID in combination with Lena at 15 mg (1–21 days) QD and Rituximab 375 mg/m^2^ weekly dose during cycle 1 and on day 1 of subsequent 28 day cycle. Both studies were later amended to remove Rituximab due to severe toxicities. 8 patients discontinued due to AEs. There were no responders in the MCL group; 1 patient had SD and 2 had PD. Among 8 FL patients, 1 patient had CR and 4 had PR (ORR 45%), 1 patient had SD and 2 had PD. The median PFS in the patients with FL was 14.4 months (5.7–29.5 months).	In patients with MCL, common AEs were ALT elevation and rash. In FL patients, neutropenia and rash were common side effects.
Idelalisib + chemotherapeutic agents, immunomodulatory agents and anti-CD20 mAbs in patients with R/R iNHL, MCL or CLL [337]	Completed	Gilead Sciences	I, NCT01088048	7 patients were initially enrolled consisting of 5 patients with FL, 1 patient with LL and 1 patient MCL. Treatment included Lena at 5 mg (days 8–21 cycle 1, days 1–21 thereafter), Rituximab at 375 mg/m^2^ IV day 1, and Ide 150 mg BID from day 1 (cycle 1, 35 days; subsequent cycles, 28 days). Out of 4 patients evaluated for response, 2 patients had Gr 3 DLTs and 2 patients had Gr 4 ALT elevation due to which Lena treatment was held in all patients. 4 patients (57%) had elevation in bilirubin levels.	ALT elevation was the most common AE reported.
Idelalisib in low-grade lymphoma patients [338]	Completed	Gilead Sciences	I/II, NCT01306643	Single-cell suspensions of biopsy specimens from healthy and iNHL patients were incubated with or without Ide and stained with pAKT S473 and pS6 S235/6 antibodies. Both cells demonstrated complete or near-complete inhibition of p-AKT levels. PI3K distal downstream signaling (S6K) was completely inhibited by Ide in healthy B cells. The degree inhibition of S6K expression was variable between iNHL patients. Variable PI3K-independence was observed amongst other lymphomas such as MCL.	-
Idelalisib in NHL patients [339]	Completed	Gilead Sciences	II, NCT01393106	25 NHL patients were enrolled. Ide was given at 150 mg BID that was increased upto 300 mg at the time of disease progression. The ORR was 20% (95% CI: 6.8%, 40.7%) with a TTR of 2.0 months. 17/25 (68%) patients experienced a reduction in target lesions with 1 complete remission and 4 partial remissions. The median DoR was 8.4 months and median PFS was 2.3 months.	Common AEs were enhanced ALT, diarrhea and colitis. 1 patient died due to severe hypoxia.
Idelalisib in indolent B-cell NHL patients (DELTA) [340]	Completed	Gilead Sciences	II, NCT01282424	125 iNHL patients who progressed on Rituximab therapy were given Ide 150 mg BID until disease progression or patient withdrawal. The response rate was 57% (95% CI, 48–66) with 71 responses in 125 patients. 7 patients (6%) had a CR, 63 patients (50%) had a PR and 1 patient (1%) had a minor response. The median PFS was 11.0 months (0.03 to 16.6) with 47% of patients remaining progression-free at 48 weeks and OS at 1 year was approximately 80%. The median follow-up time was 9.7 months.	Neutropenia, diarrhea, dyspnea, increase in ALT/AST and pneumonia.
Idelalisib + Bendamustine (Ben) and Rituximab (Ritu) for previously treated CLL patients (TUGELA) [341]	Completed	Gilead Sciences	III, NCT01569295	416 patients were enrolled and treated with Ide (150 mg BID) + Ben (70 mg/m^2^ IV on days 1 and 2 for six 28 day cycles) + Ritu (375 mg/m^2^ IV on day 1 of cycle 1 and 500 mg/m^2^ on day 1 of cycles 2–6) in 207 patients or placebo + Ben + Ritu in *n* = 209 patients. The median PFS was 20.8 months (95% CI: 16.6–26.4) in Ide arm and 11.1 months (95% CI: 8.9–11.1) in the placebo arm (hazard ratio 0.33, 95% CI: 0.25–0.44; *p* < 0.0001) with a median follow up of 14 months. TEAEs led to death in 23 (11%) patients in Ide arm and 15 (7%) in placebo arm, along with 6 deaths from infections in the Ide arm and 3 from infections in the placebo arms, respectively. The ORR in the Ide arm was 22/38 (58%) vs. 9/40 (23%) in the placebo arm. 10/40 (25%) of patients in the placebo arm had PD as the best overall response, compared with 1 patient in the Ide Arm 1/38 (3%). Median OS in Ide arm was NR (95% CI: 12.2, N.R.) and was 20.3 months (95% CI: 12.1, 31.6) in the placebo arm. 97 patients discontinued due to AEs and other factors.	Neutropenia, pneumonia, pyrexia and febrile neutropenia were common AEs in Ide group. Neutropenia and thrombocytopenia were common AEs in the placebo group.
Idelalisib + Rituximab in CLL patients [342]	Completed	Gilead Sciences	III, NCT01539512	220 patients were enrolled received Ritu ± Ide or placebo. Ide was given at 150 mg BID, Ritu was given at a dose of 375 mg/m^2^ IV, followed by 500 mg/m^2^ every 2 weeks for 4 doses and then every 4 weeks for 3 doses. The median PFS was N.R. in Ide arm (hazard ratio 0.15; *p* < 0.001). The median PFS in placebo arm was 5.5 months. PD occurred in 12 patients vs. 53 patients in the Ide group and placebo arm, respectively. Patients (*n* = 176) receiving Ide vs. placebo had ORR of 81% (95% CI: 71–88) vs. 13% (95% CI: 6–21); odds ratio, 29.92; *p* < 0.001). The OS of Ide vs. placebo arm was 92% vs. 80% (hazard ratio, 0.28; 95% CI: 0.09–0.86) at 12 months. Serious AEs were reported in 40% of patients receiving Ide + Ritu vs. 35% of patients receiving placebo and Ritu. Patients in Ide arm had a 50% or more reduction in lymphadenopathy vs. in the placebo group. In the Ide arm, severe gastrointestinal/skin disorders resulted in treatment discontinuations (*n* = 6) and infections/respiratory disorders led to *n* = 8 discontinuations in the placebo arm.	Common AEs were pyrexia, fatigue, nausea, chills and diarrhea in the Ide arm. Infusion-related reactions, fatigue, cough, nausea and dyspnea were common in the placebo treatment arm. Hepatic aminotransferase elevations was more frequently seen in patients receiving Ide + Ritu compared to those receiving placebo + Ritu. The most frequent serious effects in the two groups were pneumonia, pyrexia and febrile neutropenia.
Idelalisib + Rituximab (Ritu) in patients with CLL in UK (RETRO-idel) [343]	Completed	Gilead Sciences	N/A, NCT03582098	110 patients were enrolled. The median duration of Ide therapy was 12 months. The ORR was 88.2% (95% CI: 82.2–94.2) and median DoR 32.7 months (95% CI 19.7-N.R.). 7 (6.4%) patients had no documented response and response data were missing for 6 (5.4%) patients. Median OS was not reached (95% CI: 31.5-N.R.) and median PFS was 29.5 months (95% CI: 22.1-N.R.). There were 85 (77.2%) discontinuations, 54 (56.3%) due to toxicity, 18 (18.8%) due to PD, 11 (11.5%) by investigator discretion and 2 (2.1%) by subject decision. There were 11 (11.5%) deaths reported and Richter’s transformation was documented in 1 (0.9%) patient. 46 (41.8%) patients died during analysis.	Diarrhea, lower respiratory tract infection, neutropenia, rash, pneumonia and colitis.
Idelalisib + Obinutuzumab (Obi) in patient with R/R Waldenstrom’s Macroglobulinemia (RemodelWM3) [344]	Active, not recruiting	French Innovative Leukemia Organisation	II, NCT02962401	50 patients were enrolled. Ide was given orally at 150 mg BID + Obi IV (100 mg day 1, 900 mg day 2 then 1000 mg fixed dose on day 8, 15 of cycle 1 and every day 1 of cycles 2 to 6 in 28 days cycle). MYD88 mutations were found in 47 patients (96%). The ORR was 90% and the MRR was 76% (no CR, VGPR: 8%, PR: 68%, MR: 14%, SD: 8% and PD: 3%). The median PFS was 25.2 months. The 1-year and 2-year PFS were 90% (95% CI: 80, 100) and 70% (95% CI: 53, 93), respectively. The 1-year and 2-year OS were 98% (95% CI: 94, 100) and 85% (95% CI: 69, 100), respectively. 3 patients died due to AEs.	Common AEs were anemia, anemia, thombocytopenia, rapid evolution of monoclonal component, constitutional symptoms, hyperviscosity syndrome, thrombocytopenia, extramedullary disease and neuropathy.
Idelalisib + Ofatumumab (Ofa) in previously untreated CLL/SLL patients [345]	Active, not recruiting	Dana-Farber Cancer Institute with Gilead Sciences and GlaxoSmithKline	II, NCT02135133	24 patients were enrolled. Idea was administrated at 150 mg BID for first 56 days followed by 6 months in combination with Ofa. The initial dose of Ofa was 300 mg followed by subsequent doses of 900 mg. A decrease in peripheral blood regulatory T cells was seen in these patients. Patients with hepatic toxicities demonstrated high level of pro-inflammatory cytokine such as CCL-3 and CCL-4. Patients with and without hepatotoxicity had no significant difference in the other tested cytokines (CCL-2, CXCL-5 and VEGF) at baseline.	Hepatotoxicity, ALT/AST elevation and transaminitis.
Acalabrutinib (Acala) versus Idelalisib + Rituximab or Bendamustine + Rituximab in R/R CLL patients [346]	Active, not recruiting	Acerta Pharma BV	III, NCT02970318	310 patients (Acala = 155; Ide: *n* = 119; Ben: *n* = 36) were enrolled. 18 month PFS rates were 82% for Acala and 48% for Ide/Ben regimens, respectively. 18 month OS rate was 88% for both treatment regimens. ORR was 80% with Acala vs. 84% with Ide/Ben (ORR+ PR + lymphocytosis: 92% vs. 88%, respectively). AEs led to drug discontinuation in 16% of Acala, 56% of Ide and 17% of Ben treatment regimens, respectively.	Atrial fibrillation, hemorrhage and infection.
Tirabrutinib (Tira) with Idelalisib or Entospletinib (Ento) in previously treated CLL patients [347]	Active, not recruiting	Gilead Sciences	I, NCT02457598	53 patients were recruited in three cohorts: Tira cohort (*n* = 29), Tira + Ide cohort (*n* = 14), Tira + Ento cohort (*n* = 10). Ide was given at 50 mg BID or 100 mg QD and Tira 20–160 mg QD, Ento 200 mg or 400 mg QD and Tira 40–150 mg QD. All these doses were given in combination therapy except for Tira which was administrated at 80 mg QD in monotherapy. No DLTs were observed in CLL patients receiving either combination regimens; hence no MTD was identified in any cohort. Tira arm: ORR was 83%, mean (SD) time to response was 4.6 (1.3) months. The median PFS was not reached, median (range) follow-up time of PFS was 15.5 (0.0, 24.0) months. Tira + Ide arm: ORR was 93%. The median DoR was 27 months (95% CI: 15–27 months). The mean (SD) time to response was 5.5 (1.1) months. The median PFS was 32 months (95% CI: 8–32 months), CR was 7% and the median PFS was 34 months (range, 26–43). 2 patients had PD. Tira + Ento arm: ORR was 100%. The median DoR was not reached, and the mean (SD) time to response was 5.8 (0.7) months. The median PFS was not reached in this group and the median (range) follow-up time of PFS was 30.4 (24.7–33.2) months. CR was seen in 10% of patients in this group.	Most common AE was diarrhea, constipation, nausea, neutropenia and confusion.
**YY-20394/Linperlisib**					
Linperlisib (Lin) in patients with B-cell hematologic malignancies [348]	Unknown	Shanghai YingLi Pharmaceutical Co. Ltd.	I, NCT03757000	22 patients were enrolled and received Lin at dose range of 20–200 mg/day. All the patients completed cycle 1 safety observation without any observed DLTs. The ORR was 68% (4 patients with CR + 9 patients with PR) in *n* = 19 patients evaluated for response. The ORR in *n* = 7 patients with FL was 86% (3 patients had CR + 3 patients had PR). The DoR was 8 months in 4 evaluable patients. Observed PK parameters [AUC_(0–24h)_ and C_max_] exhibited dose proportionality with median T_max_ of 2 h.	Common AEs observed were enhanced LDH levels, pneumonia and hyperuricemia. Other hematologic AEs seen were neutropenia, lymphocythaemia, leukocytosis and leukopenia.
**INCB050465/Parsaclisib/IBI-376**					
Parsaclisib in R/R B-cell malignant patients (CITADEL-101) [349]	Active, not-recruiting	Incyte Corporation	I, NCT02018861	52 patients were enrolled: DLBCL (*n* = 14); FL (*n* = 10); MZL (*n* = 9); CLL (*n* = 6); MCL (*n* = 5), HL (*n* = 9), MCL (*n* = 5). The study was initiated with a single patient cohort treated with oral Parsaclisib 5 mg/day further extended to 3 + 3 design to study 10–45 mg of Parsaclisib. Based on PK/PD data, 20 and 30 mg daily doses of Parsaclisib were recommended. Median duration of therapy was 3.3 months, with no DLTs reported. 67% of patients discontinued their therapies due to various events such as disease progression in 31% of patients; AEs and dose interruption observed in 25% and 33% of patients, respectively. Only 4% had disease reduction. The ORs occurred at all doses except 5 mg daily dose.	Most common ≥ Gr 3 AEs: nausea, diarrhea, vomiting, neutropenia, lymphopenia, thrombocytopenia and anemia were observed. 40% of patients had serious AEs including colitis, diarrhea and hypotension. 1 patient had Gr 3 pneumonitis and none of patients showed PJP or Gr ≥ 2 ALT.
Parsaclisib in R/R DLBCL patients (CITADEL-202) [350]	Active, not recruiting	Incyte Corporation	II, NCT02998476	60 patients were enrolled in two groups: Group A; BTK naïve (*n* = 55) and Group B; BTK treated (*n* = 5). These patients were treated with oral Parsaclisib 20 mg daily upto 8 weeks followed by weekly dose. Group A; BTK naïve (*n* = 55): Observed ORR was 25.5% (14/55 patients; 8 CMR, 6 PMR); the median PFS was 2.2 months (95% CI: 2.0–4.1) and the median DoR was 4.5 months (95% CI: 2.1–5.1). Group B; BTK treated (*n* = 5): ORR was 20% (1/5 patients; 1 CMR). TEAEs led to treatment discontinuation in 7 patients (2 treatment-related), dose interruption in 20 patients (10 treatment-related) and dose reduction in 3 patients (all treatment-related).	Most TEAEs were rash, colitis, diarrhea, nausea, cough and pyrexia. Other were Gr 3/4 AST and ALT elevations; Gr 3/4 neutropenia and anemia. The most frequent serious TEAEs were pyrexia, general physical health deterioration and hypercalcemia.
**TGR-1202/Umbralisib/RP5264**					
Umbralisib with Ibrutinib in R/R B-cell lymphoma patients [351]	Completed	University of Nebraska in collaboration with NCI	II, NCT02874404	13 patients were enrolled. Patients received oral TGR-1202 daily on days 1–28 and Ibrutinib daily on days 9–28 of course 1 and days 1–28 of subsequent courses. CR was observed in 7.7% and PR in 23.1% of these patients.	Common AEs were nausea, diarrhea, abdominal pain, vomiting, dysphagia, fever, chills, upper respiratory infections, decreased appetite and cough.
Umbralisib in patients with R/R hematologic malignancies [352]	Completed	TG Therapeutics, Inc. with SCRI Development Innovations, LLC	I, NCT01767766	90 patients were enrolled and administrated with Umbralisib at initial dose of 50 mg and increased upto 1800 mg. The MTD was 1200 mg and RP2D was 800 mg. Umbralisib showed dose proportionality in plasma exposure and had an extended half-life of more than 100 h. 56 (62%) of the 90 enrolled patients had reductions in disease burden, 33 patients (37%) had an OR and 30 patients (33%) had a PR. 73 patients were eligible for inclusion in the modified intention-to-treat population for assessment of anti-tumor activity. Of these 73 patients, 53 (73%) had reductions in disease burden, including 33 (45%) patients with an OR of reductions of 50% or more, of which 3 (4%) had CR and 30 (41%) had a PR. In patients with R/R CLL, 17 (85%) of 20 patients had an OR with 10 (50%) achieving an OR, 7 (35%) achieving a PR with lymphocytosis and the remaining 3 (15%) had SD. Of 8 assessable patients with CLL, who had high-risk cytogenetic features, 6 (75%) had a response, of whom 2 (25%) had a PR with lymphocytosis and the remainder had a SD. In patients with FL, 9 (53%) of 17 patients achieved an OR, including 2 (12%) with CR; 15 with PR. In patients with DLBCL, 4/13 (31%) had an OR and 2 patients (15%) had SD. In patients with HL: 1 had CR, 4 had SD, 4 patients had PD. In patients with MZL: 1 had PR, 4 had SD; Waldenström’s macroglobulinaemia: 2 had SD; and MCL: 1 had PR, 4 had SD, and 1 had PD. The mean DoR was 13.4 months (95% CI: 7.7–19.1) in 16 patients in the CLL cohort, 6.4 months (4.5–17.3) in 4 patients in the DLBCL cohort, and 9.3 months (3.6–15.1) in 9 patients in the FL cohort. Median PFS was 24.0 months (95% CI: 7.4 months-N.R.) in 20 patients with CLL (7 patients had PD), and 16 months (9.2 months-N.R.) in 24 patients with iNHL (FL, Waldenström’s macroglobulinaemia and MZL among whom 11 disease progression events were recorded.	Pneumonia, lung infection, febrile neutropenia, colitis. Diarrhea, nausea and fatigue are TEAEs.
Ublituximab plus TGR-1202 +/− Ibrutinib or Bendamustine in patients with B-cell malignancies [353]	Completed	TG Therapeutics, Inc.	I, NCT02006485	A total 46 patients were enrolled: 24 in the dose-escalation cohort (*n* = 14 CLL/SLL; *n* = 14; B-cell NHL; *n* = 10) and 22 in the dose-expansion cohort (CLL or SLL; *n* = 9; B-cell NHL; *n* = 13). The MTD of Umbralisib was not reached. The RP2D was 800 mg Umbralisib QD plus Ibrutinib PO QD and 900 mg Ublituximab IV administered on days 1, 8, and 15 of cycle 1, day 1 of cycles 2–6, and on day 1 of cycles 9. 12.37 (84%) of 44 patients had OR (CR or PR). Among 18 patients with previously treated CLL, 8 (44%) achieved a CR and 10 (56%) achieved a PR. All eight evaluable patients with 17p deletion responded to treatment; 5 (63%) patients achieved a CR and 3 (38%) patients achieved a PR. Out of 9 CLL patients analyzed for minimal residual disease, 7 (78%) patients had no minimal residual disease, 3 (33%) achieved a CR, and 4 (44%) achieved a PR. The median DOR was 22.7 months (IQR: 8.6–25.9) in patients with CLL or SLL, 14.6 months (5.5–22.2) in patients with MCL, 20.0 months (7.3–23.8) in patients with MZL, and 23.4 months (13.8–24.3) in patients with FL. The median PFS in all patients was 38.2 months (95% CI: 21.3–N.R.). The median PFS was not reached (95% CI 24.5–N.E.) for patients with CLL or SLL, (10.2–N.E.) for patients with MCL, not reached (21.3–N.E.) for patients with MZL, 38.2 months (0.4–N.E.) for patients with FL and 1.3 months (0.5–N.E.) for patients with DLBCL.	Common AEs were diarrhea, fatigue, infusion-related reaction, dizziness, cough and nausea. Other were neutropenia and cellulitis. Gr 3 or 4 AEs were rash, pneumonia, atrial fibrillation, sepsis, abdominal pain, syncope, cellulitis, pneumonitis, headache, lung infection, skin infection, pleural effusion, pericardial infusion, upper gastrointestinal bleeding and weakness.
Umbralisib and Ibrutinib in patients with CLL or MCL patients [354]	Active, not recruiting	Dana-Farber Cancer Institute with TG Therapeutics, Inc.,The Leukemia and Lymphoma Society, Blood Cancer Research Partnership	I, NCT02268851	42/44 patients (CLL/MCL; *n* = 21 each) were enrolled. Umbralisib was given in a daily dose up to 800 mg. Ibrutinib was administrated orally once daily (420 mg for CLL or 560 mg for MCL) until disease progression or unacceptable toxicity was observed. No DLTs were seen and the MTD of was not obtained. The RP2D of Umbralisib was 800 mg daily dose when given in combination with Ibrutinib. For CLL patients, the ORR was 90% (19/21). The PR/PR-L rate was 62% (13/21), and the chronic lymphocytic leukaemia rate was 29% (6/21). The median time to best response was 2 months (range 1.4–38.1), and median time to CR was 18.4 months (range 4.7–37.5). For MCL, the ORR was 67% (14/21). The PR rate was 48% (10/21), and the CR rate was 19% (4/21). The median PFS and OS were not reached, and the 2-year PFS and OS were 90% and 95%, respectively. One of the patients in PR achieved radiographic CR but had residual marrow involvement that precluded reaching CR. The median time to best response was 2 months (range 1.7–18.3 months). Clinical benefit was observed in 3 patients who did not meet formal criteria for response, including 2 patients without baseline lymphadenopathy, one of whom had SD for 10 months. In MCL patients, the median PFS and OS were 10.5 months and 29.7 months, respectively. The 2-year PFS and OS in MCL patients were 49% and 58%, respectively.	Diarrhea, lipase elevation, atrial fibrillation, hypophosphataemia, adrenal insufficiency, transaminitis and infections.
Umbralisib in CLL patients [355]	Active, not recruiting	TG Therapeutics, Inc.	II, NCT02742090	40 patients were enrolled and received Umbralisib at 800 mg daily dose till until progression or toxicity observed. The median PFS was not reached with 90% of patients with a median follow up period of 6.5 months (range 1–15). 3 patients (7.5%) had dose reductions and were successfully re-challenged and 4 patients discontinued due to AEs.	Most AEs were arthralgia, rash, atrial fibrillation, diarrhea, bleeding, fatigue, alteration in AST/ALT level and weight loss. Other were pneumonia, pneumonitis, pancreatitis that led to treatment discontinuation.
**Tenalisib/RP6530**					
Tenalisib in T-cell lymphoma patients [356]	Completed	Rhizen Pharmaceuticals SA	I, NCT02567656	59 patients were enrolled. However, 58 patients received Tenalisib at a dose of 200 to 800 mg BID in an escalation phase (*n* = 19) and 800 mg BID in an expansion phase (*n* = 39). The MTD was achieved at 800 mg fasting dose. T_1/2_ of Tenalisib was 2.28 h. 16/35 patients (7 in dose escalation and 9 in dose expansion) responded to Tenalisib treatment. ORR was 45.7% (95% CI: 28.8–63.4) among responders. CR was reported in 3 (9%) patients while PR was noted in 13 (37%) patients. A total of 11 (31%) patients had SD while 8 (22.9%) had PD. Most of the patients responded at cycle 3 day 1. The response among PTCL and CTCL patients were almost similar [PTCL 46.7% (95% CI: 21.3, 73.4), CTCL 45% (95% CI: 23.1, 68.5)]. 3 (20%) PTCL patients had CR and 4 (26.7%) had PR while 9 CTCL patients had PR. The overall median DoR was 4.9 months (95% CI: 4.3, 12.0) with 6.5 months (95% CI: 2.9, 14.9) in PTCL and 3.8 months (95% CI: 2.3, 12.8) in CTCL patient. In 3 PTCL patients who had CR, the median DoR was 13.3 months (range 7.5-18.6). PTCL patients showed a reduction in CD30 levels from baseline to cycle 3 day 1. In CTCL patients, reductions were seen in plasma IL-31 and IL-32α levels at cycle 3 day 1 versus baseline and this reduction correlated with the responses seen.	Common TEAEs were fatigue and transaminase elevations.
**Taselisib/GDC-0032**					
Taselisib in combination with Docetaxel or Paclitaxel in HER2- locally recurrent or MBC or NSCLC patients [357]	Completed	Genentech, Inc.	Ib, NCT01862081	A total 80 patients (BC: 72; NSCLC: 7; BC/NSCLC: 1) were enrolled in the dose expansion and escalation arms. Patients (*n* = 21) received Taselisib + Docetaxel in 4 arms (Arms A, C, D, and E). Other group of patients (*n* = 59) received Taselisib + Paclitaxel in 2 arms (Arms B and F) with Taselisib at different dosing schedules. Patients in Arms A, C, D, and E were treated with IV Docetaxel (75 mg/m^2^) on day 1 of each 21 day cycle. Patients in Arms B and F received IV Paclitaxel (80 mg/m^2^) on days 1, 8, 15 and 22 of each 28 day cycle. In Taselisib + Docetaxel group: 2/4 patients had DLTs. The MTD was exceeded in Arm A with 3 mg Taselisib capsules daily once for 21 consecutive days plus 75 mg/m^2^ Docetaxel. There were no DLTs in Arms C, D and E. 7/20 patients (35.0%) in this group had PR but no CR. ORR was 35.0% (7/20) and CBR was 45.0% (9/20). The median DOR was 5.5 months (95% CI: 3.1–5.5) in *n* = 7 patients evaluated for response. The median PFS was 4.1 months (95% CI: 2.7–6.8) in *n* = 21 patients evaluated for response. In Taselisib + Paclitaxel group: A total of 4 DLTs observed; 2 each in Arms B and F, respectively. In Arm B, the MTD was not reached and an expansion cohort was opened. The MTD was exceeded at the 6 mg Taselisib in tablet for 5 days on/2 days off schedule plus 80 mg/m^2^ Paclitaxel and an Arm F expansion cohort was opened. Data from Arms B and F indicated that the cycle 1, day 15 pre-dose (C_max_) levels of Taselisib were 33.1 ng/mL (68.8 ng/mL) and 15.4 ng/mL (31.5 ng/mL), respectively. Taselisib showed moderate absorption and approximately proportional and linear increases in C_max_ concentration, AUC_1_ and AUC_(0–24)_ after a single dose and at steady state (following 15 days of daily dosing) with increasing dose levels from 2 to 6 mg capsule or tablet equivalent. 1/54 patients (1.9%) had a CR; 10/54 (18.5%) had PR. The ORR was 20.4% (11/54 patients) and CBR was 27.8% (15/54) for this group. The median DoR was 7.3 months (95% CI: 4.4–12.7). The median PFS was 4.1 months (95% CI: 3.0–7.1) in *n* = 59 patients.	Taselisib + Paclitaxel arm: Common TEAEs were diarrhea, nausea, fatigue and alopecia. Taselisib + Docetaxel arm: Common TEAEs were fatigue, diarrhea and alopecia, nausea and neutropenia.
Taselisib in previously treated PI3K+ NSCLC patients (Lung-MAP) [358]	Completed	Southwest Oncology Group in collaboration with NCI	II, NCT02785913	26 patients treated with Taselisib comprised the FEP; 21 patients comprised the PAP group. The study was closed due to futility at interim analysis with 1 responder in the PAP (5% RR, 95% CI: 0–24%). This patient had a DoR = 4.4 months. In the FEP, RR was 4% (95% CI: 0–20%) in all patients, SD in *n* = 16 patients with a disease control rate of 65% (95% CI: 47–84%). The median PFS and OS in the PAP group were 2.9 months (95% CI: 1.8–4.0 months) and 5.9 months (95% CI: 4.2–7.8 months), respectively. These numbers were similar in the FEP. The 1- and 2-year OS estimates were 23.8% and 17.9% in the PAP and 30.8% and 22.4% in the FEP. The study failed to meet its primary end point (response to Taselisib evaluated by RECIST 1.1 criteria) and was suspended after interim futility analysis in these patients.	2 possibly treatment-related deaths (1 respiratory failure and 1 cardiac arrest) were observed. 1 patient had Gr 4 (dyspnea, thrombocytopenia and pneumonitis) and 11 patients had Gr 3 AEs (hyperglycemia or diarrhea and lymphopenia).
Taselisib in combination with neoadjuvant Letrozole vs. placebo + Letrozole in post-menopausal women with ER+, HER2- early stage breast cancer (LORELEI) [359]	Completed	Genentech, Inc. in collaboration with SOLTI Breast Cancer Research Group, Breast International Group and Austrian Breast and Colorectal Cancer Group	II, NCT02273973	A total of 334 patients were enrolled in two groups: Letrozole (2.5 mg oral, daily) + placebo (5 days on, 2 days-off schedule) (*n* = 168) and Letrozole + Taselisib (4 mg oral) (Total *n* = 166). *PIK3CA* mutations were found in 152 (46%) patients, 73 (44%) of 166 randomly assigned to Taselisib and 79 (47%) of 168 assigned to the placebo group. Of the 181 (54%) patients without a detectable *PIK3CA* mutation, 92 (55%) were randomly assigned to Taselisib and 89 (53%) to placebo. 1 patient without known *PIK3C*A status was assigned to Taselisib group. The ORR at a median follow up of 4.9 months (IQR: 4.7–5.1 months) was measured in 66/168 (39%) patients in the placebo group vs. 83/166 (50%) in the Taselisib group and found as OR: 1.565 (95% CI: 1–2.38). For patients with *PIK3CA* mutation, 30/79 (38%) patients in the placebo group vs. 41/73 (56%) patients in the Taselisib group had an OR 2.03 (95% CI: 1.06–3.88). For patients with *PIK3CA* WT tumors, there was no significant difference observed in patients achieving objective response across both the groups. Ki67 decreased after 3 weeks of treatment in both groups. (1) Letrozole + placebo: 47% of patients had *PIK3CA* mutation. The decrease in Ki67 (marker of proliferation) value in patients with *PIK3CA* mutation in this group was −82 (95% CI: −78 to −87). Overall, 2%, 38% and 51% of patients had a CR, PR and SD, respectively. 3% of patients had PD. In the *PIK3CA*-mutant patients, 3%, 35%, 49% and 4% of patients had CR, PR, SD and PD, respectively. In *PIK3CA* WT patients in this group, 1%, 39%, 53% and 2% of patients had CR, PR, SD and PD, respectively. (2) Letrozole + Taselisib: 44% of patients had a *PIK3CA* mutation. More patients in this group harboring *PIK3CA* mutation had a PR. The decrease in Ki67 value in patients with *PIK3CA* mutation in this group was −85 (95% CI: −79 to −89). Overall, 5%, 45% and 40% of patients had a CR, PR and SD, respectively. 4% of these patients had PD. In the *PIK3CA*-mutant patients in this group, 7%, 49%, 38% and 1% of patients had CR, PR, SD and PD, respectively. In the *PIK3CA* WT patients in this group, 3%, 42%, 41% and 5% of patients had CR, PR, SD and PD, respectively. Taselisib increased the efficacy of Leterozole in these patients (*n* = 334) especially in patients with *PIK3CA* mutations. The combination was found to be safe with tolerable toxicity profile.	Common AEs in Taselisb group: gastrointestinal (diarrhea, nausea and stomatitis), fatigue, hyperglycemia, rash, arthralgia, and hot flush and infection. In the placebo group: vascular, gastro intestinal disorders, infection and infestations were common AEs.
Taselisib in combination with various anti-HER2 therapies in advanced HER2+ breast cancer patients [360]	Active, not recruiting	Dana-Farber Cancer Institute in collaboration with Genentech, Inc	Ib, NCT02390427	The combination of Taselisib (2 or 4 mg twice daily) and Trastuzumab emustine (T-DM1; 3.6 mg/kg thrice weekly) demonstrated preliminary efficacy and had an acceptable safety profile in these patients (*n* = 26). The median PFS was 7.6 months (95% CI: 2.9 months -NR). 4%, 29% and 50% of patients had a CR, PR and SD, respectively, in *n* = 24 evaluable patients.	Diarrhea, thrombocytopenia, vomiting, hyperglycemia, pneumonitis and elevated beta-glucan levels.
Taselisib in combination with Palbociclib (a CDK 4/6 inhibitor) in advanced solid tumors (PIPA) [361]	Active, not recruiting	Royal Marsden NHS Foundation Trust in collaboration with Institute of Cancer Research United Kingdom, Roche Pharma AG and Pfizer	Ib, NCT02389842	The combination was tested at the RP2D of Taselisib (2 mg) and Palbociclib (125 mg given 3 weeks on, 1 week off). Patients had a median of 4 prior treatments. There was modulation in pRb, pAKT, pGSK3β levels analyzed in the plasma and in tumor biopsies post-treatment. The combination was well tolerated, with promising anti-tumor activity and efficacy in these patients (*n* = 20).	Neutropenia, rash, mucositis, thrombocytopenia and increased AST/ALT. No Gr 4 or 5 AEs were observed.
Taselisib in combination with anti-androgen therapy, Enzalutamide (Enz) in AR+ metastatic TNBC patients [362]	Active, not recruiting	Vanderbilt-Ingram Cancer Center in collaboration with NCI, Translational Breast Cancer Research Consortium, Conquer Cancer Foundation and Genentech, Inc.	I/II, NCT02457910	Arm A: Patients received Taselisib (4 mg) orally once daily on days 1 to 28 and Enz (160 mg) PO QD on days 9 to 28 of cycle 1 and days 1 to 28 of subsequent cycles. Arm B: Patients received 160 mg Enz PO QD on days 1 to 28. Cycles were repeated after 28 days. The combination was safe, well tolerated and increased the CBR in these patients (*n* = 30). The CBR was 35.7% including 4 patients with SD and 1 patient with PR. The median PFS was 3.4 months with median PFS for ER+ patients (7.2 months) being substantially higher than that for TNBC patients (2.1 months). There was no difference in PFS across groups receiving drug combinations or Enz only. There was no difference in PFS or CBR with the *PIK3CA* status. However, better CBR was reported in TNBC patients with LAR subtype tumors versus other subtypes (75% vs. 12.5%; *p* = 0.06) along with better median PFS of 4.6 months vs. other subtypes (PFS = 2 months). LAR subtype tumors demonstrated a decrease in proliferation and AR target gene expression post-treatment with combination. Mutational landscape assessment of tumors suggested that LAR tumors were enriched *GATA3* and *FOXA1* genes. *TP53* mutations were frequent across all subtypes. In contrast, non-LAR tumors were enriched in *RB1*, *S82X*, *R467X* (deleterious mutations in cell-cycle) as well as *ESCO1,BRCA1*, *BRACA2*, *BAP1* and *FANCE* (DNA repair genes) along with activating mutations in MAPK pathway and growth factor receptors. Two potential oncogenic gene fusions *FGFR2-TACC2* and *FGFR2-TAOK1* were identified which represent a mechanism by which LAR tumors activated the PI3K pathway. Pathways associated with complement and innate immunity were augmented post-treatment. Treatment with Taselisib and Enz specifically increased T-cell and NK cell markers. AR splice variants might contribute to Enz resistance. The MTD was not reached and the trial was terminated early. The RP2D was achieved at 4 mg daily dose of Taselisib with 160 mg Enz/day. 13 patients were enrolled in the phase I and 17 patients (Enz; *n* = 5 and Enz + Taselisib; *n* = 12) in phase II trial.	Hyperglycemia, rash, increased AST/ALT, anemia, neutropenia, fever, fatigue, nausea, vomiting and pruritus.
Taselisib + Fulvestrant vs. placebo + Fulvestrant in postmenopausal women with ER+ HER2- locally advanced or MBC patients with disease recurrence or progression during or after aromatase inhibitor therapy (SANDPIPER) [363]	Active, not recruiting	Hoffmann-La Roche	III, NCT02340221	The combination was well tolerated and had a favorable safety profile in these patients (*n* = 516). The two groups were: (1) Taselisib (4 mg oral daily) + Fulvestrant (500 mg) (*n* = 340): The median PFS was 7.4 months. The median DoR was 8.7 months. The ORR and CBR was 28% and 51.5%, respectively. (2) placebo+ Fulvestrant (*n* = 176): The median PFS was 5.4 months. The median DoR was 7.2 months. The ORR rate and CBR were 11.9% and 37.3%, respectively. According to BICR-PFS, the Taselisib + Fulvestrant group had a significantly improved PFS vs. the placebo+ Fulvestrant group (hazard ratio: 0.7; *p* = 0.0037). The OS was not reached during the preliminary analysis. The Taselisib + Fulvestrant group had a better ORR compared to other group (*p* = 0.0002).	Diarrhea, colitis, hyperglycemia and stomatitis.
**AZD8186**					
AZD8186 as monotherapy or with combination therapy in CRPC, NSCLC and TNBC patients [364]	Completed	Astrazeneca	I, NCT01884285	A total of 52 CRPC patients were enrolled and treated with AZD8186 (*n* = 39) or in combination with prednisone (AAP) (*n* = 13). Prior treatment status: AAP (*n* = 14), Enzalutamide (enza, *n* = 10), both (*n* = 21) or AAP/enza- naive (*n* = 7). Among subjects with RECIST-measurable disease, 1 had a confirmed PR, 10 had SD, 9 had PD. 9 (17%) patients experienced a reduction in PSA >30%.	Diarrhea, nausea and colitis. 2 (4%) patients had Gr 4 AEs (thrombocytopenia and hypokalaemia); no Gr 5 AEs were observed.
**IPI-145/Duvelisib/INK1197**					
Duvelisib with Rituximab or Bendamustine/Rituximab in patients with hematologic malignancies [365]	Completed	SCRI Development Innovations, LLC with Infinity Pharmaceuticals, Inc	I, NCT01871675	A total of 46 patients were enrolled, consisting of *n* = 29 NHL patients and *n* = 17 CLL patients, of which 10 patients (Arm 1, 7; Arm 2, 3) were enrolled in the dose-escalation portion and 36 (Arm 1, 20; Arm 2, 16) in the dose-expansion portion. In the escalation study, patients received Rituximab (375 mg/m^2^ IV, day 1 once weekly 4 weeks per cycle, 2 cycles) + Duvelisib (50 mg PO BID until intolerable toxicity) in Arm 1. In Arm 2: patients received Bendamustine (90 mg/m^2^ IV, days 1 and 2 upto 6 cycles) + Rituximab (375 mg/m^2^ IV, day 1 for upto 6 cycles). No DLTs were observed. The MTD was not reached in the escalation study so Duvelisib concentration was identified to be 25 mg PO BID for expansion study. In Arm 1: 23 patients evaluated for response (NHL = 14 and CLL = 9). 18/23 patients had ORR 78.3% which includes 2 patients with DLBCL, 8/9 (88.9%) CLL patients had a PR and 1 patient (11%) had a SD. The ORR in 10/14 NHL patients was 71.4%, 3/14 patients (21.4%) had a CR, 7/14 (50%) had a PR, 3/14 patients (21.4%) had SD and 1 patient 7.1% was not evaluable. In Arm 2 (NHL = 12, CLL = 4): ORR achieved was in 10/16 (62.5%) patients with 2 DLBCL patient. In CLL patients (*n* = 4), 3/4 patients (75%) had an ORR, 1 patient (25.0%) had a CR and 2 patients (50%) had a PR. The remaining patients (25%) were not evaluable. In NHL patient group (*n* = 12); 7/12 patients (58.3%) had an ORR, 2/12 patients (16.6 %) had a CR, 5/12 patients (41.7%) had a PR, 1/12 (8.3%) patients had a SD. 2 patients each (16.7%) had a best response of disease progression or were not evaluable. The overall median PFS was 13.7 months (95% CI: 7.4-N.E.) for all patients. The median PFS for CLL patients was 22.1 months in Arm 1 (95% CI: 13.7–N.E.) but was not reached in Arm 2. The median PFS for NHL patients was10.7 months (95% CI: 4.5–N.E.) in Arm 1 and 5.3 months (95% CI: 1.2–N.E.) in Arm 2. The median DoR were 15.1 months for CLL patients in Arm 1, 16.9 for CLL patients in Arm 2, 5.0 months for NHL patients in Arm 1 and 6.2 for NHL patients in Arm 2. Median OS was reached for both NHL/CLL patients in Arm 1 or for CLL patients in Arm 2. Median OS for NHL patients in Arm 2 was 9.1 2.6- N.E.	Neutropenia, ALT/AST increase, anemia, thrombocytopenia, diarrhea and pneumonia.
Duvelisib with FCR in untreated young CLL patients [366]	Active, not recruiting	Dana-Farber Cancer Institute with Verastem, Inc	N/A, NCT02158091	N/A	Anemia, diarrhea, nausea, fatigue, vomiting and gastric disorder.
Duvelisib in iNHL patients (DYNAMO) [367]	Active, not recruiting	Verastem, Inc	II, NCT01882803	129 patients with (FL: *n* = 83, SLL: *n* = 28 and MZL: *n* = 18) were enrolled and the achieved ORR was 47.3% (95% CI: 38% to 56%) with respective ORR in SLL: 67.9%; FL: 42.2%, MZL: 38.9%. The median PFS was 9.5 months (95% CI: 8.1 to 11.8 months). The median DoR was 10 months (95% CI: 6.5–10.5 months). Median TTR was 1.87 months (range, 1.4–11.7 months) with 59% and 84% of patients responding by 2 and 4 months, respectively. Median OS was 28.9 months (95% CI: 21.4-NR) and OS at 1 year was estimated at 77%. Median duration of treatment exposure was 6.7 months (range, 0.4 to 45.5 months).	TEAEs were diarrhea, nausea, neutropenia, fatigue and cough. Gr 3 or higher TEAEs seen were neutropenia, anemia, diarrhea and thrombocytopenia.
Duvelisib versus Ofatumumab in R/R CLL/SLL patients (DUO) [368]	Active, not recruiting	Verastem, Inc.	III, NCT02004522	160 patients received Duvelisib 25 mg BID or 159 patients received Ofatumumab. With a median follow up of 22.4 months, the median PFS in the Duvelisib vs. Ofatumumab arms for all patients was 13.3 months vs. 9.9 months; hazard ratio = 0.52. PFS was also extended with Duvelisib in multiple predefined CLL/SLL subgroups examined, including patients with high-risk cytogenetic markers. The median PFS in patients with del(17p)/TP53 mutations was 13.8 months with Duvelisib and 9.5 months with Ofatumumab (hazard ratio = 0.41). The ORR was found to be significantly higher with Duvelisib (74% vs. 45%; *p* < 0.0001) regardless of del (17p) status. In the Duvelisib arm, most patients had PR (72.5%) except 2 patients: 1 patient achieved a CR (0.6%) and 1 patient had a PRwL (0.6%). In Ofatumumab arm, most patients had PRs (44.7%) with a CR (0.6%) in 1 patient. Median OS was not reached on either treatment arms with a 12 month probability of survival of 86% (hazard ratio = 0.99; 95% CI: 0.65–1.50) for both treatments arms.	Common AEs seen in patients on Duvelisib treatment were diarrhea, neutropenia, pyrexia, nausea, anemia and cough. AEs observed in patients treated with Ofatumumab were neutropenia and infusion reactions.
Eganelisib/IPI-549 [369]	Active, not recruiting	Infinity Pharmaceuticals, Inc.	I, NCT02637531	31 patients were enrolled in the 6 + 6 design received IPI-549 at 20, 30, and 40 mg QD plus nivolumab 240 mg Q2W. The IPI-549 PK/PD profile was unaffected by nivolumab administration. The MTD was not reached. 2/30 patients demonstrated PR at 8 weeks assessment. The RP2D was determined to be IPI-549 40 mg QD plus nivolumab 240 mg Q2W. Upregulation of IFNγ-responsive factors, such as PD-L1 and CXCL9/10 and dose-dependent re-invigoration/proliferation of exhausted PD1+CD8+CD45RA- T cells was observed as evidenced by Ki67 increase. 2 DLTs at IPI-549 30 mg (Gr 3 rash) and 40 mg QD (Gr 3 rash and Gr 3 ALT/AST increase) were observed.	Common TEAEs were rash, pruritus, pyrexia nausea, anemia, increase ALT/AST.

## 5. Dual PI3K/mTOR Inhibitors

Dual PI3K/mTOR inhibitors target both PI3K and mTOR signaling. To date, none of these drugs have received FDA approval for treatment of any cancer. The detailed clinical outcomes of completed and active not-recruiting studies with adverse effects are summarized in Table 3.

### 5.1. Dactolisib/BEZ-235/NVP-BEZ235

Dactolisib/BEZ235/NVPBEZ235 is a dual pan-class I PI3K inhibitor targeting p110α/ß/γ/δ as well as mTOR kinase inhibitor with IC50 values of 4 nM/5 nM/7 nM/75 nM and 20.7 nM for p110α/γ/δ/β and mTOR, respectively. Dactolisib, developed by Novartis Pharmaceuticals, is known to inhibit both mTORC1 and mTORC2 activity [370]. A phase Ib study sponsored by the Vanderbilt-Ingram Cancer Center investigated the safety and efficacy of Dactolisib with BKM120 in combination with endocrine therapy in post-menopausal patients with HR+ metastatic breast cancer [138]. Another phase I/II study investigated Dactolisib in patients with advanced breast cancer [371]. A phase I trial tested Dactolisib with RAD001/Everolimus in patients with advanced solid tumors without any significant outcomes [372]. Another such phase Ib study evaluated the safety, pharmacokinetics, and pharmacodynamics of Dactolisib with MEK162 in patients with selected advanced solid tumors [373]. A phase Ib/II study of Dactolisib with Trastuzumab versus Lapatinib and Capecitabine was completed in patients with HER2+ locally advanced or metastatic breast cancer who failed on prior Trastuzumab treatment. The trial aimed to find the MTD and the RP2D of this drug combination [374]. Another phase I trial investigated the MTD and RDE of Abiraterone Acetate in combination with Dactolosib and Abiraterone Acetate with BKM120 in CRPC patients [375]. However, despite all these studies, Dactolisib was removed from clinical trials in 2015.

### 5.2. Apitolisib/GDC-0980/RG7422

Apitolisib/GDC-0980/RG7422 is a potent, class I PI3K inhibitor targeting α/β/δ/γ isoforms with IC50 values of 5 nM/27 nM/7 nM/14 nM, respectively, in cell-free assays. Apitolisib also targets mTOR signaling. Genentech Inc. sponsored several phase I/II studies using Apitolisib. A phase Ib, dose-escalation study evaluated the safety, tolerability and PK properties of Apitolisib with Paclitaxel with or without Bevacizumab in patients with locally advanced breast cancer [376]. A phase Ib study assessed the safety and PK properties of Apitolisib in combination with Fluoropyrimidine, Oxaliplatin, and Bevacizumab in patients with advanced solid tumors [377]. A phase I study evaluated Apitolisib with either Paclitaxel and Carboplatin (with or without Bevacizumab) or Pemetrexed and Cisplatin in patients with locally advanced or metastatic solid tumors [378]. Apitolisib was evaluated as a monotherapy in patients with R/R solid tumors or with NHL. The study was sponsored by Genentech Inc. and aimed to assess the safety, tolerability, and pharmacokinetics of escalating doses of Apitolisib [379].

### 5.3. Gedatolisib/PF-05212384/PKI-587

Gedatolisib/PF-05212384/PKI-587, developed by Pfizer, is a highly potent dual inhibitor of PI3Kα/γ and mTOR with IC50 values of 0.4 nM, 5.4 nM and 1.6 nM, respectively, in cell-free assays [380]. Details of clinical studies with specific clinical outcomes are summarized in Table 3.

### 5.4. SF1126

SF1126, brought into market by SignalRx, is a small-molecule conjugate that selectively inhibits all PI3K class IA isoforms and other members of the PI3K superfamily, including DNA-PK and mTOR. It has been tested in a phase I study in patients with advanced solid tumors and B-cell malignancies [381]. A phase II study sponsored by Semafore Pharmaceuticals in collaboration with SignalRX Pharmaceuticals, Inc. evaluated the safety and tolerability of SF1126 by assessing the DLTs, MTD, and RP2D in patients with advanced or metastatic tumors [382]. A current non-recruiting phase I trial is studying SF1126 with immunotherapy (Nivolumab) in patients with advanced or metastatic hepatocellular carcinoma and Child-Pugh A-B7 cirrhosis [383].

### 5.5. Omipalisib/GSK458/GSK2126458

Omipalisib/GSK458/GSK2126458 is a reversible, selective, pan-PI3K ATP-competitive inhibitor with a Ki for the catalytic p110α subunit in the subnanomolar range that targets the mTOR pathway. The first-in-human phase I study tested Omipalisib in patients with advanced solid tumors. The study was sponsored by GlaxoSmithkline [384]. A phase I open-label, dose-escalation study evaluated the safety, efficacy, and PK of Omipalisib in patients with solid tumors or lymphoma [385].

### 5.6. Samotolisib/LY3023414

Samotolisib/LY3023414 is a dual inhibitor of PI3Kα and mTOR with potential anti-neoplastic activity [386]. A phase I trial evaluated the safety of Samotolisib in patients with advanced solid tumors [387]. A phase II study is testing the effectiveness of Samotolisib in patients with endometrial cancer [388]. Two phase I studies sponsored by Eli Lilly and Company evaluated Samotolisib in combination with chemotherapies or other targeted therapies in patients with advanced cancer or lung cancer [389,390]. In addition, a phase II study tested Samotolisib as monotherapy or in combination with other agents versus standard of care (Gemcitabine or Capecitabine) in patients with metastatic pancreatic ductal adenocarcinoma (PDAC) [391]. A phase I study is currently exploring Samotolisib with a CDK4/6 inhibitor LY2835219 (Abemaciclib) in MBC patients. The study is also sponsored by Eli Lilly and Company and aims to investigate the safety of this drug combination [392].

### 5.7. Bimiralisib/PQR309

Bimiralisib/PQR309 is a potent, oral bioavailable, pan-class I PI3K inhibitor that also targets mTORC1 and mTORC2. A phase II study sponsored by PIQUR Therapeutics AG aimed to assess the safety, tolerability, and the RP2D of Bimiralisib in patients with R/R lymphoma [393]. There are two other phase II studies that are completed in patients with R/R primary central nervous system lymphoma (PCNSL) and advanced lymphoma and HNSCC harboring *NOTCH1* loss of function mutations [394,395,396]. Another phase I/II study evaluated the safety and pharmacokentic properties of Bimiralisib in combination with chemotherapy, Eribulin in patients with locally advanced or metastatic HER2-negative cancer in the escalation part and in TNBC patients in the expansion part of the study [397].

### 5.8. Paxalisib/GDC-0084/RG7666

Paxalisib/GDC-0084/RG7666, developed by PIQUR therapeutics, is a potent dual inhibitor of PI3K and mTOR signaling. A phase I study is investigating Paxalisib in pediatric patients with newly diagnosed diffuse intrinsic pontine glioma or diffuse midline gliomas. The study is sponsored by St. Jude Children’s Research Hospital in collaboration with Kazia Therapeutics Ltd., Sydney, Australia which aims to determine the safety, MTD, RP2D, and the pharmacokinetic properties of Paxalisib in the pediatric population [398]. A phase II study is also investigating the efficacy and the pharmacokinetic properties of Paxalisib in glioblastoma patients [399].

### 5.9. XL765/Voxtalisib/SAR245409

Voxtalisib/SAR245409/XL765 is a dual inhibitor of PI3K/mTOR which targets the PI3K p110γ, DNA-PK and mTOR pathways. This drug was introduced into market by Sanofi. Sanofi sponsored a phase I/II study testing the safety and tolerability of Voxtalisib as a monotherapy or in combination regimen in patients who were benefiting from this treatment [400]. A phase II study evaluated the efficacy of Voxtalisib with MEK1/2 inhibitor, Pimasertib in patients with ovarian cancer [401]. Most of the studies using this drug are completed and the details of the clinical outcomes with adverse effects are listed in Table 3.

**Table 3 ijms-22-03464-t003:** Summary of trials, outcomes and adverse events associated with dual PI3K/mTOR inhibitors in various phases of clinical studies.

Treatment	Status	Sponsor	Phase and NCT	Clinical Outcomes	Adverse Effects
**Dactolisib/BEZ-235/NVP-BEZ235**					
Dactolisib in patients with advanced solid tumors [402]	Completed	SCRI Development Innovations, LLC in collaboration with Novartis Pharmaceuticals	I, NCT01343498	Dactolisib was well tolerated but demonstrated limited clinical response (*n* = 12). The MTD and R2PD were identified at 300 mg BID. PK parameters (day 28): The AUC_(0–24)_ was 10028.3 ng × h/mL (range: 3228.8–49640 ng × h/mL). The C_max_ and T_max_ were 655.6 ng/mL (range: 232–2700 ng/mL) and 2 h (range 0–4 h), respectively. There was a consistent increase in C_max_ and AUC with BID versus QD dosing schedule. At MTD, the median decrease in the standardized uptake value for ^18^F-FDG was 42% (rang: 6–67%). No PR and CR were reported. 45% of patients demonstrated SD.	Diarrhea, mucositis, nausea, hyperglycemia, anorexia and thrombocytopenia.
Dactolisib in adult Japanese patients with advanced solid tumors [403]	Completed	Novartis Pharmaceuticals	I, NCT01195376	35 Japanese patients were enrolled and received at least one dose of Dactolisib with either a QD (*n* = 27) or BID (*n* = 8) dosing schedule. The MTD was not reached. 2 DLTs including allergic reaction and thrombocytopenia were observed at 1200 and 1400 mg QD, respectively, while liver dysfunction was reported as a DLT at 400 mg BID. The maximum clinically tolerable dose was 1200 mg and the R2PD was 1000 mg QD. The median duration of exposure for QD and BID dosing was 56 days (range: 2–280 days) and 43.5 days (range: 21–115 days), respectively. As per RECIST v1.0 criteria, no CRs and PRs were observed. 14/27 (51.9%) patients evaluated for response demonstrated SD. C_max_ and AUC increased in a dose-dependent manner. In patients treated with 400 mg QD Dactolisib, a 50% reduction in pS6 levels was observed.	Diarrhea, decreased appetite, nausea, stomatitis, vomiting, fatigue, liver dysfunction and thrombocytopenia.
Dactolisib in Everolimus resistant patients with advanced pancreatic neuroendocrine tumors [404]	Completed	Novartis Pharmaceuticals	II, NCT01658436	Dactolisib was poorly tolerated in these patients when treated using a 400 and 300 mg BID dosing schedule (*n* = 31). Best response at 16 weeks was SD in 16 (51.6%) patients, PD in 9 (29.0%) patients and unknown in six (19.4%) patients. The estimated 16 week PFS rate was 51.6% (90% CI: 35.7–67.3%). The study did not proceed to stage II of the trial. The efficacy of Dactolisib was limited due to high intra- and inter-patient PK variability. The drug demonstrated a challenging toxicity profile.	Diarrhea, nausea, hyperglycemia, stomatitis and vomiting.
**Apitolisib/GDC-0980/RG7422**					
Apitolisib in patients with refractory solid tumors or NHL [405]	Completed	Genentech, Inc.	I, NCT00854152	Patients (*n* = 120) received 2–70 mg of Apitolisib on days 1–21 or 1–28 of 28 day cycles. Apitolisib demonstrated durable anti-tumor activity with a narrow therapeutic window and dose-proportional PK, PD effects (>90% suppression of the surrogate biomarker, platelet pAKT levels and decreased FDG-PET tumor uptake) with target modulation observed in patients treated with >16 mg dose. Stage I of the trial involved dose escalation to estimate MTD and stage II was dose expansion to establish the R2PD. The R2PD and MTD, respectively, were identified at 40 mg QD on a 28/28 dosing schedule except for malignant pleural mesothelioma (MPM) patients and 50 mg QD on a 21/28 dosing schedule. DLTs observed were Gr 4 fasting hyperglycemia at 40 mg (21/28-schedule) and Gr 3 maculopapular rash and Gr 3 fasting hyperglycemia at 70 mg (21/28-schedule). Apitolisib was rapidly absorbed and mean plasma concentration peaked after 1–2 h. The mean terminal elimination T_1/2_ was 11.3 h (range: 3.26–45.4 h). The drug accumulation was very low. The AUC showed dose proportionality. According to RECIST criteria, in stage I, 4/56 patients demonstrated PR and 2/56 patients demonstrated CR. 77% of patients demonstrated SD. Patients demonstrated tumor shrinkage and no *PIK3CA* mutations or loss of *PTEN* was identified. In stage II, 5/64 patients demonstrated PR and the median time for study was 2.9 months (range: 1–21.2 months). SD with tumor regression was observed in 20 patients treated with 40 mg drug.	Thrombocytopenia, neutropenia, hyperglycemia, hypercholesterolemia, increased AST/ALT, pneumonitis, rash, fatigue and mucosal inflammation.
Apitolisib in patients with metastatic renal cell carcinoma who progressed on or after VEGF-targeted therapy [406]	Completed	Genentech, Inc.	II, NCT01442090	Patients (*n* = 85) were randomized to receive Apitolisib at 40 mg QD or Everolimus 10 mg QD. The median PFS of Apitolisib versus Everolimus was 3.7 versus 6.1 months; hazard ratio, 2.12 (95% CI: 1.23–3.6). The ORR was 7.1% for Apitolisib versus 11.6% for Everolimus. Apitolisib PK profile indicated a direct relationship between exposure and adverse effects including rash and hyperglycemia. Retrospective biomarker analyses indicated a relationship between *VHL* mutation status and clinical outcome with Everolimus but not with Apitolisib dosing. High HIF-1α protein expression was associated with better outcome in both treatment arms. Apitolisib associated with higher-grade hyperglycemia and rash was less effective than Everolimus in this trial. This could be due to full blockade of PI3K/mTOR signaling which resulted in multiple on-target adverse events.	-
Apitolisib in patients with recurrent/persistent endometrial carcinoma (MAGGIE study) [407]	Completed	Genentech, Inc.	II, NCT01455493	56 patients including 23% of patients with well-controlled diabetes received oral Apitolisib at a dose of 40 mg QD during 28 day cycles until disease progression or intolerable toxicity occurred. The median duration of treatment was 57 days (range: 4–500 days). The six month PFS and median PFS rates were 20% (95% CI: 7–33%) and 3.5 months (95% CI: 2.7–3.7 months), respectively. The median OS was 15.7 months (95% CI: 9.2–17 months). The ORR was 6%. 3.6% and 1.8% of patients demonstrated CR and PR, respectively. 67% of patients had atleast 1 alteration in the PI3K pathway. Complete loss of *PTEN* and missense mutations in *PIK3CA* were found in 11% and 28% of patients, respectively. 24% of patients demonstrated concomitant PI3K pathway activating alterations. 5 of the patients who responded to the treatment had alterations in either *PIK3CA*, *PTEN*, *AKT1* or *ERBB2* genes. The anti-tumor activity observedat 40 mg QD Apitolisib was limited by tolerability mostly in diabetic patients.	Hyperglycemia, rash, fatigue, nausea, stomatitis, decreased appetite, vomiting, colitis and pneumonitis
**Gedatolisib/PF-05212384/PKI-587**					
Gedatolisib in combination with Cisplatin and other anti-tumor agents in TNBC patients [408]	Completed	Pfizer	I, NCT01920061	Gedatolisib could be safely administered in combination with Docetaxel, Cisplatin or Dacomitinib and had a manageable toxicity profile in these patients (*n* = 52). MTD could not be determined. The 3 study groups were: Arm A: Gedatoilisib + Docetaxel Arm B: Gedatolisib + Cisplatin Arm C: Gedatolisib + Dacomitinib No DLTs were seen in arms A or B; in arm C, DLTs observed included Gr 3 mucositis, pneumonitis, rash and Gr 2 fatigue (<75% of planned dose received).	Mucositis, nausea, neutropenia, decreased appetite, diarrhea, dermatitis acneiform, rash, neutropenia, vomiting and hypomagnesaemia.
Gedatolisib in combination with Paclitaxel and Carboplatin in patients with advanced solid tumors (IOSI-NDU-001) [409]	Completed	Oncology Institute of Southern Switzerland	I, NCT02069158	Gedatolisib demonstrated an acceptable safety profile when administrated weekly in combination with Carboplatin and Paclitaxel in these patients (*n* = 17). The RP2D for Gedatolisib was 110 mg/m^2^ IV and Paclitaxel 80 mg/m^2^ IV. 65% of patients achieved an objective response. 47% and 18% of patients achieved a PR and CR, respectively. 17% of patients demonstrated SD. DLTs were observed in 4/16 evaluable patients which included two (Gr 2 and Gr 3 neutropenia) at 110 mg/m^2^, two (Gr 2 and Gr 3 mucositis) at 130 mg/m^2^ and no DLT at 110 mg/m^2^. Anti-tumor activity was observed in 8 out of 10 patients with clear cell ovarian cancer.	Neutropenia, fatigue, thrombocytopenia, hypokalemia, peripheral neuropathy, hypomagnesemia, colitis and aortic intramural hematoma.
Gedatolisib in patients with advanced solid tumors [410]	Completed	Pfizer	I, NCT00940498	Gedatolisib had a manageable safety profile with an anti-tumor activity in these patients (*n* = 78). The MTD and RP2D for Gedatolisib was 154 mg (administered via IV route once weekly). Majority of the patients had received surgery and radiotherapy along with systemic anti-neoplatic treatments. Heavily pretreated patient population demonstrated anti-tumor activity. Mean plasma Gedatolisib concentration on single-dose administration declined rapidly in 4–24 h and more slowly for next 25–168 h. C_max_ and AUC increased proportionally with increasing dose of Gedatolisib. On multiple Gedatolisib administration, a biphasic concentration-time curve with a slower concentration decline in first 24 h for Gedatolisib was observed. The mean T_1/2_ values was identified at 33–41 h across all dose levels. T_1/2_ was ~36 h in the MTD group. The objective tumor response rate was 2.6%. The CBR was 13%. 10.4% of patients had an SD lasting for more than 6 months. Paired tumor biopsies indicated that Gedatolisib inhibited downstream effectors of the PI3K pathway. For patients treated at MTD, there was a 10.6% reduction in pAKT Ser473 levels as compared to baseline.	Stomatitis, muscosal inflammation, vomiting, fatigue, hyperglycemia, nausea, vomiting, constipation, increased AST/ALT, asthenia and decreased appetite.
Gedatolisb in combination with PTK7-ADC (Wnt pathway inhibitor) in metastatic TNBC patients [411]	Completed	Indiana University	I, NCT03243331	Patients with metastatic TNBC or low estrogen expressing breast cancer were enrolled. The primary objective of the trial was to evaluate the safety of Gedatolisib (administered weekly at 110 mg or 180 mg via IV route) in combination with PTK7-ADC (administered every 3 weeks at 1.4 mg/kg or 2.8 mg/kg dose) and the secondary objective was to determine the efficacy of this drug combination assessed by ORR, CBR and PFS.	-
**Samotolisib/LY3023414**					
Samotolisib in Japanese patients with advanced cancer [412]	Completed	Eli Lilly and Company	I, NCT02536586	Samotolisib upto 200 mg BID was safe and tolerable in the Japanese patients (*n* = 12). No DLTs were seen in 3 patients enrolled on the 150 mg dosing schedule. However, 2/9 patients who received 200 mg Samotolisib BID experienced DLTs (Gr 3 stomatitis). Samotolisib was rapidly absorbed and eliminated. The C_max_ and T_max_ at day 15 for group treated with 200 mg Samotolisib (*n* = 7) was 1010 ng/mL and 1.97 h (range: 0.97–2.95 h), respectively. The T_1/2_ for this cohort was 1.83 h (range: 1.47–2.64 h). The AUC_(0–tlast)_ and AUC_(0–∞)_ was 4100 ng × h/mL and 4210 ng × h/mL, respectively. According to RECIST v1.1 criteria, no CR and PR were observed. The DCR was 55.6% and 5 patients had best overall response. Samotolisib increased fasting glucose levels and C-peptide post-administration.	Stomatitis, nausea, diarrhea, decreased platelet count, decreased appetite, hyperglycemia, hypophosphatemia, anemia, vomiting and rash.
Samotolisib in combination with Enzalutamide in men with mCRPC after progression on Abiraterone [413]	Completed	Eli Lilly and Company	II, NCT02407054	The combination of Samotolisib (200 mg BID) and Enzalutamide (160 mg QD) was tolerable and had a clinically manageable safety profile. In the two cohorts: Samotolisib + Enzalutamide (*n* = 65): The median PCWG2-PFS was 3.7 months. The PFS for AR-V7-positive and -negative patients was 5.5 months and 13.2 months, respectively. 20% of patients had more than a 50% reduction in PSA. Placebo + Enzalutamine (*n* = 64): The median PCWG2-PFS was 2.96 months (hazard ratio: 0.66; 95% CI: 0.43–0.99; *p* = 0.02085). The PFS for AR-V7-positive and -negative patients was 3.6 months (hazard ratio: 0.52; 95% CI:0.28–0.95; *p* = 0.028) and 5.3 months (hazard ratio: 0.99; 95% CI: 0.27–3.63; *p* = 0.991), respectively. 25% of patients had more than a 50% reduction in PSA.	Diarrhea, nausea and fatigue.
Samotolisib in patients with advanced cancer [414]	Active, not recruiting	Eli Lilly and Company	I, NCT01655225	Samotolisib (200 mg BID) was well tolerated and had a manageable safety profile in patients with advanced mesothelioma (*n* = 42). The median duration of treatment was 11.2 weeks (range: 1.1–53 weeks). The median PFS was 2.83 months (95% CI: 2.53–3.98 months). According to RECIST criteria, the ORR and DCR were 2.4% and 43%, respectively. 41% of patients exhibited SD. The mean apparent clearance and volume of distribution were 71.2 L/h and 159 L, respectively, which led to a short T_1/2_ of 1.55 h. Alterations in *BAP1* (11/19 patients), *NF2* (5/19 patients) and *SETD2* (5/19 patients) genes were observed. The most common reason for study discontinuation was PD (48%).	Fatigue, nausea, vomiting, hyperglycemia, rash, decreased appetite and diarrhea.
**Bimiralisib/PQR309**					
Bimiralisib in patients with advanced solid tumors [415]	Completed	PIQUR Therapeutics AG in collaboration with Roswell Park Cancer Institute M.D. Anderson Cancer Center Mayo Clinic Hospital Clinic of Barcelona University College London Hospitals Churchill Hospital Case Western Reserve University University Hospital, Zürich	I, NCT02483858 and NCT01940133	41 patients were enrolled and administrated. As per two compartmental analysis, the absorption rate was 2.1 h^−1^, the absorption time lag was 0.4 h, the clearance rate was 4.2 L/h, the central volume was 84.4 L and the peripheral volume 211.9 L. The mean C_max_ at steady state for the 80 and 100 mg dosing groups were found to be 0.96 μg/mL and 1.46 μg/mL, respectively, and the mean AUC_24h_ at steady state was identified to be 16.28 μg h mL^−1^ and 23 μg h mL^−1^, respectively. The T_1/2_ was establsihed at 51 h. Compared to a typical 70 kg patient, the C_max_ was 27% higher for a 50 kg patient and 20% lower for a 100 kg patient at a steady state.	-
Bimiralisib in patients with advanced solid tumors [416]	Completed	PIQUR Therapeutics AG	I, NCT01940133	28 patients were enrolled and received QD Bimiralisib at range of 10–150 mg. The C_max_ and AUC_(0–last)_ increased with increasing doses of Bimiralisib in a roughly dose-proportional manner. The T_1/2_ was estimated to be ~40 h from the 0–24 h profile. The MTD and RP2D of Bimiralisib were defined as 80 mg QD for patients with activating *PIK3CA* mutations. There was no significant upregulation or downregulation of PI3K/mTOR-related mRNAs at transcriptional levels although there was a trend of *PDGFRA* upregulation post-21 days of Bimiralisib treatment. However, there was a significant inhibition of several PI3K/mTOR-associated phosphoproteins. Gr 3 or Gr 4 DLTs were seen in 13 (46%) and 3 (11%) patients, respectively.	Fatigue, nausea, hyperglycaemia, constipation, diarrhea, anorexia, rash and vomiting.
**Paxalisib/GDC-0084/RG7666**					
Paxalisib in patients with progressive or recurrent high-grade glioma [417]	Completed	Genentech, Inc	I, NCT01547546	47 patients enrolled in 8 dose-escalation cohorts (2-65 mg QD). DLTs observed included Gr 2 bradycardia, Gr 3 myocardial ischemia at 15 mg, Gr 3 stomatitis at 45 mg and 2 cases of Gr 3 mucosal inflammation at 65 mg. The MTD was established at 45 mg QD. PK analysis indicated linear and dose-proportional enhancement in exposure, with a T_1/2_ ~19 h associated with QD schedule. At steady-state concentrations (45 mg), consistent anti-tumor activity was observed in xenograft models. On FDG-PET, 5 of 27 (18.5%) evaluable patients showed a metabolic PR. At doses ≥45 mg QD, decreased median SUV in normal brain was observed, indicating CNS penetration of Paxalisib. Of all the evaluable patients, 26 patients (55.3%) had a best overall response of PD, 19 patients (40.4%) had SD.	Hyperglycemia, fatigue, nausea, rash, hypophosphatemia, hypertriglyceridemia, diarrhea, decreased appetite and stomatitis
**Voxtalisib/XL765/SAR245409**					
Voxtalisib in combination with Letrozole in HR+, HER2- MBC patients refractory to non-steroidal aromatase inhibitor [231]	Completed	Sanofi	I/II, NCT01082068	The combination had an acceptable safety profile (*n* = 35) but showed limited efficacy in these pateints. The MTD of Voxtalisib plus Letrozole was identified as 50 mg BID (Voxtalisib) and 2.5 mg QD (Letrozole). 1 DLT, Gr 3 rash was seen in Voxtalisib 50 mg BID treatment group. The median PFS and PFS rate at six months was 7.9 weeks (90% CI: 7.1–15.7 weeks) and 2% (90% CI: 1.4–22.3%), respectively. No patient had an objective response. 22% of patients had PFS at 24 weeks. 31% and 46.2% of patients had SD and PD, respectively. No association between efficacy and *PIK3CA* mutations was established. Voxtalisib did not show any drug–drug interaction with Letrozole.	Rash, increased AST/ALT, diarrhea, hyperglycemia, pruritus, lymphopenia, nausea, vomiting and fatigue.
Voxtalisib in patients with advanced solid tumors [418]	Completed	Sanofi	I, NCT00485719	Total *n* = 83 patients were enrolled and 52 patients received Voxtalisib 15–120 mg BID and 70–100 mg QD. Voxtalisib had a manageable safety profile and was efficient in inhibiting PI3K/mTOR signaling. The MTD was established at 90 mg QD or 50 mg BID. The median duration of treatment was 41 days (range: 4–371 days). 24% of patients demonstrated SD. Voxtalisib showed a relatively short plasma T_1/2_ of 2.96–7.52 h. Voxtalisib in both a once- or twice-a-day dosing schedule showed similar PK profiles. In addition, most of Voxtalisib got cleared from plasma in 12 or 24 h. The T_max_ of voxtalisib was identified at 1–4 h. Voxtalisib treatment also increased plasma insulin levels in these patient populations. pAKTT308, pAKTS473 and pEBP1 levels were reduced by 45% to 88%; 45% to 94% and 55% to 89%, respectively, in paired tumor biopsises. The best overall response was SD in 24 of 50 evaluable patients (48%) and 7 patients demonstrated tumor regression. No relationship could be establsihed between tumor molecular alterations and clinical outcomes associated with Voxtalisib treatment in this patient population.	Nausea, diarrhea, hyperglycemia, decreased appetite, fatigue, rash, dry skin, asthenia, vomiting and increased AST/ALT.
Voxtalisib in combination with Erlotinib in patients with advanced solid tumors [419]	Completed	Sanofi	I, NCT00777699	46 patients with advanced solid tumors were enrolled, which included *n* = 37 lung cancer patients who had prior anti-EGFR therapy. Voxtalisib was administrated in a 30, 50, 70, or 90 mg QD or 20/30 mg BID dosing schedule in combination with 100 mg QD Erlotinib, in a 28 day cycle. The median duration of treatment for Voxtalisib and Erlotinib was 56 (range, 0–346) and 70 (range, 9–360) days, respectively. MTDs were identified as 70 mg QD Voxtalisib in combination with 100 mg QD Erlotinib in cohort 1, and Voxtalisib (20 mg BID) plus Erlotinib (100 mg QD) in cohort 2. The MTDs in both cohorts were below the RP2D of Voxtalisib (90 mg QD or 50 mg BID) and below the registered dose of Erlotinib (150 mg QD). Best overall response was SD found in 12/32 (37.5%) of the patients. When Voxtalisib was administered in combination with Erlotinib, Voxtalisib was absorbed with median T_max_ and T_1/2_ of 1.6–2 h and 4.3–8.03 h, respectively. Increased dosing of Voxtalisib enhanced the AUC and C_max_ proportionally_._ Molecular profiling of tumor samples of these patients identified mutations in *PIK3CA*, *EGFR*, *KRAS*, *TP53* and LKB1 genes. Suppression of PI3K and EGFR/MAPK pathway biomarker expression were increased over time post-treatment as follows pAKTT308: 40–73%; p4EBP1T70: 43–67%; pEGFRY1045: 31–62%; and pERKT202/Y204: 37–75%.	Diarrhea, nausea, rash, stomatitis, vomiting, elevated AST/ALT and photophobia.
Voxtalisib in patients with solid tumors [420]	Completed	Sanofi	I, NCT01596270	49 patients were enrolled and received Voxtalisib treatment as follows: 3 patients in the 50 mg QD cohort, 17 patients in the 60 mg QD cohort, 12 patients in the 70 mg QD cohort, 7 patients in the 50 mg BID cohort, 6 patients in the 40 mg BID cohort and 4 patients in the 30 mg BID dosing cohort, respectively. The median duration of treatment in the QD cohorts was 5.9 weeks (range, 0.9–8.1) and the median duration of treatment in in the BID cohorts was 7.7 weeks (range, 0.9–8.1). The MTD of Voxtalisib was established as 40 mg BID and 60 mg QD. Of 31 and 16 patients evaluable for anti-tumor activity in the QD and BID cohorts, respectively, 9/31 patients (29%) and 8/16 patients (50%) had SD as best response. Voxtalisib exhibited a linear PK profile, with an increase in plasma concentration with increased dosing. Voxtalisib showed no significant accumulation owing to its short T_1/2_ of 3–4 h. Drug accumulation with BID dosing was greater versus QD dosing regimen. High variability in C_max_ and AUC values was observed. Administration of Voxtalisib with food decreased drug exposure. Overall, Voxtalisib had a manageable safety profile in these patients.	Fatigue, nausea, diarrhea, maculopapular rash, dyspnea, decrease in appetite, anemia and hyponatremia.
Voxtalisib in combination with a MEK inhibitor, Pimasertib in patients with advanced or metastatic solid tumors [421]	Completed	EMD Serono in collaboration with Sanofi	I, NCT01390818	146 patients with advanced cancer were treated, which included 63 patients in the dose escalation and 83 patients in the expansion part of the study. The MTD was identified as Pimasertib 90 mg QD and Voxtalisib 70 mg QD. The RP2D was established at 60 mg Pimasertib and 70 mg Voxtalisib. The best overall responses included a CR in one patient (1%), PR in 5 patients (5%), SD in 51 patients (46%) and PD in 45 patients (41%). At the RP2D, 74 patients needed dose interruption (73%), 20 patients had dose reduction (20%) and 26 had discontinued treatment because of TEAEs (26%). Both drugs were absorbed rapidly and had a median T_max_ ~ 1–2 h each. Voxtalisib PK profile demonstrated close proportionality as compared to Pimasertib in the daily regimens. The half-lives using daily dosing schedule were similar across both the groups in the dose-escalation part of the study. The half-lives were 4.3 to 6.1 h on day 1 and 5.5 to 7.2 h on day 15 for Pimasertib; and 3.1 to 3.6 h on day 1 and 3.4 to 5.2 h on day 15 for Voxtalisib treatment, respectively. There was variable accumulation across doses by day 15 post-QD Pimasertib and no known accumulation was seen with Voxtalisib administration. Pimasertib dosing reduced Voxtalisib exposure AUC_(0–t)_ by 11% (90% CI: 67.0–118.8) and Voxtalisib treatment enhanced Pimasertib exposure AUC_(0–t)_ upto ~49% (90% CI: 106.2–208.9). No CR or PR was achieved; therefore no conclusion could be drawn in terms of the correlation between clinical activity of Voxtalisib and mutational profile for predictive genomic markers involving *BRAF*, *KRAS*, *NRAS*, or *PIK3CA* mutationsin the patient dosed at RP2D. In addition, near complete suppression of both pERK and pS6 expression was identified across for both drugs at most of the dosing interval. Overall, the combination of Voxtalisib with Pimasertib demonstrated poor long-term tolerability and limited anti-tumor activity in these patients (*n* = 146).	TEAEs were diarrhea, fatigue and nausea.
Voxtalisib in combination with Temozolamide (TMZ) ± radiotherapy (RT) in patients high-grade glioma [422]	Completed	Sanofi	I, NCT00704080	Total *n* = 54 patients were enrolled and received Voxtalisib 30–90 mg QD or 20–50 mg BID in combination with 200 mg/m^2^ IV TMZ (*n* = 49), or Voxtalisib 20 mg QD with 75 mg/m^2^ IV TMZ and RT (*n* = 5). The combination had a favorable safety profile in these patients with DLTs reported in 7 patients (13%). The MTDs for Voxtalisib in combination with 200 mg/m^2^/day TMZ were established at 90 mg QD and 40 mg BID dosing. However, the MTD of Voxtalisib plus RT and TMZ treatment could not be determined. The best overall response included PR and SD in 2/47 (4%) and 32/47 (68%) of the patients, respectively evaluated for response. The T_max_ for Voxtalisib and Voxtalisib plus Temozolamide was 0.75–2 h (range: 0.5–4 h) and 1.5–6 h (range: 1–8 h), respectively. Compared with cycle 1 day 1, the mean C_max_ and AUC_τ_ on cycle 1 day 22 were 1.4-fold and 1.5-fold higher, respectively, for QD dosing and 1.7-fold and 1.4-fold higher, respectively, for BID dosing of Voxtalisib. The mean C_max_ of TMZ on cycle 1 day 1 post-200 mg/m^2^/day TMZ administration plus different doses of Voxtalisib, ranged from 3650 ng/mL to 8490 ng/mL and achieved a median T_max_ of 1.50–6.00 h (range 1.00–8.00 h). The mean TMZ AUC_(0–24)_ ranged from 24,800 to 35,900 ng × h/mL when administered in combination with Voxtalisib. Voxtalisib in combination with TMZ demonstrated moderate inhibition (31–49%) of PI3K/mTOR signaling.	Fatigue, nausea, diarrhea, maculopapular rash, thrombocytopenia, increased AST/ALT, lymphopenia and decreased platelet counts.
Voxtalisib in patients with recurrent glioma who were surgical resection candidates [423]	Completed	Sanofi	I, NCT01240460	Total *n* = 21 patients enrolled in cohort 1 (*n* = 6), 2 (*n* = 6) and 3 (*n* = 7), received Voxtalisb at 50 mg BID, 200 mg QD and 90 mg QD, respectively for >10 days prior to tumor resection and were evaluated for PK/PD analysis. PK analyses indicated a mean tumor to plasma ratio of 0.38 and 0.40 in cohorts 1 and 3 and 0.27 in cohort 2, respectively. Voxtalisib when administered in a 50 mg BID or 90 mg QD dosing schedule demonstrated higher distribution in CNS of these patients along with an acceptable safety profile. Treatment with Voxtalisib decreased pS6K1 levels and Ki67 expression as revealed by IHC staining.	-
Voxtalisib in combination with Rituximab and Bendamustine in patients with R/R B-cell malignancies [424]	Completed	Sanofi	I, NCT01403636	Total *n* = 37 patients enrolled and received Voxtalisib in combination with Rituximab plus Bendamustine in 3 treatment arms. Arm A: iNHL, MCL or CLL patients (*n* = 15) received Rituximab (375 mg/m^2^ IV) weekly on days 1, 8, 15 and 22 of a 28 day cycle for 2 cycles and increasing doses of Voxtalisib (30 or 50 mg BID orally). Arm B1: iNHL or MCL patients (*n* = 8) received fixed doses of Bendamustine (initially 90 mg/m^2^ IV subsequently reduced to 70 mg/m^2^ on days 1 and 2 and Rituximab (375 mg/m^2^ IV) on day 1 of each 28 day cycle for upto 8 cycles and increasing doses of Voxtalisib (30 or 50 mg BID orally). Arm B2: Patients (*n* = 12) with R/R CLL administrated with fixed doses of Bendamustine (70 mg/m^2^ IV) on days 1 and 2 and Rituximab (375 mg/m^2^ IV on day 1 of cycle 1 and 500 mg/m^2^ on day 1 of cycles 2 to 6) for upto 6 cycles, and increasing doses of Voxtalisib (30 or 50 mg BID orally). PK analysis: Voxtalisib exposure was similar for patients in all the 3 treatment arms, both after the first dose (cycle 1 day 1) and at steady state (cycle 2 day 1). Daily BID dosing did not cause accumulation of Voxtalisib in plasma of these patienst. Rituximab PK parameters were similar for patients in all 3 treatment arms. Voxtalisib did not pharmcokinetically interact with Rituximab or Bendamustine. The best overall response rate (ORR; CR + PR) was 48.6% for all patients. A total of 17 patients (48.6%) were reported as progression free at six months. The median PFS was 32.1 weeks for all patients. Arm A: The median PFS was 33.4 weeks (95% CI: 26.3–60.1 weeks). The ORR was 40%. The CR, PR and SD were 6.7%, 33.3% and 46.7%, respectively. Arm B1: The median PFS was 33.4 weeks (95% CI: 0.7–53.4 weeks). The ORR was 50%. The CR, PR and SD were 12.5%, 37.5% and 37.5%, respectively. Arm B2:The median PFS was 26.1 weeks (95% CI: 8.4–44 weeks). The ORR was 58.3%. The CR, PR and SD were 16.7%, 41.7% and 33.3%, respectively. The RP2D for Voxtalisib in combination with both drugs was established at 50 mg BID. This drug combination demonstrated activity in heavily pretreated patients.	Headache, fatigue, neutropenia, pyrexia, constipation, rash, vomiting, anemia, thrombocytopenia and decreased appetite.
Voxtalisib in patients with relapsed or refractory NHL or CLL [425]	Completed	Sanofi	II, NCT01403636	Total 167 patients were enrolled, which included 42 patients with MCL, 47 with FL, 42 with DLBCL and 36 patients with CLL/SLL. These patients received 50 mg BID Voxtalisib orally. Voxtalisib showed an acceptable safety profile and was efficacious in the patients with FL but demonstrated limited efficacy in patients with MCL, DLBCL or CLL/SLL. The median duration of treatment was 10.7 weeks (IOQ: 5.9–31.9 weeks). The ORR for all patients was 18.3% (95% CI: 12.7–25.1%). The overall CR, PR and SD for all patients were 5% and 13.4% and 33.5%, respectively. The median PFS was 14.4 weeks (95% CI: 9–19.43 weeks). 32.3% of patients had a PFS of more than 24 weeks. The main reasons for treatment discontinuation were disease progression (overall 107 patients of 167 (64.1%); 73.8% (31/42) in the MCL, 42.6% (20/47) in the FL, 76.2% (32/42) in the DLBCL and 66.7% (24/36) in the CLL/SLL groups, respectively. The clinical response seen in patients with different cancers is mentioned below: MCL patients: The median duration of treatment was 8 weeks (IQR: 5.6–19.9 weeks). The ORR was 11.9% (95% CI: 4–25.6%). The CR, PR and SD were 7.1%, 4.8%, 33.3%, respectively. The median PFS was 8.9 weeks (95% CI: 7.86–12.86 weeks). 21.4% of patients had a PFS of more than 24 weeks. FL patients: The median duration of treatment was 29 weeks (IQR: 8.1–64 weeks). The ORR was 41.3% (95% CI: 27–56.8%). The CR, PR and SD were 10.9%, 30.4% and 30.4%, respectively. The median PFS was 58 weeks (95% CI: 26 weeks–not calculated). 54.3% of patients had a PFS of more than 24 weeks. DLBCL patients: The median duration of treatment was 6 weeks (IQR: 3.6–8.1 weeks). The ORR was 4.9% (95% CI: 0.6–16.5%). The PR and SD were 4.9% and 9.8%. No patients achieved CR. The median PFS was 7.1 weeks (95% CI: 5.14–8.14 weeks). 7.3% of patients had a PFS of more than 24 weeks. CLL/SLL patients: The median duration of treatment was 19.8 weeks (IQR: 11.8–36.9 weeks). The ORR was 11.4% (95% CI: 3.2–26.7%). The PR and SD were 11.4% and 65.7%. No patients achieved CR. The median PFS was 24.1 weeks (95% CI: 16.57–31.57 weeks). 45.7% of patients had a PFS of more than 24 weeks. No genetic alterations were established in response to Voxtalisib treatment in any patient samples.	Fatigue, diarrhea, nausea, cough, dyspnea, pyrexia, anemia, decreased appetite, pneumonia, increased AST/ALT and thrombocytopenia.

## 6. Conclusions

Resistance to current PI3K therapies continues to push the need to expand our understanding of cancer treatment. Very few PI3K inhibitors are approved by the FDA as PI3K inhibitors suffer from many adverse effects, and poor solubility and permeability. However, these issues are addressed via improved PK techniques. Designing new PI3K inhibitors based on variable structural and binding sites (mutant selective inhibitors) could be used to increase selectivity and reduce toxicity for isoform-specific PI3K inhibitors. Targeting other pathways with ERK, EGFR, CDK4/6, SGLT2 inhibitors, anti-estrogen therapies or HDAC as a combinatorial therapy approach could increase the efficacy for PI3K inhibitors. The mechanism of action which includes the synergistic combination of PI3K with mTOR inhibitors must be explored for better targeting of the PI3K pathway. In this context, Yang et al. have already shown that addition of Dactolisib, a dual PI3K/mTOR inhibitor or ZSTK474, a pan-PI3K inhibitor, in combination with Temsirolimus, a mTORC1 inhibitor, synergistically inhibited cancer cell growth and overcame cellular resistance to Temsirolimus regardless of PTEN status [426]. Emerging evidence in a study by Tang et al. suggested that a combinatorial approach using dual PI3K/mTOR inhibitors could achieve better anti-cancer outcomes [427]. Other factors that could overcome the challenges to PI3K inhibitors in the clinic include selection of accurate patient population with activating *PIK3CA* mutations, optimizing the dosing schedules, etc. Another challenge in the clinic is a lack of selective PI3Kγ inhibitors. The biggest issue in developing small-molecule inhibitors against PI3Kγ (member of PI3K class IB) is to obtain inhibitors that have good selectivity for PI3Kγ against other PI3K isoforms as PI3Kγ show high sequence similarity in the conserved ATP-kinase-binding site with class I isoforms, leading to difficulties in developing specific γ inhibitors. Further research is needed to understand whether using a dual or multi-targeted approach inhibiting two or more PI3K enzymes or direct inhibition of PI3Kγ could help to achieve maximal efficacy against these inhibitors [428]. In addition, there are several new P13K drugs such as RIDR-PI-103, B591 and TG-100–115 with limited in vitro and in vivo studies which need to be further explored using PK and PD models so that it could be translated to clinic.

## Figures and Tables

**Figure 1 ijms-22-03464-f001:**
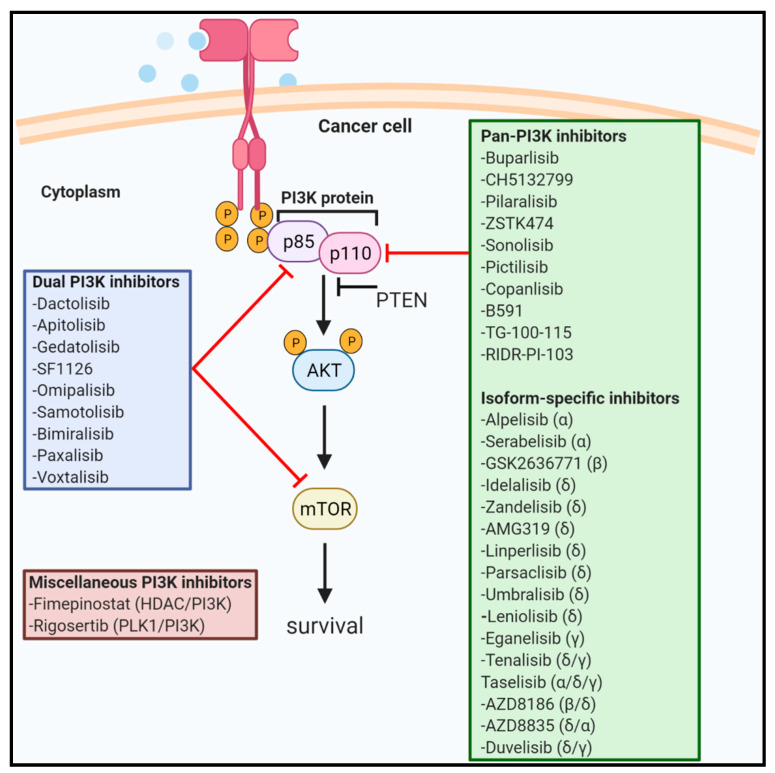
PI3K inhibitors: pan, isoform-specific and dual PI3K/mTOR inhibitors to treat patients with different cancers (created with Biorender; www.biorender.com accessed on 20 March 2021).

## Data Availability

Not applicable.

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
