# Peer review of "PI3K Inhibitors in Cancer: Clinical Implications and Adverse Effects"

_ijms, 2021, doi:10.3390/ijms22073464_

Round 1

Reviewer 1 Report

The manuscript contains too many details of clinical trials whose data are not summarized. 

I would have much liked to see how many cpds of the PI3Ki only class (on which this review focus on) have been approved and how many cpds have despite many clinical trials not made it. just summarizing the clinical trial design with some AEs is not very helpful.

Would have started to describe the approved cpds in each category (pan vs selective) etc.

The list of selective cpds is not complete.  Leniolisib and Eganalisib are missing among others !!      

Also i would have mentioned the fact that some of these cpds have been tested outside cancer like alpelisib in PROS and leniolisib in ADPS.

I do not see a great value in summarizing clinical trials without more comments.

Would have mentioned the lack of selective  PI3Kgamma inhibitors

Also i would have mentioned the dual inhbitors (PI3K-mTOR) as well as selective mTOR inhbitors in cancer

Author Response

We sincerely appreciate the critique and input from the reviewers as well as the opportunity you give us to address their concerns and resubmit our work. We will address below point by point issues/concerns raised in the Critique. All changes made to the manuscript are under track changes.

Reviewer #1 comments (shown in bold):

I would have much liked to see how many cpds of the PI3Ki only class (on which this review focus on) have been approved and how many cpds have despite many clinical trials not made it. just summarizing the clinical trial design with some AEs is not very helpful.

Answer: We thank reviewer for these comments. We have specified about the drugs approved by FDA (Line 63).  To date, five PI3K inhibitors (copanlisib, idelalisib, umbralisib, duvelisib and alpelisib) have been approved by United States Food and Drug Administration (FDA).  Copanlisib is a pan PI3K inhibitor and idelalisib, umbralisib, duvelisib and alpelisib belong to isoform specific inhibitors. All these drugs are highlighted in yellow in the specific categories of PI3K inhibitor.

Would have started to describe the approved cpds in each category (pan vs selective) etc.

Answer: We thank the reviewer for the valuable feedback. Under each category of pan and isoform PI3K inhibitors, we have listed which of these drugs are approved by the FDA and highlighted in yellow the manuscript: Copanlisib (Ref 168 and 169) in pan PI3K drugs, Idelalisib (Ref 272), umbralisib (Ref 296), duvelisib (Ref 305, 306) and alpelisib (Ref 254, 255)  under isoform category.

For clarity we have included here:

Line 427: Out of all pan PI3K inhibitors, only copanlisib has been approved by the FDA.

Line 608: Copanlisib was brought into market by Bayer and approved by FDA for treatment of patients with relapsed follicular lymphoma (FL).

Line 872: Alpelisib was approved by FDA for PIK3CA-mutated, HR+, HER2- advanced breast cancer who had received endocrine therapy previously.

Line 939: Idelalisib/GS-1101/CAL-101/Zydelig, an oral PI3Kδ inhibitor with IC50 value of 2.5 nM in cell-free assays was the first FDA approved PI3K inhibitor for treatment of R/R CLL or LL/FL progressed on prior therapies.

Line 1025 Umbralisib is recently approved by FDA for treatment of patients with R/R marginal zone lymphoma (MZL) who had at least one prior anti-CD20-based therapy and for adults with R/R follicular lymphoma (FL) relapsed on three prior lines of systemic therapy .

Line 1055: Duvelisib (September 2018, NDA 311155) was approved by the FDA for treatment of patients with R/R CLL/ SLL.

The list of selective cpds is not complete.  Leniolisib and Eganalisib are missing among others !! 

Answer: We thank the reviewer for the constructive feedback. We have added Leniolisib in section 4.15 and Eganalisib in section 4.16 under isoform-specific categories of PI3K inhibitors along with clinical outcome in Table 2 (Ref 369).

Also i would have mentioned the fact that some of these cpds have been tested outside cancer like alpelisib in PROS and leniolisib in ADPS.

Answer: We have added the following to address this concern:

Line 867: The role of alpelisib has also been identified outside cancer in PIK3CA-related overgrowth syndromes (PROS) which are caused by mosaic gain-of-function mutations in the PIK3CA gene. It is shown that daily oral administration of alpelisib in the mouse model of PROS post tamoxifen administration, effectively suppressed PIK3CA pathway activation, which prevented the PROS phenotype and enhanced survival [253].

Line 1063: Leniolisib/CDZ173 is a potent and selective PI3Kδ inhibitor. Most of the studies using leniolisib are sponsored by Novartis Pharmaceuticals. Although no studies are known in cancer but it is implicated in immunodeficiency disorders such as activated pi3kδ syndrome/p110δ-activating mutation causing senescent          T cells, lymphadenopathy and immunodeficiency (APDS/PASLI) where it is tested in phase II/III trails. Rao et al have shown that treatment with leniolisib resulted in dose-dependent inhibition of PI3Kδ pathway hyperactivation in cells expressing APDS-causative p110δ variants and in T-cell blasts derived from patients. In addition it was shown that oral administration of leniolisib in six APDS patients resulted in dose-dependent inhibition of PI3K/AKT pathway evaluated via ex-vivo assay and had enhanced immune dysregulation. Thus, leniolisib is known to be well tolerated and with improved laboratory and clinical parameters in patients with APDS, supporting the specific inhibition of PI3Kδ as a promising new targeted therapy in patients with APDS and other diseases characterized by hyperactivation of the PI3Kδ pathway [309]. Leniolisib is also tested in patients with primary sjÖgren’s syndrome (PSS) in phase II trials where its oral administration resulted in inhibition of phosphorylated AKT in ex-vivo stimulated B cells, decreased the serum CXCL13 levels and reduced the frequency of circulating follicular T helper-like cells [310].

I do not see a great value in summarizing clinical trials without more comments.

Answer: We are discussing a very complex topic discussing all PI3K inhibitors in all cancer types and have tried our best to include most detail information about each inhibitor in the clinical trials. We have also added dual PI3K/mTOR inhibitors as suggested.

Would have mentioned the lack of selective PI3Kgamma inhibitors.

Answer: We thank the reviewer for these comments. We have included the rationale for lack of PI3Kγ inhibitors in the clinic in the conclusion section of the manuscript (Ref 428).

Also i would have mentioned the dual inhibitors (PI3K-mTOR) as well as selective mTOR inhbitors in cancer.

Answer: We have added dual inhibitors (PI3K-mTOR) in the manuscript. The detail clinical outcomes have been described in Table 3. We are not discussing selective mTOR inhibitors as this is outside of scope of our manuscript.

We hope that this rebuttal, changes, and additions have improved our paper to a point in which it is acceptable for reevaluation by the Editorial Board of International Journal of Molecular Sciences. Thank you very much for your time and consideration.

Sincerely,

Joan T. Garrett

Reviewer 2 Report

The review of Mishra et al. focuses on the various PI3K inhibitors, on the treatment outcomes of the clinical trials in different cancers and the adverse events leading to treatment failure.

Overall, even if the review discusses a complex topic, it is clearly written and it reads very well. The Figure and Tables are very well organized and presented. However, the readability could be improved by using more punctuation, especially in the long sentences.

Some minor edits are suggested below.

  • The paragraph 2.1. “Inactivation or loss of PTEN activity” is skimpy, so the author could extend the description with further information, in example Juric D et al. (2015) treats as loss of PTEN leads to clinical resistance to a PI3K inhibitor.
  • In the “Conclusion” paragraph the authors should insert the mechanism for synergy with combined mTOR and PI3 kinase inhibitors that is missing, as previously described in example by Yang S. et al.(2011) and also add the emerging evidences suggesting that a class of dual PI3K/mTOR inhibitors, which bind to and inactivate both PI3K and mTOR, may achieve better anti-cancer outcomes (Tang KD et al., 2014).
  • There are some typo errors.

At lines 192, 206, 211, 220 substitute LncRNA with lncRNA.

At line 304 substitute Buparlisib/BKM120 with Buparlisib /NVP-BKM120/BKM120 and at 578 substitute YY-20394 with YY-20394/ Linperlisib.

Author Response

We sincerely appreciate the critique and input from the reviewers as well as the opportunity you give us to address their concerns and resubmit our work. We will address below point by point each of the issues/concerns raised in the Critique. All changes made to the manuscript are under track changes.

Reviewer #2 comments (shown in bold)

Overall, even if the review discusses a complex topic, it is clearly written and it reads very well. The Figure and Tables are very well organized and presented. However, the readability could be improved by using more punctuation, especially in the long sentences.

Answer: We thank the reviewer for the appreciation and we have modified the sentences as per the comment.

The paragraph 2.1. “Inactivation or loss of PTEN activity” is skimpy, so the author could extend the description with further information, in example Juric D et al. (2015) treats as loss of PTEN leads to clinical resistance to a PI3K inhibitor.

Answer: We thank the reviewer for this thoughtful comment. We have added the following sentence to the manuscript: A study by Juric et al indicates that loss of PTEN can lead to clinical resistance to a PI3K inhibitor, alpelisib in breast cancer [20].

In the “Conclusion” paragraph the authors should insert the mechanism for synergy with combined mTOR and PI3 kinase inhibitors that is missing, as previously described in example by Yang S. et al. (2011) and also add the emerging evidences suggesting that a class of dual PI3K/mTOR inhibitors, which bind to and inactivate both PI3K and mTOR, may achieve better anti-cancer outcomes (Tang KD et al., 2014).

Answer: We thank the reviewer for this valuable feedback. We have added these two references: Yang S. et al. (2011): Ref  426 and Tang KD et al., 2014 : Ref 427 in the conclusion section of the manuscript. We have indicated the role of dual PI3K/mTOR inhibitors in overcoming resistance to PI3K inhibitors in the conclusion section of the manuscript.

There are some typo errors. At lines 192, 206, 211, 220 substitute LncRNA with lncRNA.

Answer: We have made these edits which are highlighted using track changes option in the manuscript.

At line 304 substitute Buparlisib/BKM120 with Buparlisib /NVP-BKM120/BKM120 and at 578 substitute YY-20394 with YY-20394/ Linperlisib.

Answer: We have made the substitution at the specific lines and these changes are highlighted in the manuscript using track changes options.

We hope that this rebuttal, changes, and additions have improved our paper to a point in which it is acceptable for reevaluation by the Editorial Board of International Journal of Molecular Sciences. Thank you very much for your time and consideration.

Sincerely,

Joan T. Garrett

Round 2

Reviewer 1 Report

the revision are Ok with me